# Development of the CMA-GFS-AERO 4D-Var assimilation system v1.0- Part 1: System description and preliminary experimental results

Yongzhu Liu[1,2,3], Xiaoye Zhang[2,4], Wei Han[1,2,3], Chao Wang[1,2,3], Wenxing Jia[2,4], Deying Wang[2,4], Zhaorong Zhuang[1,2,3], Xueshun Shen[1,2,3]

[1]CMA Earth System Modeling and Prediction Centre, China Meteorological Administration, Beijing, 10081, China

[2]State Key Laboratory of Severe Weather, China Meteorological Administration, Beijing, 10081, China

[3]Key Laboratory of Earth System Modeling and Prediction China Meteorological Administration, China Meteorological Administration, Beijing, 10081, China

[4]Key Laboratory of Atmospheric Chemistry of CMA, Chinese Academy of Meteorological Sciences, Beijing, 10081, China

*Correspondence to:* Xiaoye Zhang (xiaoye@cma.gov.cn), Xueshun Shen (shenxs@cma.gov.cn)

**Abstract.** We developed a strongly coupled aerosol-meteorology four-dimensional variational (4D-Var) assimilation system, CMA-GFS-AERO 4D-Var, for investigating the feedback of aerosol data assimilation on meteorological forecasts. This system was developed on the basis of the framework of the incremental analysis scheme of the China Meteorological Administration Global Forecasting System (CMA-GFS). CMA-GFS-AERO 4D-Var includes three component models: forward, tangent linear, and adjoint models. CMA-GFS-AERO forward model was constructed by integrating an aerosol module containing main physical processes of black carbon (BC) aerosol in the atmosphere into the CMA-GFS weather model. The tangent linear and the adjoint of the aerosol module was further developed and coupled online with the CMA-GFS tangent linear and adjoint models, respectively. In CMA-GFS-AERO 4D-Var, the BC mass concentration was used as the control variable and minimized together with atmospheric variables. The validation of this system includes the tangent linear approximation, the adjoint correctness test, the single observation experiment and the full observation experiment. The results show that CMA-GFS-AERO tangent linear model performs well in tangent linear approximation for BC, and adjoint sensitivity agrees well with tangent linear sensitivity. Assimilating BC observations can generate analysis increments not only for BC but also for atmospheric variables, highlighting the capability of CMA-GFS-AERO 4D-Var in exploring the feedback of BC assimilation on atmospheric variables. The computational performance of CMA-GFS-AERO 4D-Var also indicates the potential in operational application. This study focuses on the theoretical architecture and practical implementation of the system, the detailed analysis of a series of cycling assimilation experiments will be described in part 2 of this paper.

## 1 Introduction

Coupled chemistry meteorology models (CCMM) are atmospheric chemistry models that concurrently simulate meteorological processes and chemical transformations (Zhang, 2008; Baklanov et al., 2014; Bocquet, 2015). They are more

recent compared to chemical transport models (CTM), which rely on meteorological fields as inputs (Seinfeld and Pandis, 1998). CCMM account for the feedback mechanism between aerosols and meteorology, specifically the moisture and temperature perturbations resulting from aerosol microphysics and radiative forcing, which, in turn, affect atmospheric dynamics such as convection, circulation, and stability, whereas CTM lack the capability to incorporate these feedback mechanisms (Guerrette and Henze, 2015).

CCMM provide the possibility to assimilate both meteorological and chemical data, enabling the production of an optimal initial condition for improving air quality predictions and developing re-analysis of three-dimensional (3D) chemical concentrations over the past decades (Bocquet, 2015). One of the first applications of data assimilation with a CCMM was conducted at Météo-France. Semane et al. (2009) used four-dimensional variational (4D-Var) data assimilation to assimilate the vertical profiles of ozone ($O_3$) concentrations obtained from the Microwave Limb Sounder (MLS) aboard the Aura satellite into the ARPEGE/MOCAGE (Action de Recherche Petite Echelle Grande Echelle/Modèle de Chimie Atmosphérique de Grande Echelle) chemistry meteorology integrated system, and found that the assimilation of $O_3$ reduces the wind bias in the lower stratosphere. This general approach is also adopted by the European Centre for Medium-range Weather Forecasts (ECMWF), although without considering the influence of chemical species on meteorological variables (Flemming et al.,2011; Inness et al., 2013). Flemming et al. (2011) utilized the 4D-Var system of the Integrated Forecast System (IFS) coupled with three different $O_3$ chemistry mechanisms, including a linear chemistry, the MOZART3 (Model for Ozone and Related Chemical Tracers, version 3) chemistry, and the TM5 (Transport Model, version 5) chemistry, to assimilate $O_3$ data from four satellite-borne sensors to improve the simulation of the stratospheric $O_3$ hole in 2008. Previous efforts have also explored the application of ensemble-based methods for data assimilation with a CCMM (Pagowski and Grell, 2012; Bocquet et al., 2015). Pagowski and Grell (2012) assimilated surface measurements of fine aerosols using the Weather Research and Forecasting-Chemistry model (WRF-Chem) and the Ensemble Kalman filter (EnKF) method. Bocquet et al. (2015) also presented an application of the EnKF to assimilate surface fine particulate matter observations and meteorological observations with the WRF-Chem model over the eastern part of North America. Results demonstrated that a large positive impact of aerosol data assimilation on aerosol concentrations, while the effect of meteorological observation assimilation on aerosol concentration is rather minor. All the preceding studies have laid good foundations for data assimilation with CCMM. However, since CCMM are fairly recent, the development and applications of data assimilation in CCMM are still limited. Further research and more attention are required, especially in terms of the potential feedback of chemical data assimilation on meteorological forecasts.

Additionally, EnKF estimates background error covariance through ensemble forecasts, which rely on a limited number of ensemble members (Zhu et al., 2022). In high-dimensional problems, the limited number of samples may not be able to fully capture all the error characteristics, resulting in inaccuracies in the estimation of background error covariance. Although

ensemble Kalman smoothers (EnKS) extend the EnKF framework by incorporating an assimilation window to leverage temporal observational information, they remain constrained by similar limitations in ensemble size. In contrast, 4D-Var explicitly integrates both the complete observational dataset and the full model dynamics within the assimilation window to constrain state evolution, rather than relying solely on ensemble statistics. This generally allows 4D-Var to achieve higher accuracy in high-dimensional problems by making better use of both observational data and model constraints, leading to more precise state estimation. While the flow dependence of the background error covariance is implicitly realized within the assimilation window in 4D-Var, modeling the cross-variable component of the covariance presents a significant challenge in data assimilation for CCMM. Furthermore, the tangent linear model (TLM) and the adjoint model (ADM) are essential components of 4D-Var, but their development is often fraught with difficulties.

Significant efforts have been made in the field of atmospheric chemistry adjoint modeling. Elbern and Schmidt (1999) first constructed the ADM of a 3D CTM, EUARD (The University of Cologne European Air Pollution Dispersion Chemistry Transport Model). Inspired by this work, various ADM of CTM have been successively developed, mainly including CHIMERE (Menut et al., 2000; Vautard et al., 2000; Schmidt and Martin, 2003), IMAGES (Intermediate Model of Global Evolution of Species; Müller and Stavrakou, 2005), STEM-III (Sulfur Transport Eulerian Model; Sandu et al., 2005), CAMx (Comprehensive Air Quality Model with Extensions model; Liu, 2005), CMAQ (Community Multiscale Air Quality model; Hakami et al., 2007) and GEOS-Chem (Henze et al., 2007). An et al. (2016) and Wang et al. (2022) constructed the ADM of GRAPES-CUACE (Global/Regional Assimilation and PrEdiction System coupled with CMA Unified Atmospheric Chemistry Environmental Forecasting System), an independently developed CCMM in China (Wang et al., 2010, 2018). ADM of these widely used CTM play an important role in inverse modelling and chemical data assimilation (Menut et al., 2000; Müller and Stavrakou, 2005; Sandu et al., 2005; Hakami et al., 2007; Henze et al., 2009). However, these CTM do not take into account the influence between chemical species and meteorological variables, resulting in certain uncertainties in adjoint sensitivity, which in turn affects the effectiveness of 4D-Var. Although GRAPES-CUACE is a CCMM, its ADM only includes the adjoint of the chemical model and not the adjoint of the meteorological model, leading to uncertainties in the sensitivity calculation as well.

Black carbon (BC) aerosol, a major component of the fine particulate matter ($PM_{2.5}$) defined by an aerodynamic diameter of 2.5 micrometers or less, primarily originates from the incomplete combustion of biomass and fossil fuels (Kuhlbusch, 1998). As an important atmospheric pollutant, BC is porous and adsorbs other solid and gaseous pollutants (e.g., $SO_2$, $O_3$, etc.), and provides catalytic conditions for them, which plays an important role in photochemical and heterogeneous reactions and gas-particle conversion processes (Koch, 2001). BC is also the main optically absorbing component of atmospheric aerosols, effectively absorbing solar radiation in the visible to infrared wavelength range, thus affecting the temperature field throughout the atmosphere, including the surface temperature. The climatic effects of BC have been widely

reported, but the extent to which it affects weather forecasting requires further investigation (Chung and Seinfeld, 2002;

Menon et al., 2002; Bond et al., 2013).

To deeply investigate the feedbacks of aerosol data assimilation on meteorological forecasts, we utilized BC as a starting point to develop the strongly coupled aerosol-meteorology 4D-Var system. Firstly, we constructed a coupled aerosol-meteorology system, named CMA-GFS-AERO, by integrating an aerosol module (AERO-BC) containing main aerosol physical processes of BC in the atmosphere into the operational version of the weather model CMA-GFS V4.0 (Shen

et al., 2023), which was developed by the China Meteorological Administration (CMA). Then, the tangent linear and the adjoint of the AERO-BC module was constructed and coupled online with the TLM and ADM of CMA-GFS (Liu et al., 2017, 2023; Zhang et al., 2019), respectively. Thus, CMA-GFS-AERO ADM includes not only the adjoint of physical processes of BC, but also the adjoint of the meteorological model. Moreover, the BC adjoint variables and the meteorological adjoint variables mutually influence each other throughout the adjoint integration process, leading to a

notable enhancement in the precision of adjoint sensitivity of aerosol and meteorology state. Based on the CMA-GFS-AERO forward model and its TLM and ADM, we further constructed the CMA-GFS-AERO 4D-Var by adding BC as a control variable into the incremental analysis scheme of CMA-GFS 4D-Var. The rationality and capability of CMA-GFS-AERO 4D-Var in capturing the feedbacks of aerosol data assimilation on meteorological analysis were verified using the single observation experiment and the full observation experiment. The following part is divided into five sections. Section 2

introduces the methods, Section 3 describes the development of CMA-GFS-AERO 4D-Var, Section 4 provides the model setup, Section 5 presents the results, and the conclusions are found in Section 6.

## 2  Methodology

### 2.1  Model description

#### 2.1.1  CMA-GFS

The China Meteorological Administration Global Forecasting System (CMA-GFS, formerly known as GRAPES-GFS) is an operational global numerical weather model independently developed by the CMA (Chen and Shen, 2006; Chen et al., 2008; Shen et al., 2023). For this work, we used CMA-GFS version 4.0 (CMA-GFS v4.0). The dynamic core of CMA-GFS utilizes the fully compressible non-hydrostatical equations formulated on spherical coordinate with latitude and longitude, and adopts the height-based, terrain-following coordinate which is shown in Fig. S1 (Yang et al., 2007). The model employs

semi-implicit and semi-Lagrangian in two-level time integration (Yang et al., 2007). The spatial differential adopts Arakawa-C grid in the horizontal, and Charney-Philips variable staggering in the vertical. The large-scale transport processes utilize a hybrid Piecewise Rational Method (PRM) and Quasi-Monotone Semi-Lagrangian (QMSL) scheme (Su et al., 2013). The physical parameterization schemes used in this work are consistent with those adopted in the operational application of

CMA-GFS v4.0, which have been proven to perform well in global numerical weather prediction. The selected schemes mainly include the Simplified Arakawa Schubert (SAS) cumulus convection scheme (Arakawa and Schubert, 1974; Liu et al., 2015), the double-moment cloud microphysics scheme (Liu et al., 2003a, 2003b; Li et al., 2024), the Rapid Radiative Transfer Model for the GCM (RRTMG) longwave and shortwave radiation schemes (Mlawer et al., 1997; Morcrette et al., 2008), the Common Land Model (CoLM) land surface scheme (Dai et al., 2003), and the New Medium Range Forecast (NMRF) boundary layer scheme (Hong and Pan, 1996; Han and Pan, 2011). The state variables of the CMA-GFS nonlinear model (NLM) include non-dimensional pressure ($\pi$), potential temperature ($\theta$), the east-west component of horizontal wind ($u$), the north-south component of horizontal wind ($v$), the vertical component of wind ($\widehat{w}$), and the specific humidity ($q$).

### 2.1.2 CUACE

CUACE (CMA Unified Atmospheric Chemistry Environmental Forecasting System) is an air quality model developed by the Chinese Academy of Meteorological Sciences to study both air quality forecasting and climate change (Gong and Zhang, 2008; Wang et al., 2010; Zhou et al., 2012). CUACE mainly includes three modules: the aerosol module, the gaseous chemistry module and the thermodynamic equilibrium module. CUACE adopts CAM (Canadian Aerosol Module; Gong et al., 2003), which employs the size-segregated multicomponent aerosol algorithm, as its aerosol module. CAM involves six types of aerosols: BC, sulfate (SF), nitrate (NI), sea salt (SS), organic carbon (OC) and soil dust (SD), and each of them utilizes the sectional representation method (Gelbard et al., 1980; Meng et al., 1998; Gong et al., 2003), in which the aerosol size distribution is generally approximated by a set of contiguous, nonoverlapping and discrete size bins, to represent particle size distributions. The core of CAM is the major aerosol processes in the atmosphere, including hygroscopic growth, coagulation, nucleation, condensation, dry deposition/sedimentation, and below-cloud scavenging.

### 2.2 Incremental 4D-Var

The CMA-GFS 4D-Var data assimilation system has been in operation at CMA since 1 July 2018 (Zhang et al., 2019). CMA-GFS 4D-Var applies the incremental analysis scheme proposed by Courtier et al. (1994). The cost function is defined as

$$J(\delta\mathbf{x}) = \frac{1}{2}\delta\mathbf{x}^T\mathbf{B}^{-1}\delta\mathbf{x} + \frac{1}{2}\sum_{i=0}^{n}(\mathbf{H}_i\mathbf{M}_{0\rightarrow i}\delta\mathbf{x} + \mathbf{d}_i)^T\mathbf{R}_i^{-1}(\mathbf{H}_i\mathbf{M}_{0\rightarrow i}\delta\mathbf{x} + \mathbf{d}_i) + J_c, \tag{1}$$

where $\delta\mathbf{x} = \mathbf{x}_a - \mathbf{x}_b$ represents the analysis increment of the model variables, $\mathbf{x}_a$ is the analysis field, $\mathbf{x}_b$ is the background state, $\mathbf{d}_i = H_iM_{0\rightarrow i}(\mathbf{x}_b) - \mathbf{y}_i$ is the observation innovation at time $i$, $\mathbf{y}_i$ is the observation at time $i$, $H_i$ represents the observation operator at time $i$, $M_{0\rightarrow i}$ denotes the nonlinear model integration from the analysis time to time $i$, $\mathbf{H}_i$ is the linear operator corresponding to $H_i$, $\mathbf{M}_{0\rightarrow i}$ is the linear model corresponding to $M_{0\rightarrow i}$, $\mathbf{B}$ represents the error covariance matrix of $\mathbf{x}_b$, $\mathbf{R}_i$ denotes the observation error covariance matrix at time $i$, and $J_c$ is the weak constraint term on the basis of the digital filter. $J_c$ is not relevant to the current work, so the formula described below omits $J_c$ term from

the cost function for the sake of simplicity.

After the physical and preconditioning transformations of the control variables, the cost function can be expressed as (Courtier et al., 1994; Lorenc et al., 2000; Zhang et al., 2019)

$$J(\mathbf{w}) = \frac{1}{2}\mathbf{w}^T\mathbf{w} + \frac{1}{2}\sum_{i=0}^{n}(\mathbf{H}_i\mathbf{M}_{0\to i}\mathbf{U}\mathbf{w} + \mathbf{d}_i)^T\mathbf{R}_i^{-1}(\mathbf{H}_i\mathbf{M}_{0\to i}\mathbf{U}\mathbf{w} + \mathbf{d}_i), \tag{2}$$

where $\mathbf{w}$ denotes the control variables after the physical and preconditioning transformations, and the analysis increment is expressed as $\delta\mathbf{x} = \mathbf{U}\mathbf{w}$, $\mathbf{U}$ ($\mathbf{U}\mathbf{U}^T = \mathbf{B}$) is the square root matrix of the background error covariance matrix.

The gradient of the cost function $J(\mathbf{w})$ with respect to the control variable $\mathbf{w}$ is

$$\nabla_{\mathbf{w}}J = \mathbf{w} + \sum_{i=0}^{n}\mathbf{U}^T\mathbf{M}_{0\to i}^T\mathbf{H}_i^T\mathbf{R}_i^{-1}(\mathbf{H}_i\mathbf{M}_{0\to i}\mathbf{U}\mathbf{w} + \mathbf{d}_i), \tag{3}$$

where $\mathbf{H}_i^T$ is the adjoint operator of $\mathbf{H}_i$, and $\mathbf{M}_{0\to i}^T$ is the adjoint model of $\mathbf{M}_{0\to i}$, which denotes the backward integration of the ADM from the time $i$ to the analysis time.

Currently, the CMA-GFS 4D-Var system adopts a 6-h cycle and is performed four times a day, with assimilation windows of 0300 UTC-0900 UTC, 0900 UTC-1500 UTC, 1500 UTC-2100 UTC and 2100 UTC-0300 UTC. The assimilation process is divided into two parts: the outer loop and the inner loop. In the outer loop, the CMA-GFS NLM ($\mathbf{M}_{0\to i}$) is integrated at high resolution for 6 hours to obtain the trajectory, which is a collection of stored values of all model state variables at all time steps within the assimilation window. The observation innovation $\mathbf{d}_i$ is calculated in the outer loop as well. In the inner loop, the CMA-GFS TLM and ADM are integrated at low resolution to calculate the cost function ($J(\mathbf{w})$) and its gradient ($\nabla_{\mathbf{w}}J$). The gradient is further provided to the Lanczos-CG algorithm (Lanczos, 1950; Liu et al., 2018) to perform the minimization, obtaining the optimal analysis increments to control variables.

## 3 Development of CMA-GFS-AERO 4D-Var

The computational cost is an important factor to be considered when developing a coupled aerosol-meteorology 4D-Var system with potential for operational application (Flemming et al., 2015). The CUACE model is computationally expensive since it includes more than one hundred chemical variables for aerosols and gases, as well as hundreds of gas-phase chemical reactions. It is difficult to construct a coupled aerosol-meteorology 4D-Var system directly based on the CUACE model. On the other hand, BC has an important impact on the climate and can be used to study the two-way feedback interactions between aerosol and meteorology (Chung and Seinfeld, 2002; Menon et al., 2002; Bond et al., 2013). Therefore, we utilized BC as a starting point to construct the strongly coupled aerosol-meteorology 4D-Var system (CMA-GFS-AERO 4D-Var).

Creating CMA-GFS-AERO 4D-Var required three important components: (1) CMA-GFS-AERO forward model, (2) CMA-GFS-AERO TLM and ADM, and (3) 4D-Var framework. This section provides a detailed description of the construction of the CMA-GFS-AERO 4D-Var from these three aspects.

## 3.1 CMA-GFS-AERO forward model

In this work, for the sake of interest in BC and the consideration of computational efficiency, we developed the CMA-GFS-AERO forward model by integrating the aerosol module AERO-BC into CMA-GFS v4.0. The AERO-BC module was created by extracting BC-related codes from the CUACE model, with its functionality aligning with the BC aerosol processes in the CAM module of CUACE. In other words, the physical processes for BC in AERO-BC are identical to those in the CAM module, with no changes made. The main differences lie in the engineering aspect: (1) while the CAM module was originally written in Fortran 77, the AERO-BC code has been rewritten in Fortran 90; (2) since CAM in CUACE deals with six types of aerosols, the code structure is somewhat complex and redundant, whereas AERO-BC focuses solely on BC, resulting in a simpler and more streamlined structure. These updates improve code readability and enhance computational efficiency, without affecting the underlying physical processes.

In AERO-BC, BC is represented by 6 bins with particle diameters of 0.01-0.04, 0.04-0.16, 0.16-0.64, 0.64-2.56, 2.56-10.24, and 10.24-40.96 μm, where the radius range is calculated by the geometric progression method to satisfy $i = 1 + \ln\left[(r_i/r_1)^3\right]/\ln[V_{RAT}]$, and $V_{RAT}$ is the average volume ratio between adjacent bins (Jacobson et al., 1994). Thus, six new prognostic variables for the mass mixing ratio of BC, denoted as $\psi_{bc}^n$ (unit: kg/kg), where $n = 1, ..., 6$, are added in the dynamical framework of CMA-GFS.

The main processes in AERO-BC include: (1) calculating the emission flux of BC through the surface flux calculation module, (2) calculating the vertical diffusion trend of BC by solving the vertical diffusion equation, and (3) simulating key BC aerosol processes in the atmosphere, including hygroscopic growth, coagulation, nucleation, condensation, dry deposition/sedimentation, and below-cloud scavenging. For more details, please refer to the relevant literature on the CAM module (Gong et al., 2003; Gong and Zhang et al., 2008; Wang et al., 2010; Zhou et al., 2012). The AERO-BC module is designed as one-dimensional (1-D) column module, which operates at individual vertical columns corresponding to fixed horizontal locations (i.e., fixed latitude and longitude). In the integration of AERO-BC with CMA-GFS, the interface programs transfer meteorological parameters (e.g., temperature, wind, and humidity) from CMA-GFS to AERO-BC, extend the spatial dimension from 1-D to 3-D, and read emissions for AERO-BC. The transport processes for $\psi_{bc}^n$ are the same as those for the variables associated with the different water species in CMA-GFS, using the hybrid PRM and QMSL schemes (Su et al., 2013).

Besides, according to the vertical distribution of BC in the MERRA-2 (Modern-Era Retrospective analysis for Research and Applications, Version 2) reanalysis data (https://daac.gsfc.nasa.gov), we observed that the BC mass mixing ratio decreases rapidly after entering the stratosphere, reaching values of about $10^{-12}$ kg/kg. This is 2-3 orders of magnitude smaller compared to the surface. To improve computational efficiency and balance memory usage with the effectiveness of BC forecasting, we set the height of $\psi_{bc}^n$ in the CMA-GFS-AERO model to 65 levels (approximately 30 hPa), which

corresponds to the middle layer of the stratosphere. Regarding the absence of BC above model level 65, we handled vertical transport by assuming that any BC concentrations above this level are negligible. This approximation does not significantly affect the model's performance, as the BC mass mixing ratio is very small in the upper layers. Correspondingly, in the adjoint code, BC concentrations above model level 65 are also treated as negligible, and this does not significantly affect the adjoint calculations.

## 3.2 CMA-GFS-AERO TLM and ADM

In developing the TLM and ADM of the CMA-GFS-AERO model, we firstly constructed the tangent linear and adjoint codes of the AERO-BC module, subsequently coupled them with the TLM and ADM of CMA-GFS model (Liu et al., 2017, 2023; Zhang et al., 2019), respectively. Since adjoint codes generated by automatic differentiation tools often suffer from issues such as poor readability and maintainability, low efficiency and even errors due to the complexity of numerical models (Zou et al., 1997), the tangent linear and adjoint codes in this study were written line-by-line manually, without using any automatic differentiation tool.

The AERO-BC can be symbolically written as

$$\mathbf{y}_{AERO} = \boldsymbol{F}(\mathbf{c}), \tag{4}$$

where $\boldsymbol{F}$ denotes the AERO-BC model operator, $\mathbf{c}$ and $\mathbf{y}_{AERO}$ are vectors representing the input and output variables of the AERO-BC, respectively.

The TL of the AERO-BC can be obtained by linearizing $\boldsymbol{F}$, expressed as

$$\delta\mathbf{y}_{AERO} = \mathbf{F}\delta\mathbf{c} = \frac{\partial \boldsymbol{F}}{\partial \mathbf{c}}\delta\mathbf{c}, \tag{5}$$

where $\mathbf{F}$ is the TL model operator, $\delta\mathbf{c}$ and $\delta\mathbf{y}_{AERO}$ represent perturbations of input and output variables of the AERO-BC, respectively.

The adjoint of the AERO-BC is essentially the transpose of the AERO-BC TL, expressed as

$$\delta\mathbf{c}^* = \mathbf{F}^T\delta\mathbf{y}_{AERO}^*, \tag{6}$$

where $\mathbf{F}^T$ is the adjoint operator of $\mathbf{F}$, $\delta\mathbf{y}_{AERO}^*$ and $\delta\mathbf{c}^*$ represent input and output variables of the adjoint of AERO-BC, respectively.

In constructing the TL and the adjoint of AERO-BC, no simplifications were made to the AERO-BC processes. Specifically, no regularization was applied to the nonlinear equations, nor were any complex processes, which were difficult to linearize, omitted. As a result, the TL and the adjoint of AERO-BC fully include all processes related to emission flux, vertical diffusion, and aerosol physical processes as described in Section 3.1.

The TL and the adjoint of AERO-BC are 1-D column modules, meaning that they operate independently at each fixed horizontal grid point (i.e., fixed latitude and longitude), with vertical variation only. To extend them to 3-D, the tangent linear

and the adjoint of the interface programs were also constructed. Furthermore, the tangent linear and the adjoint of BC

transport processes follow the same framework as those for the variables associated with the different water species in the CMA-GFS TLM and ADM, utilizing the tangent linear and the adjoint of QMSL. In this way, the 3-D parameters could be transferred from CMA-GFS to AERO-BC. Thus, we obtained the CMA-GFS-AERO TLM and ADM.

### 3.3 CMA-GFS-AERO 4D-Var

On the basis of the CMA-GFS-AERO forward model and its TLM and ADM, we further constructed the CMA-GFS-AERO 4D-Var by adding BC as a control variable into the incremental analysis scheme introduced in Section 2.2. We also provided a detailed introduction to the BC observation and errors, the BC observation operator, and the background error covariance for BC.

#### 3.3.1 BC mass concentration as control variable

The establishment of a strongly coupled aerosol-meteorology 4D-Var system based on the CMA-GFS 4D-Var requires the addition of aerosol analysis. Although the six variables for the mass mixing ratio of BC ($\psi_{bc}^n$) have been used in the CMA-GFS-AERO forward model, they can constitute a heavy burden for the analysis if they are all included in the control vector. The reasons for this, as mentioned by Benedetti et al. (2009), mainly include: (1) background error statistics would have to be generated for all variables separately, (2) the control vector would be significantly larger in size, which would

consequently increase the cost of the iterative minimization, and most importantly, (3) the BC analysis would be under constrained since the surface observations of BC are mass concentrations (unit: $\mu g/m^3$), which do not distinguish between size bins, resulting in one observation of BC mass concentration being used to constrain six BC variables. To address these issues, the BC mass concentration is selected as the control variable, denoted as $C_{bc}$ (unit: $\mu g/m^3$), and is added to the control vector ($\mathbf{x}_u = (\psi, \chi_u, \pi_u, q)^T$, $\psi$ is the stream function, $\chi_u$ is the unbalanced velocity potential, $\pi_u$ is the

unbalanced Exner pressure, and $q$ is the specific humidity) of CMA-GFS 4D-Var. Thus, the control vector for the CMA-GFS-AERO 4D-Var is $\mathbf{x}_u = (\psi, \chi_u, \pi_u, q, C_{bc})^T$, assuming that these five variables are independent of each other. The conversion relationship between $C_{bc}$ and $\psi_{bc}^n$ is

$$C_{bc} = \sum_{n=1}^{6} \psi_{bc}^n * \rho * 10^9, \tag{7}$$

where $\rho$ is the atmospheric density. In order to obtain the BC initial field that can be used in the CMA-GFS-AERO model

from the analysis field, it is also necessary to convert $C_{bc}$ to $\psi_{bc}^n$. Firstly, the distribution weights ($\omega^n$) of each size bin of $\psi_{bc}^n$ in the background field are calculated based on the entire three-dimensional domain, following the equation $\omega^n = \frac{\Sigma_1^N \psi_{bc}{}^n}{\Sigma_{n=1}^6 (\Sigma_1^N \psi_{bc}{}^n)}$, where $N$ represents the number of three-dimensional grid points. Secondly, the analysis increment of $\psi_{bc}^n$ ($\delta\psi_{bc}^n$) is calculated based on the analysis increment of $C_{bc}$ ($\delta C_{bc}$), following the equation

$$\delta\psi_{bc}^n = \omega^n * \frac{\delta C_{bc}}{\rho*10^9},\tag{8}$$

Finally, $\delta\psi_{bc}^n$ is interpolated and superimposed on $\psi_{bc}^n$ in the background field to obtain the initial field of BC.

Similarly, in the minimization process of the inner loop of CMA-GFS-AERO 4D-Var, the conversion between the tangent linear variable of BC ($\delta\psi_{bc}^n$) and the analysis increment of $C_{bc}$ ($\delta C_{bc}$) is also calculated according to the derivative of Eq. (7) ($\delta C_{bc} = \sum_{n=1}^{6} \delta\psi_{bc}^n * \rho * 10^9$) and Eq. (8).

### 3.3.2 BC observation and errors

The BC observations used in the CMA-GFS-AERO 4D-Var system are the BC surface concentrations obtained from the China Atmospheric Monitoring Network (CAWNET), which was established by the CMA and has been monitoring the BC surface mass concentration in China since 2006 (Xu et al., 2020). The BC observation data were collected from 32 stations (Guo et al., 2020), including 11 urban, 17 rural and 4 remote stations. The distribution of these stations is shown in Fig. S2. The monitoring of BC in CAWNET was conducted using an Aethalometer, AE31, which is one of the models produced by Magee Scientific (USA, https://www.aerosolmageesci.com). The AE31 determines mass concentration of BC particles collected from air samples, flowing through a quartz filter. The instrument measures the transmission through the filter over a wide spectrum of wavelengths from 370 nm to 950 nm. Light at the selected wavelength is transmitted through control and sample filters, and the attenuation change in the filter is then translated into the BC mass concentration. In this study, we used the BC concentration measured at the recommended wavelength of 880 nm. The AE31 measures BC concentrations every 5 minutes. We performed quality control on the original data and obtained the hourly average values, which were used in the BC assimilation experiments. The quality control procedures are as follows:

(1) Eliminating abnormal values. During the calculation of hourly averages from the 5-minute sampled data, any BC concentration values that differ significantly from the hourly average (i.e., those where the absolute difference exceeds three times the standard deviation) are considered abnormal and discarded. Additionally, any bad data flagged by the instrument's monitoring system are also removed.

(2) Filling in missing values. If more than one-third of the data for a given hour is missing, or if there are more than three consecutive missing values, the entire hour's data is discarded. For other cases, linear interpolation is applied to fill in the missing values.

The observation error covariance matrix **R** in Eq. (1) contains both measurement and representativeness errors. Following the formula described by Chen et al. (2019), which is an improvement on the method proposed by Pagowski et al. (2010) and Schwartz et al. (2012), we calculated the measurement error $\varepsilon_0$. The formula is expressed as

$$\varepsilon_0 = 1.0 + 0.0075 \times O_{bc},\tag{9}$$

where $O_{bc}$ denotes the observed BC concentrations (unit: μg/m³).

Representativeness errors reflect the inaccuracies in the observation operator and in the interpolation from the model grid to the observation location. We used the representativeness error ($\varepsilon_r$) expression defined by Elbern et al. (2007) as follows

$$\varepsilon_r = \gamma \varepsilon_0 \sqrt{\frac{\Delta x}{L}}, \tag{10}$$

where $\gamma$ is an adjustable parameter scaling $\varepsilon_0$ ($\gamma = 0.5$ was used here), $\Delta x$ is the grid spacing (100 km in this work), and $L$ is the radius of influence of a BC observation. According to Elbern et al. (2007), $L$ was set to 2 km, 10 km, and 20 km for urban, rural, and remote stations, respectively. The total BC observation error ($\varepsilon_{bc}$) was defined as

$$\varepsilon_{bc} = \sqrt{\varepsilon_0{}^2 + \varepsilon_r{}^2}, \tag{11}$$

which constituted the diagonal elements in the **R** matrix.

### 3.3.3 BC observation operator

The observation operator in the CMA-GFS-AERO 4D-Var system performs two basic tasks: (1) transforming model state variables into observed physical quantities, and (2) interpolating the background field (or analysis field) to the location of the observation. The transformation of the physical quantities is related to the type of observations, and the spatial interpolation operator consists of both horizontal and vertical interpolation. Since the CMA-GFS-AERO 4D-Var system adopts the Charney-Philips staggered grid in the vertical direction and the Arakawa-C grid in the horizontal direction, the observation operator must account for the staggered locations of different physical variables. To minimize errors introduced by variable transformations and spatial interpolation, appropriate handling of horizontal staggering and vertical layer transitions is required. The steps to construct the BC observation operator are as follows:

(1) Based on Eq. (7), the BC mass mixing ratios ($\psi_{bc}^n$) of six size bins are summed and converted into the mass concentrations ($C_{bc}$), which are further interpolated to the observation locations by the horizontal bilinear interpolation to obtain the equivalent BC concentrations that are consistent with the units of the observations.

(2) According to the heights of BC surface observations, the corresponding vertical interpolation schemes are selected to obtain the equivalent BC observations. If the height of BC surface observation is greater than the height of the first model layer, the cubic spline interpolation is used to process the BC concentration interpolation. If the observation height is less than the height of the first model layer and the difference between the two heights is less than 300 meters, the BC concentration at the first model layer is regarded as the equivalent BC observation. However, if the difference between the two heights is greater than or equal to 300 meters, the data from that site is discarded.

### 3.3.4 Background error covariance for BC

The variable fields involved in variational assimilation are all three-dimensional, and it is challenging to directly deal with the correlations of these three-dimensional fields due to their high dimensionality. Therefore, in the CMA-GFS 4D-Var assimilation system, a simplification is made by assuming that the correlation coefficient can be expressed as the product of

the vertical correlation coefficient and the horizontal correlation coefficient (Zhang et al., 2019). The horizontal correlation is calculated using the spectral filtering method, while the vertical correlation is calculated through EOF decomposition (Zhang et al., 2019).

In the CMA-GFS-AERO 4D-Var system, the background error covariance for the control variable BC adopts a modeled structure. The background error variance varies with height as shown in Fig. 1a. The vertical correlation model of the background error is derived through a combination of theoretical considerations (Bergman, 1979) and experimental tuning, with particular reference to the methodology used for humidity in the CMA-GFS 4D-Var system. It is expressed as

$$R(z_i, z_j) = \frac{1}{1 + k_z(z_i - z_j)^2} \, , \tag{12}$$

where $z_i$ and $z_j$ are the model terrain heights of level $i$ and $j$, respectively. $k_z = \frac{g^2}{(R_d T_0)^2} k_p$, $g$ denotes the gravitational acceleration, $R_d$ represents the gas constant for dry atmospheric air, $T_0$ is the standard temperature (273.15 K), and $k_p$ is the constant coefficient (Bergman, 1979). Following the value of $k_p$ used for the control variable of humidity in the CMA-GFS 4D-Var system, we set $k_p$ to 10 for the control variable BC. Figure 1b depicts the distribution of the vertical correlation coefficients of the background error of the 1st, 10th, and 20th layers with other layers.

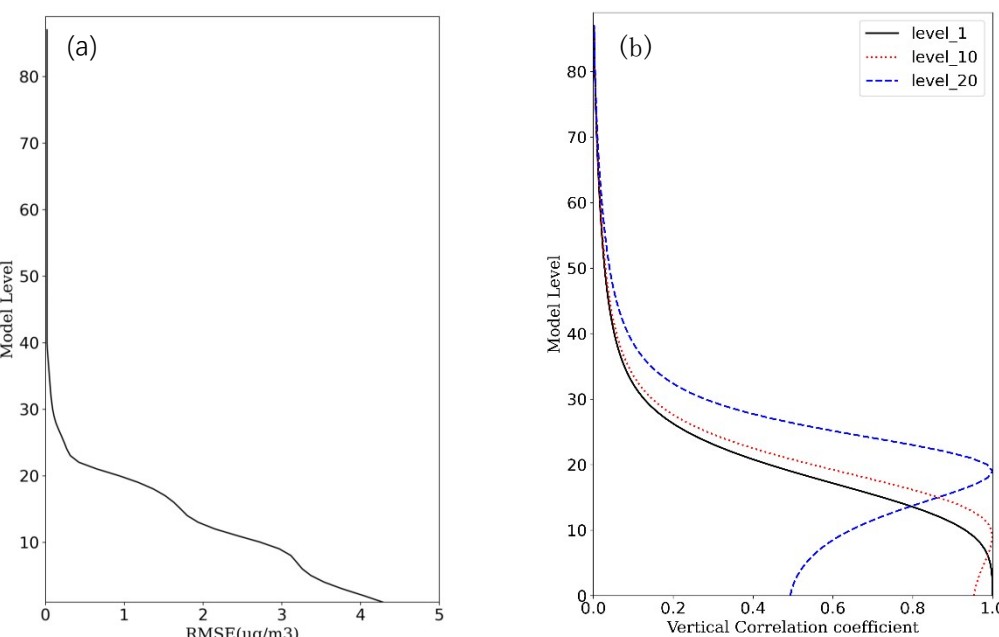

**Figure 1: (a) Vertical profile of background error variance for BC, (b) vertical correlation coefficients of background error between the 1st, 10th, and 20th layers with other layers for BC.**

The horizontal correlation of the background error for the control variable BC is calculated by the second-order auto-regressive (SOAR) correlation function, which is commonly used in operational data assimilation systems (Ballard et al., 2016), expressed as

$$r_{i,j} = \left(1 + \frac{d_{i,j}}{L}\right) \exp\left(-\frac{d_{i,j}}{L}\right), \tag{13}$$

where $d_{i,j}$ is the arc length of the great circle between two points $i$ and $j$, $L$ is the characteristic horizontal length scale

The length scale for the control variable BC varies with height in the model, following the way the length scale of the humidity variable varies with height in the CMA-GFS 4D-Var system, which is shown in Table 1.

**Table 1: Characteristic horizontal length scales of the background error.**

| Height (km) | length scale (km) |
| --- | --- |
| 0.50 | 165 |
| 1.43 | 172 |
| 5.56 | 175 |
| 10.5 | 209 |
| 16.3 | 234 |
| 23.9 | 234 |

### 3.3.5 Flow-dependent background error covariance in CMA-GFS-AERO 4D-Var

In the strongly coupled aerosol-meteorology assimilation system, interactions between the atmospheric variables and BC allow BC observations to influence the analysis increments of atmospheric variables and vice versa. The incremental 4D-Var algorithm implicitly evolves the background error covariances (**B**) throughout the assimilation window according to the TL

model dynamics. This process modifies prior background error variance estimates and induces non-zero correlations between model variables (Smith et al., 2015). By utilizing the fully coupled TLM and ADM in the inner loops of the strongly coupled assimilation system, cross-covariance information between BC and atmospheric variables is generated. This enables observations of one variable to produce analysis increments in the other, leading to more consistent analyses.

Specifically, if the BC observation is assumed to take place at the beginning of the assimilation window only, and under the

assumption of a single, non-cycling assimilation window, the 4D-Var assimilation behaves similarly to the 3D-Var assimilation. In this case, since the BC variable is assumed to be uncorrelated with the atmospheric variables in the static **B**, and there is no direct relationship between the BC observation operator and the atmospheric variables, the BC observation does not lead to the generation of the analysis increments of atmospheric variables. Therefore, the merits of a coupled data assimilation system cannot be fully manifested by assimilating a BC observation at the beginning of the window only. If the

BC observation is assumed to take place at the middle and the end of the assimilation window, **B** evolves within the assimilation time window through the TLM $\mathbf{M}_{0 \to i}$, obtaining the implicit background error covariance matrix $\mathbf{M}_{0 \to i}\mathbf{B}\mathbf{M}_{0 \to i}^{T}$ at the observation time. $\mathbf{M}_{0 \to i}\mathbf{B}\mathbf{M}_{0 \to i}^{T}$ includes the cross-covariances information of BC and atmospheric variables, and can

realize the feedback of the BC observation to the atmospheric variables through the CMA-GFS-AERO ADM $\mathbf{M}_{0 \to i}^{T}$, further producing analysis increments of atmospheric variables. In other words, the distribution of the analysis increment at the observation time is determined by the cross-time error matrix $\mathbf{M}_{0 \to i}\mathbf{B}$.

## 4  Model setup

In this work, the horizontal resolution of the CMA-GFS-AERO forward model in the outer loop was set to 0.25°, with an integration step of 300 s, and the horizontal resolution of the CMA-GFS-AERO TLM and ADM in the inner loop was 1.0°, with an integration step of 900s. The model has 87 vertical layers, with the top being approximately 0.1 hPa (Fig. S1). Referring to the running scheme of the CMA-GFS 4D-Var system described in Section 2.2, the CMA-GFS-AERO 4D-Var system also adopts the same 6-h cycling schedule and assimilation windows. The forecast of the CMA-GFS-AERO model started at 0300 UTC on October 1, 2016, and was restarted every 6 h. The meteorological initial fields for each 6-h cycle were obtained from the operational CMA-GFS analysis. The BC field was initialized with null concentrations at 0300 UTC on October 1, 2016. From the second forecast cycle onward, the initial conditions of BC were derived from the BC field at the end of the previous 6-h forecast, allowing the BC field to be cycled. The first 9 days were used as the spin-up time to establish a realistic BC distribution. The maximum minimization iteration number in the inner loop was set to 50, while the outer loop was performed only once. This setting is consistent with the operational configuration of the CMA-GFS 4D-Var system and has been found sufficient for achieving convergence in our experiments. The atmospheric observations used in this work are shown in Table S1.

Anthropogenic emission sources used in this study were from the Multi-resolution Emission Inventory for China (MEIC) (Li et al., 2017; Zheng et al., 2018), the Copernicus Atmosphere Monitoring Service global and regional emissions (CAMS) (Granier et al., 2019), and the global datasets of the Task Force Hemispheric Transport of Air Pollution (HTAP) (Janssens-Maenhout et al., 2015). These inventories include various gases ($NO_x$, CO, $SO_2$, $NH_3$, $CH_4$ and NMVOC) and particulates (OC, BC, $PM_{2.5}$ and $PM_{10}$), where $PM_{10}$ refers to the inhalable particulate matter with an aerodynamic diameter of 10 micrometers or less. These data were processed into grid-point emission data applicable to the CUACE model through the EMIPS emission source processing system (Chen et al., 2023). To improve computational efficiency, they were further simplified into emission source data containing only BC as input to the CMA-GFS-AERO model.

At present, we have run the CMA-GFS-AERO 4D-Var system for three months from October 1, 2016. This section mainly shows the experiment results of random time in these three months to present the rationality and stability of the system. The detailed analysis of the cycling assimilation experiments will be further elaborated in part 2 of this paper.

## 5  Results

### 5.1  Validation of CMA-GFS-AERO TLM and ADM

Validation of the tangent linear and adjoint models is an important part of introducing a new modeling component, such as the AERO-BC module. Considering that CMA-GFS TLM and ADM have been validated and documented in Liu et al. (2017, 2023) and Zhang et al. (2019), here we mainly present the validation of tangent linear and adjoint of the newly developed AERO-BC module.

The correctness of the AERO-BC TL can be verified by checking whether the following equation is satisfied (Mahfouf and Rabier, 2000; Liu et al., 2017; Tian and Zou, 2020):

$$\Phi(\alpha) = \frac{\|\boldsymbol{F}(\boldsymbol{c}+\alpha\cdot\delta\boldsymbol{c})-\boldsymbol{F}(\boldsymbol{c})\|}{\|\boldsymbol{F}(\alpha\cdot\delta\boldsymbol{c})\|} = 1 + O(\alpha), \tag{14}$$

where $\|.\|$ denotes the norm of the vector, $\alpha$ is the scale factor of initial perturbations with the range from 1.0 to $10^{-14}$. As the scale factor $\alpha$ becomes smaller and smaller, the function $\Phi(\alpha)$ is expected to approach unity in an approximately linear manner.

We firstly verified all submodules in the AERO-BC TL, finding that the tangent linear approximation of each submodule was correct. Subsequently, we conducted a set of six experiments with the integration time from 1 to 6 h to verify the correctness of the full AERO-BC TL. The background field and analysis increment generated by the CMA-GFS-AERO 4D-Var system were used as the basic-state initial field and the perturbation initial field of the CMA-GFS-AERO TLM for 6-hour forecasting. The atmospheric and BC state variables $\boldsymbol{c}$ and their perturbations $\delta\boldsymbol{c}$ of these six time periods were used as inputs of the AERO-BC and its TL, and the tangent linear approximation of the output variable (the perturbation of the mass mixing ratio of BC, $\delta\psi_{bc}^{n}$) of the AERO-BC TL is tested using Eq. (14).

Figure 2 shows the results of the six correctness experiments. As expected, in each verification experiment, as the scale factor $\alpha$ becomes smaller and smaller for certain ranges of $\alpha$ values, the values of $\Phi(\alpha)$ gradually get closer and closer to unity. When $\alpha$ is too small (such as $10^{-12}$), the accuracies of the $\Phi(\alpha)$ values start to be affected by the machine round-off errors and drift away from unity. This indicates that the tangent linear approximation of the AERO-BC TL is correct.

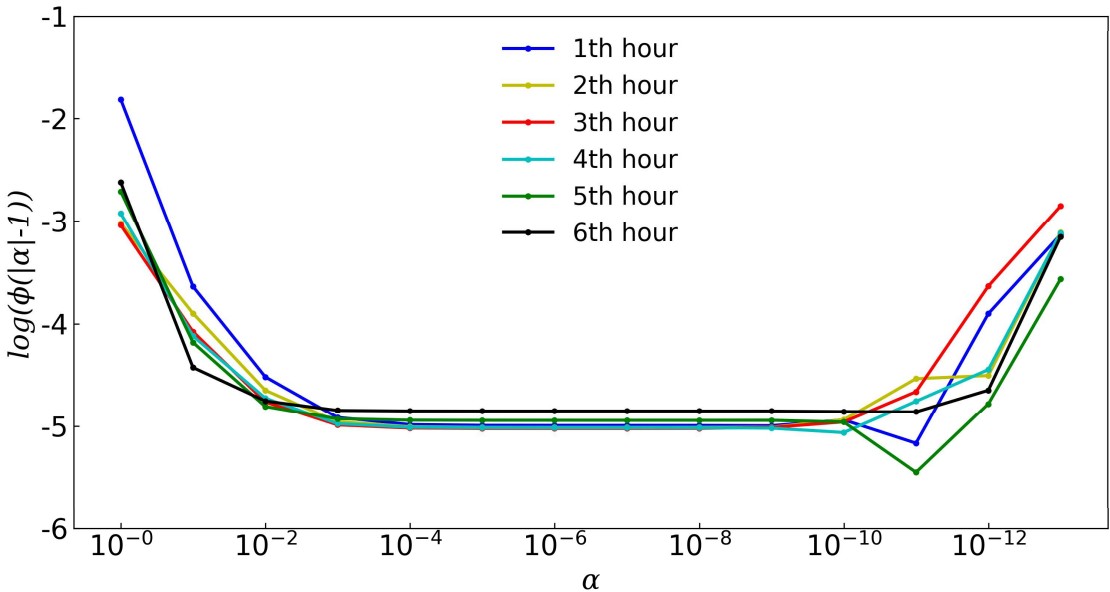

**Figure 2: Variations in the function $|\Phi(\alpha) - 1|$ for the correctness check of the AERO-BC TL for the 6-h forecast length, where $\alpha$ is the scale factor of initial perturbations.**

We further diagnosed the impact of linearized physical processes on the forecast effectiveness of CMA-GFS-AERO TLM.

Generally, the diagnostic method is to calculate the relative error ($r$) between the tangent linear perturbation forecast $\mathbf{M}(\delta\mathbf{x})$ and the nonlinear perturbation forecast $\Delta\boldsymbol{M}(\delta\mathbf{x})$ (Mahfouf, 1999; Liu et al., 2019; Zhang et al., 2019), which can be expressed as

$$r = \frac{|\mathbf{M}(\delta\mathbf{x}) - \Delta\boldsymbol{M}(\delta\mathbf{x})|}{\Delta\boldsymbol{M}(\delta\mathbf{x})}. \tag{15}$$

The nonlinear perturbation forecast $\Delta\boldsymbol{M}(\delta\mathbf{x})$ is the difference between the NLM forecasts from two different initial

conditions: the analysis field $\mathbf{x}_a$ and the background field $\mathbf{x}_b$, that is $\Delta\boldsymbol{M}(\delta\mathbf{x}) = \boldsymbol{M}(\mathbf{x}_a) - \boldsymbol{M}(\mathbf{x}_b)$. And the tangent linear perturbation forecast $\mathbf{M}(\delta\mathbf{x})$ is integrated using the analysis increment $\delta\mathbf{x}$ ($\delta\mathbf{x} = \mathbf{x}_a - \mathbf{x}_b$) as the initial perturbation field. $r$ needs to be calculated for each model variable at each grid.

The forecast period for this experiment was 6 h starting from 0300 UTC on October 25, 2016 (randomly selected time). For the nonlinear perturbation test, which includes the full physical processes, the two initial conditions were the analysis field

$\mathbf{x}_a$ and the background field $\mathbf{x}_b$ generated by the CMA-GFS-AERO 4D-Var system at 0300 UTC on October 25, 2016. For the tangent linear perturbation test, the initial condition was the analysis increment $\delta\mathbf{x}$ ($\delta\mathbf{x} = \mathbf{x}_a - \mathbf{x}_b$) at 0300 UTC on October 25, 2016. The model trajectory required for the tangent linear perturbation forecast was calculated by the CMA-GFS-AERO NLM including the full physical process with the background field $\mathbf{x}_b$ as the initial field. The nonlinear and tangent linear models were performed at the same resolution of 1.0°, and the analysis field $\mathbf{x}_a$ and the background field

$\mathbf{x}_b$ were interpolated from 0.25° to 1.0° based on the 3D interpolation method (Huo et al., 2022).

Figure 3 depicts the results of the nonlinear perturbation forecast and the tangent linear perturbation forecast. Figure 3a-b show the differences in vertically accumulated and latitudinally averaged BC mass concentration (unit: μg/m³) after 6-h integration of the CMA-GFS-AERO NLM with two initial conditions of $\mathbf{x}_a$ and $\mathbf{x}_b$, respectively, and Fig. 3c-d present the vertically accumulated and latitudinally averaged BC mass concentration perturbations after 6-h integration of CMA-GFS-AERO TLM with the initial condition of $\delta\mathbf{x}$ ($\delta\mathbf{x} = \mathbf{x}_a - \mathbf{x}_b$), respectively. It can be seen that after 6-h forecast, the distribution of the results of CMA-GFS-AERO NLM and TLM, both horizontally and vertically, are very similar with only minor differences. This indicates that CMA-GFS-AERO TLM shows good performance in tangent linear approximation for BC.

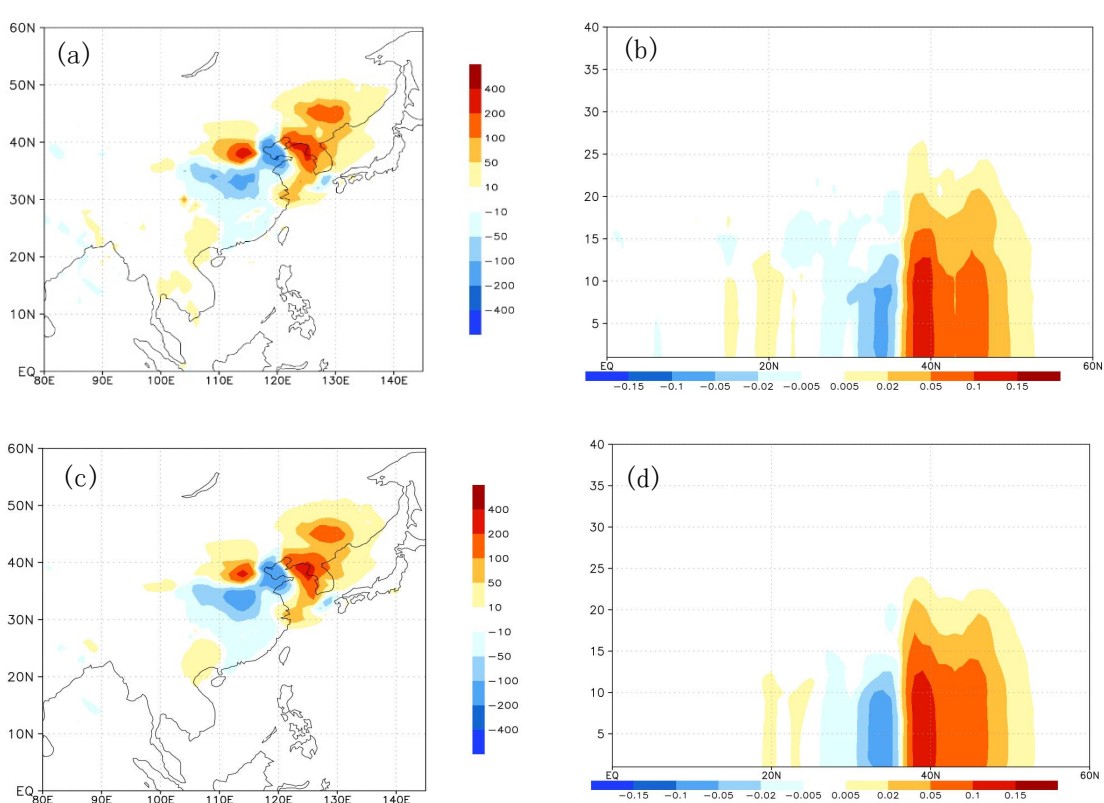

**Figure 3: Differences in (a) vertically accumulated and (b) latitudinally averaged BC mass concentration (unit: μg/m³) after 6-h integration of the CMA-GFS-AERO NLM with initial conditions $\mathbf{x}_a$ and $\mathbf{x}_b$, and perturbations in (c) vertically accumulated and (d) latitudinally averaged BC mass concentration after 6-h integration of CMA-GFS-AERO TLM with the initial condition $\delta\mathbf{x}$ ($\delta\mathbf{x} = \mathbf{x}_a - \mathbf{x}_b$).**

The vertical distribution of the globally averaged relative error between the perturbation forecasts of CMA-GFS-AERO TLM and NLM, which was calculated according to Eq. (15), is shown in Fig. 4. It can be seen that below the 20th model layer, the tangent linear approximation for BC is better than that for wind field, potential temperature, and specific humidity. Although the tangent linear approximation for BC is slightly worse above the 20th model layer, it is still far better than that for specific humidity. It's worth noting that the BC concentration above the 20th model level is quite low (Fig. 3b), so the impact of the tangent linear approximation is minimal. This phenomenon indicates that, in comparison to variables such as

potential temperature and specific humidity in the CMA-GFS-AERO model, the tangent linear approximation for BC is quite effective, making it well-suited for constructing a 4D-Var system.

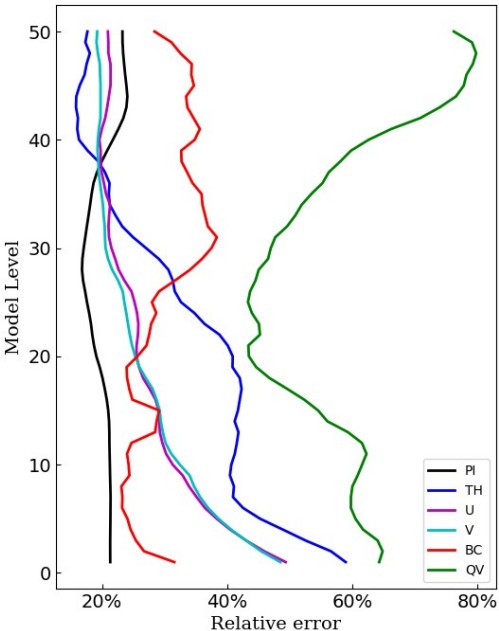

**Figure 4: The vertical distribution of the globally averaged relative error between the perturbation forecasts of the CMA-GFS-AERO TLM and NLM at the resolution of 1.0°. (black line: non-dimensional pressure, blue line: potential temperature, red line: BC, magenta line: u-wind, cyan line: v-wind; green line: specific humidity).**

The correctness of the AERO-BC adjoint can be verified by checking whether the following equation is satisfied (Mahfouf and Rabier, 2000; Liu et al., 2017; Tian and Zou, 2020)

$$\langle \mathbf{F}(\delta\mathbf{c}), \mathbf{F}(\delta\mathbf{c}) \rangle = \langle \delta\mathbf{c}, \mathbf{F}^T(\mathbf{F}(\delta\mathbf{c})) \rangle, \tag{16}$$

where $\langle , \rangle$ denotes the inner product. Using $\delta\mathbf{c}$ as the input of the AERO-BC TL, the output of the AERO-BC TL $\mathbf{F}(\delta\mathbf{c})$ can be obtained and the left-hand side (LHS) of Eq. (16) can be calculated. Then, taking $\mathbf{F}(\delta\mathbf{c})$ as the input of the AERO-BC adjoint, we can get its output $\mathbf{F}^T(\mathbf{F}(\delta\mathbf{c}))$ and calculate the right-hand side (RHS). If the AERO-BC adjoint is developed correctly, the LHS and RHS of Eq. (16) is expected to agree with the machine accuracy of the data type declared in the program, which is double precision in the AERO-BC.

Following Eq. (16), we conducted five experiments with the integration time equal to 1, 6, 12, 24, and 36 steps with the time step of 900 s. Considering the mass mixing ratio of BC ($\psi_{bc}^n$) as an example, for each experiment, the atmospheric variables and $\psi_{bc}^n$ perturbations in the analysis increment generated by the CMA-GFS-AERO 4D-Var system was used as the input of the AERO-BC TL. We run the tangent linear codes once to obtain the value of the tangent linear output, and calculated the

490 LHS of Eq. (16). Then, taken the tangent linear output as input, the AERO-BC adjoint codes was run once to obtain the

sensitivity value, which further was used to calculated the RHS of Eq. (16) with the $\psi_{bc}^n$ perturbation. The validation results are presented in Table 2. The resulting LHS and RHS from the five tests agree with the precision of machine accuracies, indicating the correctness of the AERO-BC adjoint model.

Table 2: Correctness check results of the newly developed AERO-BC adjoint model when it is integrated for 1, 6, 12, 24, and 36 steps.

| Step | LHS | RHS | (LHS-RHS)/LHS |
|------|-----|-----|---------------|
| 1 | 6. 048801009887637E-015 | 6. 048801009887634E-015 | 5.2166431260112900E-16 |
| 6 | 5. 661147803064362E-015 | 5. 661147803064381E-015 | 3.3443150371477720E-15 |
| 12 | 5. 608184349558140E-015 | 5. 608184349558160E-015 | 3.6572234893387934E-15 |
| 24 | 5. 694921201673081E-015 | 5. 694921201673082E-015 | 1.3852007381406021E-16 |
| 36 | 5. 845344664075793E-015 | 5. 845344664075791E-015 | 2.6991082666833257E-16 |

LHS: left-hand side of Eq. (16); RHS: right-hand side of Eq. (16).

## 5.2 Single observation experiment

In order to evaluate the rationality of the CMA-GFS-AERO 4D-Var system, we performed the single observation experiment for BC. The experiment period was 6 h starting from 0300 UTC on November 24, 2016 (randomly selected time), and the forecast field of the CMA-GFS-AERO model at this time was selected as the background field. During the assimilation process, no atmospheric observations were added. We adopted the BC surface observation at Nanjiao station (116.47°E, 39.8°N), which is located in Beijing, at 0300 UTC on November 24, 2016. The altitude of Nanjiao station is 31.3 meters, and the observed BC concentration is 10.0 μg/m³. Figure S3 shows the location of the BC observation and the wind field at 925hPa, which moves from northwest to southeast. The BC observation was placed at 0300, 0600, and 0900 UTC, respectively, corresponding to the beginning, the middle, and the end of the assimilation time window.

Theoretically, the analysis increment at the beginning time for 4D-Var assimilation is $\delta\mathbf{x} = \mathbf{B}\sum_{i=0}^{n}\mathbf{M}_{0\to i}^T\mathbf{H}_i^T\left(\mathbf{H}_i\mathbf{M}_{0\to i}\mathbf{B}\mathbf{M}_{0\to i}^T\mathbf{H}_i^T + \mathbf{R}_i\right)^{-1}(-\mathbf{d}_i)$. If we only assimilate the observation at time $t_i$, the analysis increment at the observation time is $\mathbf{M}_{0\to i}\delta\mathbf{x} = \mathbf{M}_{0\to i}\mathbf{B}\mathbf{M}_{0\to i}^T\mathbf{H}_i^T\left(\mathbf{H}_i\mathbf{M}_{0\to i}\mathbf{B}\mathbf{M}_{0\to i}^T\mathbf{H}_i^T + \mathbf{R}_i\right)^{-1}(-\mathbf{d}_i)$. When assimilating the single observation, $\left(\mathbf{H}_i\mathbf{M}_{0\to i}\mathbf{B}\mathbf{M}_{0\to i}^T\mathbf{H}_i^T + \mathbf{R}_i\right)^{-1}(-\mathbf{d}_i)$ is a vector with only one factor. If the observation position and the analysis grid coincide, the spatial interpolation in the observation operator can be ignored. Thus, the analysis increment at the observation time can reflect the structure of the background error covariance $\mathbf{M}_{0\to i}\mathbf{B}\mathbf{M}_{0\to i}^T$ at the observation time. Figure 5 shows the analysis increments of BC at the first model layer at the observation times, with the BC observation placed at 0300,

0600, and 0900 UTC, respectively. When the BC observation is placed at 0300 UTC (the observation innovation ($\mathbf{d}_i = \mathbf{H}_i\mathbf{M}_{0 \to i}(\mathbf{x}_b) - \mathbf{y}_i$) is -1.2 μg/m³ at 0300 UTC), the 4D-Var assimilation behaves similarly to the 3D-Var assimilation, and the horizontal distribution of the BC analysis increment is determined by the static background error covariance model $\mathbf{B}$. Since the CMA-GFS-AERO 4D-Var system uses a homogeneous second-order autoregressive spatial correlation model, the BC analysis increment at 0300 UTC (Fig. 5a) is essentially isotropic, and only the background error covariance, which varies with latitude, causes the analysis increment to differ somewhat in the north-south direction. When the BC observation is placed at 0600 UTC (the observation innovation is -9.5 μg/m³ at 0600 UTC) and 0900 UTC (the observation innovation is -9.0 μg/m³ at 0900 UTC), the BC analysis increments show anisotropic characteristics (Fig. 5b-c), which is consistent with the movement of the wind at 925hPa (Fig. S3), indicating that the background error covariance varies with the weather situation. Meanwhile, it can also be seen that the values of the BC analysis increments at 0600 and 0900 UTC are much larger than those at 0300 UTC. This is because the BC observation innovation at 0600 and 0900 UTC are greater than those at 0300 UTC.

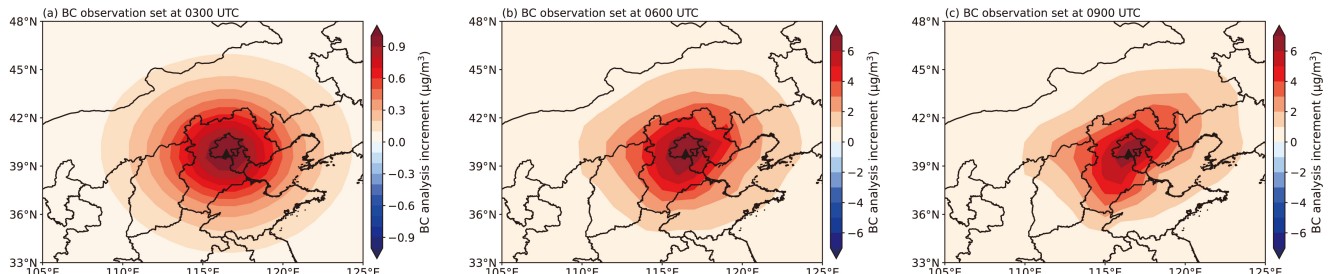

**Figure 5: The analysis increments of BC at the first model level at the observation times, with the BC observation placed at (a) the beginning of the assimilation time window, 0300 UTC; (b) the middle of the assimilation window, 0600 UTC; (c) the end of the assimilation time window, 0900 UTC. The black triangle represents the ideal observation location (116.47°E, 39.8°N).**

Figure 6 presents the evolved analysis increments of BC at the first model level at the end of the assimilation time window (0900 UTC) obtained by CMA-GFS-AERO TLM, with the BC observation placed at 0300 and 0600 UTC, respectively. For the case where the BC observation is placed at 0300 UTC, the initial analysis increment at 0300 UTC (Fig. 5a) exhibits an isotropic structure due to the static $\mathbf{B}$. In contrast, the propagated analysis increment at the end of the assimilation time window (0900 UTC, Fig. 6a) exhibits an anisotropic structure under the influence of the flow-dependent $\mathbf{M}_{0 \to i}\mathbf{B}\mathbf{M}_{0 \to i}^{\mathrm{T}}$. Similarly, when the BC observation is placed at 0600 UTC, both the initial analysis increment at 0600 UTC (Fig. 5b) and the propagated analysis increment at 0900 UTC (Fig. 6b) exhibit an anisotropic structure. In addition, the horizontal distribution structure of the BC analysis increments in Fig. 6a and Fig. 6b closely resembles that of the analysis increments at the observation time of 0900 UTC (Fig. 5c). This indicates the significant impact of flow-dependent dynamics on the evolution of the analysis increments. No matter what time the observation is placed at, the spatial propagation of the observation

information is effectively achieved through the model integration. In this idealized single observation experiment, the propagation of BC increments is primarily dominated by advection due to the limited observational constraint. When more comprehensive observations are assimilated, advection remains a key factor, but its dominance is less pronounced as other processes also influence the adjustment of BC distributions (see Section 5.3).

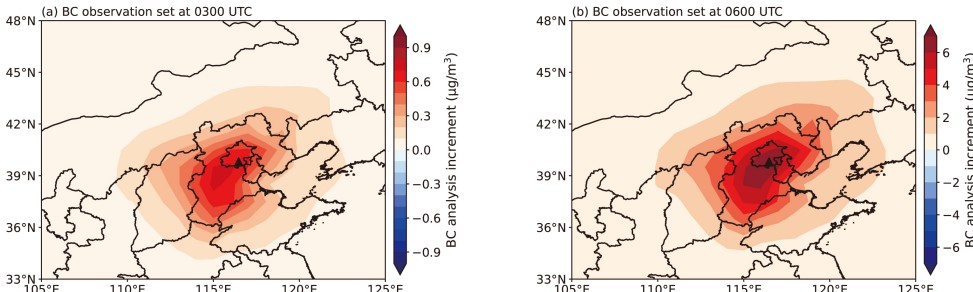

**Figure 6: The analysis increments of BC at the first model level at the end of the assimilation time window, 0900 UTC, with the BC observation placed at (a) the beginning of the assimilation window, 0300 UTC; (b) the middle of the assimilation window, 0600 UTC. The black triangle represents the ideal observation location (116.47°E, 39.8°N).**

Figure 7 depicts the analysis increments of temperature at the first model level at the beginning time of the assimilation time window (0300 UTC), with the BC observation placed at 0600 and 0900 UTC, respectively. In this specific case, the analysis increments of temperature are positive, with the value of about 0.02 K near the observation location, when the BC observation is placed at 0600 and 0900 UTC. The temperature analysis increment depends on several factors, including the BC observation innovation, the location of the observation, and the meteorological conditions during the assimilation time window. Here, the positive analysis increments of temperature may be due to the fact that the BC observation innovations at 0600 and 0900 UTC are negative (-9.5 μg/m³ at 0600 UTC and -9.0 μg/m³ at 0900 UTC), indicating that the background BC concentration is lower than the observed values. Assimilation of these observations increases the BC concentrations in the analysis, which, under the prevailing meteorological conditions, leads to positive temperature increments near the observation site. As explained in Section 3.3.5, the coupling between BC and atmospheric variables within the system allows this type of feedback to occur.

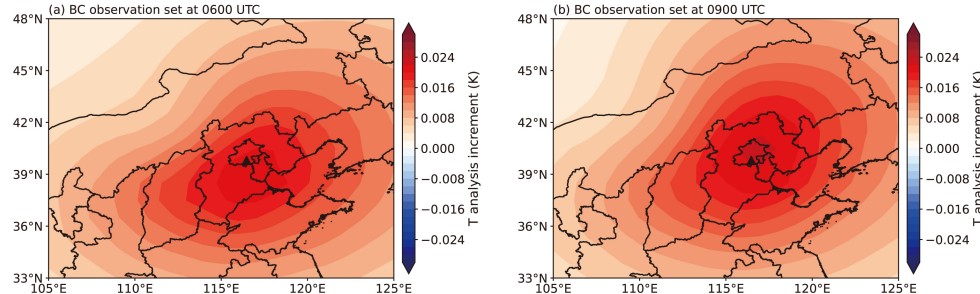

**Figure 7: The analysis increments of temperature at the first model layer at the beginning of the assimilation time window, 0300 UTC, with the BC observation placed at (a) the middle of the assimilation window, 0600 UTC; (b) the end of the assimilation time window, 0900 UTC. The black triangle represents the ideal observation location (116.47°E, 39.8°N).**

Figure 8 shows the analysis increments of pressure at the first model level, as well east-west component of horizontal wind, and relative humidity at the same level at the beginning of the assimilation time window (0300 UTC), with the BC observation placed at 0900 UTC. The assimilation of a single BC observation produces noticeable analysis increments in pressure, east-west component of horizontal wind, and relative humidity in North China, which shows that the CMA-GFS-AERO 4D-Var coupled assimilation system can reflect the impact of BC assimilation on atmospheric increments. In reality, unlike the single observation experiment, the BC observation is distributed within the assimilation time window, rather than just at a fixed moment, thus, the advantages of the CMA-GFS-AERO 4D-Var strong coupling assimilation system can be fully utilized to explore the feedback of BC assimilation on atmospheric variables.

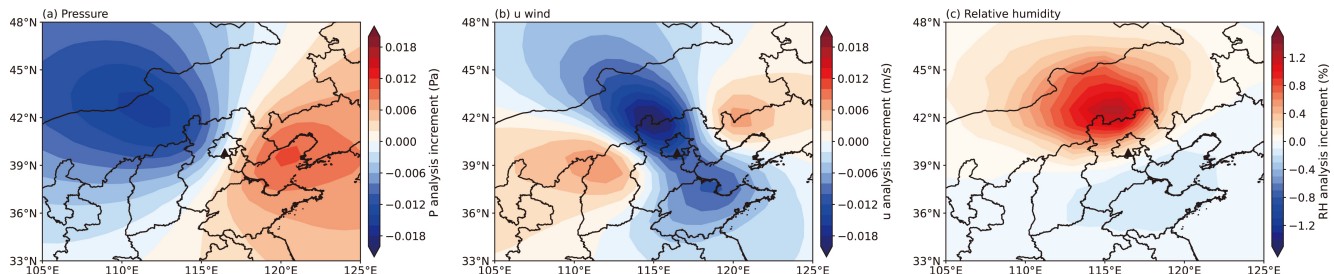

**Figure 8: The analysis increments of (a) pressure, (b) east-west component of horizontal wind, and (c) relative humidity at the first model layer at the beginning of the assimilation time window, 0300 UTC, with the BC observation placed at the end of the assimilation time window, 0900 UTC. The black triangle represents the ideal observation location (116.47°E, 39.8°N).**

### 5.3 Case study on BC and atmosphere assimilation

On the basis of the single observation experiment, we further conducted the full observation experiment for BC and atmospheric variables. The experiment period was also 6 h starting from 0300 UTC on November 24, 2016 (the same time as the experimental setup in Section 5.2), and the forecast field of the CMA-GFS-AERO model at this time was selected as the background field. We conducted a set of four experiments to investigate the impact of different BC assimilation strategies on both BC and atmospheric variables. These experiments are listed in Table 3. Different from the single observation experiment in Section 5.2, in which the observations are placed at a fixed time, we assimilated all available BC observations with an hourly frequency within the assimilation time window in the full observation experiment. For the DA_MET_then_BC experiment, the CMA-GFS-AERO 4D-Var system was executed twice sequentially within the same assimilation window. In the first step, only operational meteorological observations were assimilated, and the resulting

analysis was used as the background field for the second step, in which only BC surface observations were assimilated.

Except for the observational datasets, the model configurations and assimilation settings in both steps remained identical. This two-step procedure allows us to separate the effect of BC observations from the influence of meteorological observations and their associated background adjustment, thereby facilitating a clearer attribution of the BC assimilation impact. In contrast, DA_MET_BC_simult assimilated both operational meteorological observations and BC surface observations simultaneously within a single 4DVar run. This one-step assimilation strategy allows all observations to jointly

influence the analysis field, reflecting the integrated effect of both meteorological and BC observations. In the following analysis, we primarily compare the BC analysis increments obtained from DA_BC, DA_MET_then_BC, and DA_MET_BC_simult experiments, noting that the BC analysis increments from the DA_MET experiment are very small (figure omitted). Additionally, we compare the atmospheric analysis increments caused by BC assimilation in DA_BC, DA_MET_then_BC (DA_MET_then_BC-DA_MET), and DA_MET_BC_simult (DA_MET_BC_simult-DA_MET).

**Table 3: Design of four assimilation experiments.**

| Experiments | Description |
| --- | --- |
| DA_BC | Assimilating only BC surface observations while excluding operational meteorological observations |
| DA_MET | Assimilating only operational meteorological observations while excluding BC surface observations |
| DA_MET_then_BC | First assimilating operational meteorological observations, then assimilating BC surface observations |
| DA_MET_BC_simult | Assimilating both operational meteorological and BC surface observations simultaneously |

Figure 9 presents the analysis increments of BC at the first model layer from the DA_BC, DA_MET_then_BC, and DA_MET_BC_simult experiments. These analysis increments are valid at the beginning of the assimilation window, as is

standard in 4D-Var systems. When only BC surface observations are assimilated (DA_BC), the BC analysis increment is mainly concentrated in North China and Eastern China, with a maximum value of about 6.0 $\mu g/m^3$ (Fig. 9a). When operational meteorological observations are assimilated first, followed by BC surface observations (DA_MET_then_BC), or when both operational meteorological and BC surface observations are assimilated simultaneously (DA_MET_BC_simult), the distribution and the value of BC analysis increments are nearly identical to those of DA_BC, with only minor differences

(Fig. 9b-c). This indicates that the three BC assimilation strategies have similar assimilation effects on BC, further

demonstrating that the assimilation of meteorological observations has a relatively small impact on BC analysis increments.

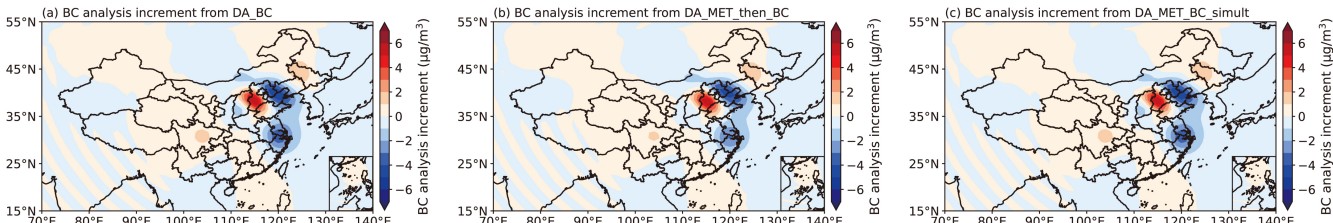

**Figure 9: The analysis increments of BC at the first model layer from (a) DA_BC, (b) DA_MET_then_BC, and (c)**
**DA_MET_BC_simult.**

We further explored the impact of different BC assimilation strategies on analysis increments of atmospheric variables. Figure 10 shows the analysis increments of temperature, pressure, east-west component of horizontal wind, and relative humidity at the first model layer, resulting from BC assimilation in DA_BC, DA_MET_then_BC, and DA_MET_BC_simult.
It is worth noting that in DA_BC, only BC observations are assimilated, so the analysis increments of atmospheric variables purely reflect the response to BC. In contrast, both DA_MET_then_BC and DA_MET_BC_simult assimilate BC and meteorological observations, and thus their analysis increments include the combined effects of both types of observations. To isolate the influence of BC assimilation alone on atmospheric variables, and under the assumption that the contribution of meteorological observations is comparable between DA_MET_then_BC/DA_MET_BC_simult and DA_MET, we calculated
the differences between the analysis increments of these experiments and those from DA_MET, which assimilates only meteorological observations. This subtraction effectively removes the contributions from meteorological observations, allowing the resulting increments to be attributed solely to the assimilation of BC observations. In this way, a more direct and fair comparison can be made with DA_BC. Panels 10a, 10d, 10g, and 10j display the analysis increments of these variables from BC assimilation in DA_BC. Panels 10b, 10e, 10h, and 10k show the increments due to BC assimilation in
DA_MET_then_BC, obtained by subtracting the atmospheric increments in DA_MET from DA_MET_then_BC (DA_MET_then_BC-DA_MET). Panels 10c, 10f, 10i, and 10l illustrate the increments caused by BC assimilation in DA_MET_BC_simult, obtained similarly by subtracting the increments in DA_MET from DA_MET_BC_simult (DA_MET_BC_simult - DA_MET).

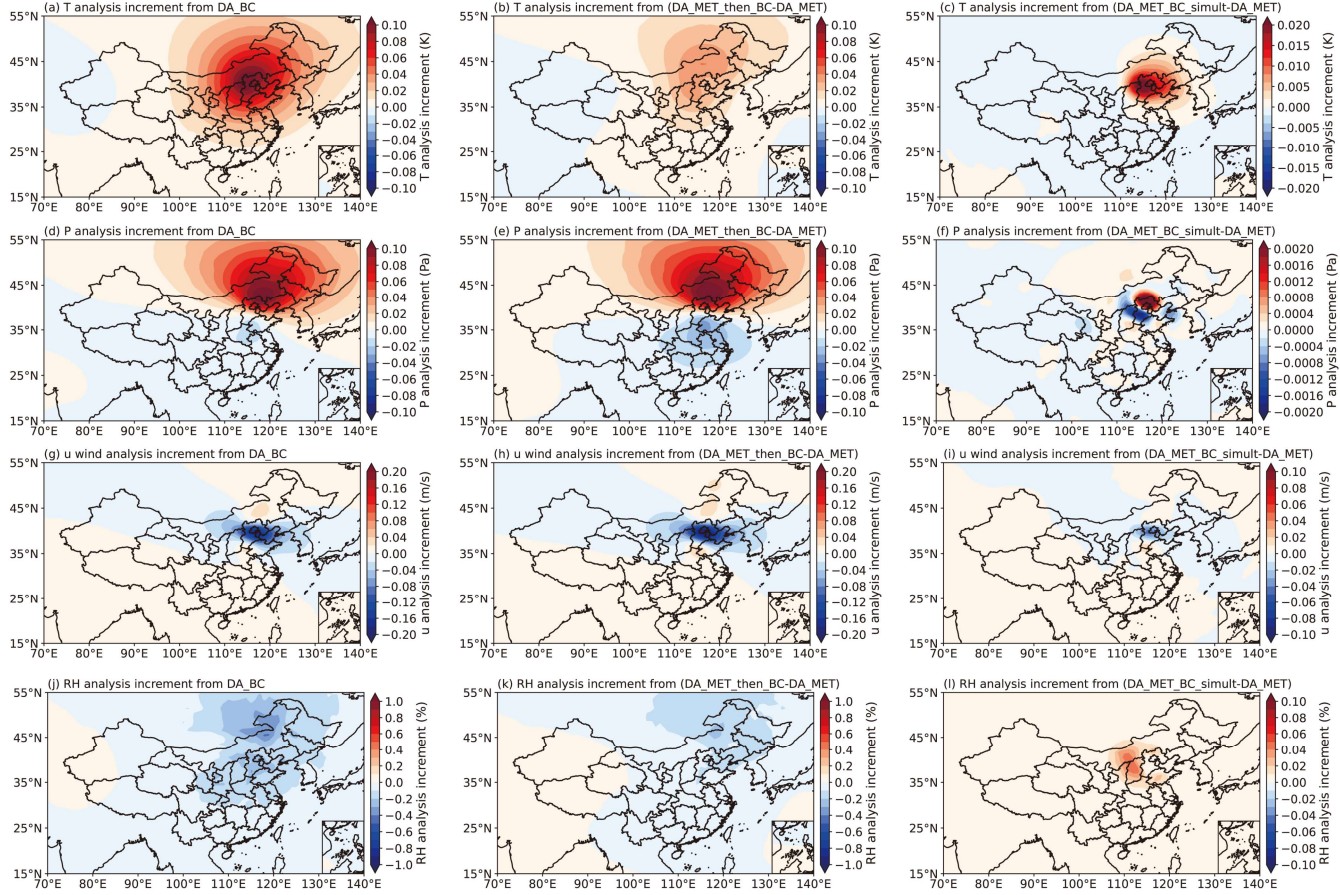

**Figure 10: The analysis increments of (a, b, c) temperature, (d, e, f) pressure, (g, h, i) east-west component of horizontal wind, and (j, k, l) relative humidity at the first model layer caused by BC assimilation. (a, d, g, j) are analysis increments from DA_BC, (b, e, h, k) are the differences in analysis increments between DA_MET_then_BC and DA_MET (DA_MET_then_BC minus DA_MET), and (c, f, i, l) are the differences in analysis increments between DA_MET_BC_simult and DA_MET (DA_MET_BC_simult minus DA_MET).**

When only BC surface observations are assimilated (DA_BC), analysis increments of temperature (Fig. 10a), pressure (Fig. 10d), east-west component of horizontal wind (Fig. 10g), and relative humidity (Fig. 10j) are present in North China and Eastern China. The value of the analysis increments for temperature, pressure, east-west component of horizontal wind, and relative humidity reach approximately 0.1 K (Fig. 10a), 0.1 Pa (Fig. 10d), -0.2 m/s (Fig. 10g), and 0.8% (Fig. 10j), respectively.

When operational meteorological observations are assimilated first, followed by BC surface observations (DA_MET_then_BC), the distributions and the values of the analysis increments of these four atmospheric variables due to BC assimilation (Fig. 10b, e, h, k) are basically consistent with those of DA_BC. This is because, although the DA_MET_then_BC experiment assimilates meteorological observations before BC surface observations, the background field of BC remains unchanged. While the assimilation of meteorological observations updates atmospheric variables, it does not directly alter the BC background field. Therefore, the observation-minus-background (OMB) values for BC observations in DA_MET_then_BC are very close to those in DA_BC, with only minor differences caused by the slight influence of

updated meteorological fields on the observation operator. As a result, the analysis increments of atmospheric variables due to BC assimilation are similar between the two experiments. Additionally, the values in each sub-image of the middle panel in Fig. 10 differ slightly from those on the left. These differences are attributed to the distinct basic-state values of the atmospheric variables used in the tangent linear and adjoint processes. Specifically, in DA_BC, the basic-state values of the atmospheric variables are derived from the atmospheric background field without meteorological assimilation, while in DA_MET_then_BC, they are taken from the atmospheric analysis field after assimilating the operational meteorological observations. These differences in the input to the TLM and ADM can lead to subtle variations in the analysis increments.

The overall distribution and pattern of the analysis increments of temperature (Fig. 10c), pressure (Fig. 10f), and the east-west component of horizontal wind (Fig. 10i) caused by BC assimilation in DA_MET_BC_simult are consistent with those in DA_BC and DA_MET_then_BC. However, the increment values in DA_MET_BC_simult are smaller, with values reaching approximately 0.02 K (Fig. 10c), 0.002 Pa (Fig. 10f), and -0.05 m/s (Fig. 10i), respectively. The analysis increment of relative humidity (Fig. 10l) due to BC assimilation in DA_MET_BC_simult shows a small positive value distribution, whereas in DA_BC and DA_MET_then_BC, it exhibits a negative value distribution. The differences in analysis increments of the four atmospheric variables caused by BC assimilation between DA_MET_BC_simult and DA_BC/DA_MET_then_BC may be attributed to the stronger constraints imposed by the atmospheric observations. In both DA_MET_then_BC and DA_BC, only BC surface observations are incorporated during the BC assimilation step. At this stage, the system relies solely on BC observations to correct the initial field. In the absence of atmospheric observations, BC observations play a dominant role, leading to larger analysis increments of atmospheric variables. In contrast, in DA_MET_BC_simult, both operational meteorological observations and BC surface observations are assimilated simultaneously. In this scenario, atmospheric observations may provide additional constraints on the adjustment of atmospheric fields, thereby moderating the impact of BC observations during the assimilation process. As a result, a more balanced adjustment of atmospheric variables is achieved in DA_MET_BC_simult. This behavior also highlights the importance of properly specifying the observation error covariance matrix. In future work, we plan to further examine the specification of the BC observation errors and their impact on assimilation performance.

The preliminary results obtained from this set of four experiments indicate that different BC assimilation strategies have little impact on BC analysis increments but significantly affect the analysis increments of atmospheric variables. When only BC observations are assimilated, the influence of BC on atmospheric variables is more pronounced, whereas the simultaneous assimilation of meteorological observations moderates this influence. This suggests that the presence of meteorological observations during assimilation may provide additional constraints on the adjustment of atmospheric fields, potentially reducing the degree to which the assimilation of BC observations alone can alter the atmospheric state. In this way, the integration of meteorological observations helps stabilize the adjustment process, supporting more consistent and

interpretable assimilation results. Moreover, the four experiments demonstrate that the CMA-GFS-AERO 4D-Var system has been technically implemented and is able to produce credible analysis increments in both BC and atmospheric fields. These increments display realistic spatial structures and amplitudes, indicating that the system performs as intended under the current configuration and available observations. These results offer practical evidence of the system's functionality and its potential utility for exploring the feedback of BC data assimilation on meteorological forecasts. In the future, we will conduct cycling assimilation experiments using CMA-GFS-AERO 4D-Var to gain deeper insights into the role of BC assimilation in numerical weather prediction and further refine the system for broader applications.

## 5.4 Computational performance of CMA-GFS-AERO 4D-Var

This section presents the computational performance of CMA-GFS-AERO 4D-Var from three aspects: (1) forward model, (2) TLM and ADM, and (3) 4D-Var system. We firstly evaluated the computational performance of a CMA-GFS-AERO simulation and compared it with that of the CMA-GFS simulation. Table 4 shows the computational costs for 6 h, 24 h, and 120 h integrations of CMA-GFS and CMA-GFS-AERO models. It can be seen that for 6 h, 24 h, and 120 h forecasts with the same integration time step (300 s), the same horizontal resolution of 0.25°, and the same number of CPU cores (1920 cores), the CMA-GFS-AERO simulations increase only about 10% of the computational time of the CMA-GFS simulations (As a reference, the microphysics process accounts for approximately 5% of the total computation time in CMA-GFS simulations). This shows the high efficiency of CMA-GFS-AERO forward model , which is an important factor in developing a strongly coupled aerosol-meteorology 4D-Var system.

**Table 4: Computational costs (unit: s) for 6 h, 24 h, and 120 h integrations of CMA-GFS and CMA-GFS-AERO models.**

| Model/Integration time | 6 h | 24 h | 120 h |
|---|---|---|---|
| CMA-GFS | 111.5 | 366.6 | 1725.2 |
| CMA-GFS-AERO | 121.9 | 403.5 | 1930.5 |

Note: The CMA-GFS and CMA-GFS-AERO models are integrated with the same time step (300 s), the same horizontal resolution of 0.25°, and the same CPU cores (1920 cores).

Table 5 presents the computational costs for 12 h integrations of CMA-GFS TLM/ADM and CMA-GFS-AERO TLM/ADM, and Table 6 shows the computational costs for 6 h integrations of CMA-GFS 4D-Var and CMA-GFS-AERO 4D-Var. It is apparent that with an increasing number of CPU cores, the acceleration effects of CMA-GFS-AERO TLM, ADM, and 4D-Var are comparable to those of CMA-GFS TLM, ADM, and 4D-Var. When using 1440 CPU cores, the total time of CMA-GFS-AERO TLM, ADM, and 4D-Var are approximately 1.1 times, 1.2 times, and 1.4 times those of CMA-GFS TLM,

ADM, and 4D-Var, respectively. This highlights the high efficiency and good scalability of CMA-GFS-AERO TLM, ADM, and 4D-Var, making the coupled aerosol-meteorology 4D-Var system potentially suitable for operational application.

**Table 5: Computational costs (unit: s) for 12 h integrations of CMA-GFS TLM/ADM and CMA-GFS-AERO TLM/ADM.**

| Model\CPU core | 480 | 960 | 1440 |
|---|---|---|---|
| CMA-GFS TLM | 14.63 | 8.95 | 7.04 |
| CMA-GFS ADM | 19.25 | 11.14 | 8.07 |
| CMA-GFS-AERO TLM | 16.58 | 10.18 | 7.55 |
| CMA-GFS-AERO ADM | 22.92 | 12.96 | 9.31 |

Note: CMA-GFS TLM/ADM and CMA-GFS-AERO TLM/ADM are integrated with the same time step (900 s) and the same horizontal resolution of 1°.

**Table 6: Computational costs (unit: s) for 6 h integrations of CMA-GFS 4D-Var and CMA-GFS-AERO 4D-Var.**

| 4D-Var system\CPU core | 480 | 960 | 1440 |
|---|---|---|---|
| CMA-GFS 4D-Var | 803 | 515 | 428 |
| CMA-GFS-AERO 4D-Var | 1013 | 640 | 591 |

Note: CMA-GFS 4D-Var and CMA-GFS-AERO 4D-Var are integrated with the same time step of 300 s/900 s (outer loop/inner loop), the same horizontal resolution of 0.25°/1° (outer loop/inner loop), and the same number of minimization iteration of 35 steps.

## 6 Conclusions

In this study, we developed CMA-GFS-AERO 4D-Var, a strongly coupled aerosol-meteorology data assimilation system, under the framework of the incremental analysis scheme of CMA-GFS 4D-Var. CMA-GFS-AERO 4D-Var includes three model components: forward, tangent linear, and adjoint models. CMA-GFS-AERO forward model was constructed by integrating the AERO-BC module, an aerosol module containing main aerosol physical processes of BC in the atmosphere, the code of which was extracted from the CUACE air quality model and further optimized in this work, into the CMA-GFS

weather model. The tangent linear and the adjoint of the AERO-BC module was developed and coupled online with the TLM and ADM of CMA-GFS, respectively. Thus, CMA-GFS-AERO ADM includes not only the adjoint of physical processes of BC, but also the adjoint of the meteorological model. The BC mass concentration was used as the control variable and minimized together with atmospheric variables. The background error covariance of the control variable BC adopted a modeled structure. The assimilation system used BC surface observations from the China Atmospheric Monitoring Network.

The observation error and the observation operator of BC were described in detail as well.

CMA-GFS-AERO TLM and ADM were verified by tangent linear approximation and adjoint correctness test. The results show that CMA-GFS-AERO TLM exhibits good performance in tangent linear approximation for BC, and adjoint sensitivity agrees well with tangent linear sensitivity. The CMA-GFS-AERO 4D-Var system was validated for its accuracy and rationality by the single observation experiment and the full observation experiment. The results show that assimilating BC observations can generate analysis increments not only for BC but also for atmospheric variables such as temperature, pressure, wind field, and relative humidity. This demonstrates that the newly developed CMA-GFS-AERO 4D-Var system has been technically implemented and is capable of producing credible assimilation outcomes, highlighting its potential as a useful tool for exploring the feedback of BC data assimilation on meteorological forecasts. Additionally, the computational performance of CMA-GFS-AERO 4D-Var was evaluated, and the results indicate that when using 1440 CPU cores for 6 h integrations, the total time of CMA-GFS-AERO 4D-Var are approximately 1.4 times that of CMA-GFS 4D-Var, highlighting the high efficiency of CMA-GFS-AERO 4D-Var and the potential in operational application.

The next steps are as follows. We intend to explore the impact of assimilating surface BC observations on the forecast fields of BC and atmospheric variables through cycling assimilation experiments. The CMA-GFS-AERO 4D-Var still needs to be applied to control variables for BC emission scaling factors. Further development of CMA-GFS-AERO 4D-Var will aim to assimilate more aerosol species while ensuring computational efficiency, providing an effective way to study the impact of aerosol assimilation on the analysis and forecast fields of atmospheric variables.

**Data and code availability.** The CMA-GFS model and its 4D-Var system and CUACE model were distributed by CMA Earth System Modeling and Prediction Centre (CEMC) and the Chinese Academy of Meteorological Sciences (http://www.camscma.cn/), respectively. The model was run on the PI-SUGON high-performance computer with an Intel Fortran Compiler. Due to copyright restrictions of CEMC, the full codes of the system are not freely available, interested users can contact the operational management department of CEMC or the author, Y. Liu (liuyzh@cma.gov.cn), for further assistance. Codes related to this study, including the tangent linear and adjoint interface codes for black carbon (BC), the observation operator codes for BC and the CMA-GFS-AERO 4D-Var main program, are available on Zenodo (https://zenodo.org/records/14880420; Liu et al., 2025). Model outputs of the four assimilation experiments of BC and atmosphere used in this study are also available at this website.

**Author contributions.** XZ, XS and WH envisioned and oversaw the project. YL and CW developed the model code and performed the simulations. YL, WJ, and CW prepared the manuscript with contributions from all co-authors.

**Competing interests.** The contact author has declared that none of the authors has any competing interests.

**Acknowledgements.** This work was supported by Major Program of National Natural Science Foundation of China (42090032) and National Natural Science Foundation of China (42305111). The assimilation and forecast experiments were performed on the high-performance computer Pi-SUGON of China Meteorological Administration. The development of the CMA-GFS-AERO 4D-Var system is a collaborative effort involving contributions from many colleagues. We sincerely thank the entire team for their cooperation. We also thank YaQiang Wang from CAMS for providing anthropogenic emission sources data and black carbon observations. Special thanks are extended to the three anonymous reviewers for their insightful comments and constructive suggestions, which contributed significantly to the improvement of the initial version of this paper.

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
