# Peer review of "Development of the CMA-GFS-AERO 4D-Var assimilation system v1.0- Part 1: System description and preliminary experimental results"

_Geoscientific Model Development, 2024_

## Referee Comment (RC1)

A Review of

"Development of the CMA-GFS-AERO 4D-Var assimilation system v1.0- Part 1: System description"

submitted to Geoscientific Model Development by Liu et al. (2024)

Review Decision: Major Revisions

Manuscript type: _Development and technical papers_

General Comments:

This manuscript is the first part of a two-part paper that documents the development of a strongly coupled aerosol-meteorological 4D-Var assimilation system with part I focusing on the system description. This paper lands on an important topic, coupled data assimilation, which has gained more and more attention as the general consensus is to consider different components of the Earth system as a whole. Although the structure of the paper is quite well organized and the topic is very relevant to the geoscientific modeling community, I find the current form of the paper difficult to understand in two aspects: 1) English writing and 2) descriptive but lacks interpretation. For 1), there are many spelling errors, grammar errors, and inadequate use of words. In addition, the writing is unclear to an extent that I am unsure whether my interpretation of the concepts addressed in the paper is correct. For 2), it is important for a research paper to provide a thorough description of the result as well as to provide interpretation of the result (what does it imply, what could have caused the result, or does it make sense or not, etc). This paper did a good job at the descriptive part, but lacks interpretation, especially when discussing results from the single observation experiments and the real case experiments. With that, I recommend major revisions with many comments and questions listed below.

1. As stated earlier, the writing is quite difficult to comprehend, preventing the readers from understanding the many seemingly important concepts and the value of the paper as a whole. Here are a few major concerns:

    I.  Too many acronyms are used without being introduced at all. I only listed a few here: AURA/MLS, ARPEGE/MOCAGE, MOZART3, TM5, EUARD, IMAGES, STEM-III, CAMx, CMAQ, GEOS-Chem, GRAPES-CUACE, etc.

Please make sure to introduce them when they were first mentioned and pay attention when referring to them at a later time.

II. The word "field" is spelled incorrectly as "filed" in many places. Also, background field error covariance can be shortened to background error covariance. Furthermore, "feedback" does not always have to go with "effect".

III. Inadequate use of word: the most frequently mis-used word is "set" in this paper, often times causing confusion and misunderstanding. For example, "the observation is set at 0300 UTC". A more accurate way could be "the observation is placed at 0300 UTC" or "the observation is assumed to take place at 0300 UTC". Here are a few more examples of imprecise use of word: "we set five experiments", "we set the single-point observation ideal experiment for BC", and "we further set the full observation experiment", etc.

IV. There is nothing wrong with calling it observation "increment", but a more appropriate term is observation "innovation". Also, I believe "single observation experiment" is a common term in data assimilation and there is no need to press on its "idealized" part.

V. There are many grammar errors and sentences that don't quite make sense. See comments below.

2. Abstract: I am not against calling it a *chemistry* meteorology coupled data assimilation system, however, it makes more sense and less misleading to call it an *aerosol* meteorology coupled one since only black carbon (BC) aerosol is considered so far. Besides, the name of the system, CMA-GFS-AERO, actually already suggests that it is an aerosol-meteorology coupled system. Otherwise, it would be called CMA-GFS-Chem.

3. Introduction: While it makes sense to review the previous efforts on CCMM data assimilation focusing on the variational perspective, given that this study uses a 4DVar approach, it is also important to address the previous efforts on coupled aerosol-atmosphere data assimilation using the ensemble-based approaches. I suggest shortening the description on variational approach in the introduction and include some description of the ensemble approaches and highlighting the pros and cons of a variational choice relative to an ensemble approach. Obviously 4DVar is used since it's part of the CMA-GFS, it makes sense to extend upon the

CMA-GFS 4DVar framework for aerosol coupling. Nevertheless, it is important to point out to the readers what to expect from coupling under a variational setup as opposed to an ensemble approach. For example, in a variational setup, the modeling of cross-variable component in background error covariance could be difficult, especially for aerosol vs. atmospheric processes, while in an ensemble setup one relies on ensemble estimation for cross-variable correlations. On the other hand, in a variational setup, the TLM and ADM are essential, and this can serve a natural transition to the next paragraph on the importance of ADM starting at line 52.

4. Sections 3.1 & 3.2: It reads like there is a bunch of processes, programs, subroutines, and interfaces, but how they all work together to fulfill a coupled system is unclear. Please consider re-organize/re-write these two sections to increase clarity and make sure to stay consistent with what is being shown in Figs. S2 and S3. The key is to address the main processes in AERO-BC and describe what each process does. With that, it would make the readers easier to follow the subsequent TLM and ADM of AERO-BC section since all the pieces are there in the forward section already. In addition, it is not very clear what the interfaces that connect CMA-GFS with AERO-BC in all three model components (forward, TLM, and ADM) actually do in terms of coupling, other than knowing that they act to "couple" the aerosol with the atmosphere.

5. "Section 4.1 Model setup" should be separated from the Result section since it is not a result but a description of model configuration or model setup. It might be better to consider it as a standalone section or to be included as a sub-section of section 3.

6. Page 2, Lines 32-34: What exactly are these moisture and temperature perturbations? And what these perturbations to dynamics?

7. Page 2, Line 35: "enabling to produce the optimal initial values for …" > "enabling the production of an optimal initial condition for …"

8. Page 2, Line 52: I am not sure whether "international mainstream" is a good way to say it here. How about just "major"? Also, it should be "numerical weather prediction centers", not "numerical weather centers".

9. Page 3, Lines 72-74: only the surface temperature? What happened to the 3D temperature field?

10. Page 3, Line 86 & Page 8, Line 211: "adding the control variable of BC into …" > "adding BC as a control variable into …"

11. Page 4, Lines 103-104: I am not sure what is meant by "freely combinable"? These physical parameterization processes are common to many global models. What is more important is which "schemes" are being used in each of these physical processes in CMA-GFS.

12. Page 4, Line 114: what is sectional representation method? And is there a reference for that?

13. Page 5, Lines 125: 137: $\mathbf{M}_{0>i}$ and $\mathbf{M}^T_{0>i}$ are actually linear and adjoint "models", not "operators".

14. Page 5, Lines 134-135: "after the physical and preconditioning transformation" can be omitted since it has already been stated in line 132.

15. Page 6, Line 159: To be consistent with the wordings at line 155, please consider using "forward model" instead of CCMM.

16. Page 6, Lines 160-164: These are not very relevant information.

17. Page 6, Lines 163-165 and Figure S2: These descriptions are not consistent with what is shown in Fig. S2 (a) and (b). If the idea is to show that bc_driver is part of the CMA-GFS-AERO model and acts as the interface of AERO-BC to CMA-GFS, if can be simply stated without showing Fig. S2a. As for Fig. S2b, while sf_bc, trac_vert_diff, and aerosol_bc are listed, the constant/parameter program (as stated in the texts) is missing. If the subroutines under each program is important for the readers to know and will be used/mentioned in the later part of the paper, then they deserve some explanation (e.g., what is cal_aerosol_prop? some sort of calculation of aerosol optical properties?), otherwise, they need not to be mentioned or shown. For example, the q2rh program seems to be irrelevant to AERO-BC, perhaps it can be omitted to help the readers put their focus on only the relevant parts.

18. Page 6, Line 172: it makes more sense to mention the index for size bin of BC here, instead of later at section 3.3.1, as the idea of 6 diameter bins is introduced here: $\Psi_{bc} > \Psi_{bc}^n$ where n = 1, 6.

19. Page 6, Line 173: "water-matter variables": are these water vapor and hydrometeor habits mass mixing ratios?

20. Page 7, Lines 183-184: this last sentence about TLM and ADM codes being written line-by-line manually doesn't seem quite necessary. Why is it important to mention that the code is written manually without using any automatic differentiation tool?

21. Page 8, Lines 231-233: does this suggest that distribution weight only depends on the size bin, and does not vary spatially? meaning that all grid points use the same distribution weight for a given size bin? If so, is it guaranteed that BC mass conserved after the re-distribution?

22. Page 8, Line 245: what is AE31?

23. Page 9, Lines 246-247: what are the quality control procedures?

24. Page 9, Line 257: According to Table 3 of Elbern et al. (2007), the radius of influence varies with station types, and a radius of 10 km corresponds to a rural station. Since 10 km is selected here, does that mean all 32 CAWNET stations are all rural stations? If not, please provide justifications for using a radius of 10 km.

25. Page 9, Lines 264-268: I have trouble understanding this sentence… what is point jump and what does layer jump mean?

26. Page 10, Line 269: "accumulated" > "summed" ?

27. Page 10, Lines 281-282: what is the physical meaning of such a simplification that assumes correlation coefficient is a product of vertical one times the horizontal one? What does this simplification imply?

28. Page 10, Line 290: what does $K_p$ represent and why set it to 10 here?

29. Page 11, Lines 301-302: "referenced to the relationship between length scale of humidity and the height": I have trouble understanding this one as well. Why is a relationship between humidity length scale and "height" being used for the "horizontal" length scale of BC?

30. Page 12, Lines 313-315: does this mean that BC is not cycled since the model is restarted every 6 h from CMA-GS analysis that does not have BC? But the next sentence seems to indicate that 6-h forecast of BC is used as background for the next cycle… these are conflicting ideas.

31. Page 12, Line 321: what does a global scale actually mean here? Resolution, data coverage, etc?

32. Page 12, Line 329: "an important part of introducing an adjoint model" > "an important part of introducing a new modeling component, such as the AERO-BC module"?

33. Page 13, Lines 345-346: "in an approximately linear way" > "in an approximately linear manner"?

34. Page 14, Lines 377-378: 6-h integration seems a rather long time. Is it possible that the AERO-BC processes are not very nonlinear?

35. Page 15, Lines 391-392: I have trouble understanding this one. Which coupled variable? And which physical process variable? Is it also possible that AERO-BC processes are not very nonlinear such that TL approximation is not too much different from the NL one?

36. Page 18, Lines 453-457: It will be quite helpful to add more texts to address the links between Fig. 5a and Fig. 6a as these two figures are results from the same single observation experiment with observation placed at the beginning of the window (i.e., 0300 UTC) where Fig. 5a shows the initial analysis increment while Fig. 6a shows the propagated analysis increment valid at the end of the window. Same idea for Fig. 5b and Fig. 6b, while the only difference is the timing of the observation.

37. Page 19, Lines 467-469: while I think I understand what the authors are trying to say, it is not entirely correct and perhaps not necessary to end the sentence like this. The way the system is setup (i.e., the CMA-GFS-AERO 4DVar system) by minimizing both BC and atmospheric variables together suggests it is a coupled assimilation. I think what the authors are trying to suggest is that the merits of a coupled data assimilation system cannot be fully manifested or exploited by only assimilating a BC observation at the beginning of the window.

38. Page 19, Lines 467-476: I think it is nice to have a paragraph detailing the processes in the 4DVar component of CMA-GFS-AERO that induces non-zero cross-covariance between the atmosphere and BC variables via evolving the initial background covariance with the TL modeling, even though the initial cross-covariance is zero. The current paragraph is trying to do so but remains rather descriptive and lacks interpretation. For that, I suggest checking out Section 2.1 "Coupled data assimilation" of Smith et al. (2015).

Smith, P. J., Fowler, A. M., & Lawless, A. S. (2015). Exploring strategies for coupled 4D-Var data assimilation using an idealised atmosphere–ocean model. Tellus A: Dynamic Meteorology and Oceanography, 67(1). https://doi.org/10.3402/tellusa.v67.27025

In addition, I do not think "co-correlation" is a proper word.

39. Page 19, Line 487: "in fact" should be "in reality" and one can also go on to say "in reality, unlike the single observation experiment, the BC observation is …" to further distinguish the real case from the single observation case.

40. Page 20, Lines 507-508: "assimilated all observations within the assimilation time window": How frequent is BC observation available for assimilation? I realized that this is actually mentioned in section 3.3.2 that the BC observations are hourly averaged. However, it still didn't say how frequent BC observations are assimilated in the real-case experiments.

41. Section 4.4: are BC and atmospheric variables minimized together in EXP1 and EXP2 as well? If so, please consider adding a new column in Table 3 to address whether these variables are minimized together or separately. In addition, it might be a good idea to use names that reflects the design of the experiments instead of calling them in numerical order. For example, EXP1 to EXP4 may be renamed to SCDA_BC, SCDA_MET, WCDA_BC+MET, SCDA_BC+MET where SCDA stands for strongly coupled data assimilation while WCDA refers to weakly coupled data assimilation.

42. Page 21, Lines 517-519: I am not sure if one can really say so without showing results from EXP2.

43. Pages 21-22, Lines 539-541: ok, but why? please consider including some interpretation. Are BC and atmospheric variables minimized together in EXP1 but separately in EXP3? It doesn't seem quite straightforward and easy to understand, at least to me, why would assimilating only BC observations in a strongly coupled setup leads to similar impact from assimilating both BC and atmospheric observations in a weakly coupled setup? What could be the mechanism that leads to such a consequence?

44. Page 22, Lines 553-556: I am not sure if one can make this statement by comparing the *differences* of analysis increments between EXP4 and EXP2 with *actual* analysis increments from EXP1 or EXP3. In addition, I am puzzled while trying to understand how the feedback of BC assimilation on atmospheric

variables is reduced by having also assimilated atmospheric observations in a coupled setup without actually seeing the analysis increments in EXP2 and EXP4. Some thought processes and reasonings from the authors are definitely required to be stated.

45. Page 22, Lines 556-558: This statement is maybe a little too strong. It sounds like having amplified feedback is not a good thing. Without verifying the analysis with the truth (e.g., re-analysis, or observations that are not assimilated), we do not know if the strongly coupled analysis is actually more accurate than the other ones. Hence, we do not know if amplified feedback is good or not good. Although we'd like to think (or theoretically correct to think) that analysis from a strongly coupled setup is better, we still need some evidence to prove it.

46. Page 23, Line 565: "only 10%": does this mean 10% is not much of an increase? And what is 10% increased computation time relative to? Say, if the microphysics process also takes about 10% computation time, then the readers can have a reference to judge whether 10% is large or small. Without any context, it is just a number.

47. Page 24, Lines 591-592: "three component models" > "three model components"

48. Figure 2: I believe the x-axis is missing a base 10 and a minus sign in the power of 10.

49. Figures S2-S3 and almost all figures: figure captions are rather vague and not very helpful. Both Figs. S2 and S3 present rather complicated ideas and deserve a clearer and informative description.

50. Figure 9: When are these analysis increments valid at? beginning, middle, or the end of the window?

---

## Referee Comment (RC2)

**Review of the paper "Development of the CMA-GFS-AERO 4D-Var assimilation system v1.0 - Part I: System description."**

by Yongzhu Liu & Colleagues, DOI: https://doi.org/10.5194/gmd-2024-148

**Summary & General Comments**

The article presents an overview of the development of a strongly coupled 4D-Var assimilation system where an aerosol atmospheric component, the total mass concentration of black carbon (BC), is added to the 4D-Var control vector. The article contains a detailed explanation of how the necessary linear models have been developed, by extracting and re-coding the BC-related aerosols physical modelling codes and by formulating a specific B-matrix model (and control-vector conversion) for the BC mass concentration. Rather technical validation results are displayed showing the correctness of the TL and AD models, along with preliminary experimental results. The article finishes with an outlook mostly referring to a Part II where the authors intend to discuss comprehensive experimental results on the impact of assimilating BC in a strongly coupled formulation, on the other atmospheric fields (wind, temperature, pressure, humidity).

This article Part I is overall clearly structured, with each section well introduced. As stated by the authors, the aim of the paper is to present the methodology without entering into a complete, comprehensive evaluation of experimental results in full, long-period 4D-Var assimilation experiments. Taking into account that strongly coupled atmosphere-aerosol-chemistry assimilation systems have been very little presented so far in open literature (to the reviewer's knowledge), the authors' choice to propose such an introductory Part I can be supported. Nevertheless, the paper focuses too strongly on technical sanity checks (such as the results of tests of TL and AD models which are standard and well-known tests when developing variational codes) which for themselves bring no innovative information. Conversely, the paper lacks explanations on specific scientific challenges that would strengthen the scientific interest of the paper:

1. Compared to the BC physics available in the original CUACE codes, how much has the BC physics for the CMA-GFS-AERO codes been adapted in terms of the representation of the physical processes, such as transport, chemical transformation and the interaction with radiative processes ? Taking this comment one step further, has there been any kind of simplification made when developing the BC physics modules for the linear models, any step of regularization of a non-linear formulation, or any omission of specific complex processes whose linearization was felt too difficult (at least for this v1.0 of the system) ? More explicit explanations should be provided, likely in Sections 3.1 and 3.2.

2. More explanation of why the strongly coupled case provides significantly different results on the analysis fields of the "traditional" atmospheric fields, compared with no BC assimilation or with the weakly coupled case, is missing in Section 4.4. Two striking results are displayed but eventually with very little physical interpretation while both seem to be systematic results:

a. Adding BC in the modelling and assimilation system rather than omitting this component induces a positive analysis increment on temperature. So question here: should one understand that adding BC in the forecast trajectory *anyway* will slightly increase temperature via the radiative effect of absorption? Is this effect then very systematic ? Is it local or even global ?

b. Why precisely is strongly coupled assimilation of BC causing an overall decrease of the amplitude of the analysis increments by an order of magnitude ? What are the damping retro-actions ?

The article is fairly clearly written though some specific checking of English phrase construction could be worthwhile. In the specific comments below, a few particularly unclear phrasings are stressed, which deserve further attention and rewrite by the authors. In the bibliographical section, 5 references relate to documents in Chinese. It is unclear to the reviewer what GMD's policy about references in languages other than English is. It might be appropriate that the authors confirm that they can commit to make available translated texts, should they be asked by future readers.

In conclusion, my recommendation is to accept the paper, as a Part I component to be complemented by a Part II, after revision. The goal of the revision, following the comments above, should be to strengthen the scientific explanations of the implementation of BC in the 4D-Var framework as well as to strengthen the physical interpretation of the experimental results displayed in Section 4. A further recommendation could be to extend the paper's title from "System description" to "System description and preliminary experimental results".

**Specific Comments & Typos**

Section 1.
line 69: what does "PM" stand for ?

Section 2. None.

Section 3.
line 169: "The transport processes for $\psi_{bc}$ are the same as *those* for the variables *associated with the different water species* …"
Re-phrase "water-matter" everywhere in the paper (not sure this is a good English wording, though it is understandable)

lines 175-179 (end of section 3.1):
1. the whole text should be re-written, splitting it into two separate sentences.
2. An additional explanation of how the absence of BC above model level 65 is dealt with in the models should be added. What happens regarding vertical transport for instance ? What's the impact in the adjoint code ?

lines 231-234: "Firstly …" and later "Secondly …" => reformulated these two sentences such that there is a verb. Perhaps, try with "Firstly, the distribution weights … are calculated.". The same construction would apply to the next sentence.

lines 274-277: reformulate that sentence (much too long). Make two separate ones.

line 282: I don't think that a sentence should start abruptly by "And …". Simply remove this word with no loss of clarity of the text ?

lines 286-288:
1. The vertical correlation model of the background error *is* expressed as …
2. Some additional explanation of how this formula has been obtained is required (by analogy to the water species case ? by specific experimental trials ? from external works and then add a reference ?)

Section 4.
lines 315-316: link the two sentences together and remove the start with "And …" (just use "... and …")

line 322: again, what does "PM" stand for ?

lines 364-365: replace "filed" by "*field*" . Note that the same typo appears several times later in the text, so the simplest is to make a systematic search and replace

lines 390-391: The sentence "This phenomenon indicates that …" definitely *requires a complete reformulation*. It is currently simply not understandable ! What do you want to explain ?

line 395: the caption of figure 4 mentions "simple physics" => what does "simple physics" refer to ? Do the authors refer to specific simplified physics involved in the 4D-Var models (TLM, ADM) ? If this is the case, then more explanations should be provided earlier in the text, for instance in section 3.2 (and also check my general comment above)

line 406: Remind explicitly that the time step is 300s as it's of interest here for the reader to promptly be able to convert time steps into a forecast time length

line 460 and Figure 6. Is the propagation of the BC increment by the wind the generally dominant effect ? Is this what the authors quite generally have been observing in their results ? Or conversely does this statement only apply to the very simplified context of the single-point observation experiments ? (this is what I would derive from the results later in section 4.4 when the effect of full observations strongly-coupled assimilation is shown). Nevertheless, an additional sentence here could be clarifying, in order to avoid misinterpretation with other results shown later on.

line 470. "Figure 7 depicts … *at the initial time of* the assimilation window …"

lines 516-519: should be totally re-written as they are not clear at present. A proposal : "These results suggest that the assimilation of meteorological observations has a small impact on the BC analysis increments. Furthermore, weakly and strongly coupled assimilation seem to lead to similar BC analysis increments."

line 533: "there are certain degrees of analysis increments …" => this wording is very obscure, *please reformulate*.

lines 552-558: only to mention that this is the part of section 4.4 that explicitly describes what seem to be interesting physics-related results of assimilating BC, already in these preliminary experiments. This is the part where more physical interpretation of these results is expected, on the feedback mechanisms in strongly-coupled assimilation and about the warm bias on the temperature analysis increment. (Refer to my general comments)

Section 4.5.
Only to mention that I am supportive of this section explaining the computational figures of the enhanced 4D-Var system.

Section 5.
line 614: "surface BC observation*s*"

line 615: "through batch test*s*"

Acknowledgments.
line 633: "The development of *the* CMA-GFS-AERO 4D-Var system is a systematic project"
=> what do you mean by "systematic project" ?

---

## Author Comment (AC4)

**Response to Reviewer #1 for Geoscientific Model Development:**
**Manuscript gmd-2024-148**
**By Liu et al.**

We sincerely thank Reviewer #1 for thoughtful and constructive feedback. We have carefully considered each comment and made every effort to implement all the suggested changes. The notes below address each comment in detail. Please note that Reviewer's comments are shown in bold type and our responses in plain type.

**Reviewer #1**

**General Comments:**
**This manuscript is the first part of a two-part paper that documents the development of a strongly coupled aerosol-meteorological 4D-Var assimilation system with part I focusing on the system description. This paper lands on an important topic, coupled data assimilation, which has gained more and more attention as the general consensus is to consider different components of the Earth system as a whole. Although the structure of the paper is quite well organized and the topic is very relevant to the geoscientific modeling community, I find the current form of the paper difficult to understand in two aspects: 1) English writing and 2) descriptive but lacks interpretation. For 1), there are many spelling errors, grammar errors, and inadequate use of words. In addition, the writing is unclear to an extent that I am unsure whether my interpretation of the concepts addressed in the paper is correct. For 2), it is important for a research paper to provide a thorough description of the result as well as to provide interpretation of the result (what does it imply, what could have caused the result, or does it make sense or not, etc). This paper did a good job at the descriptive part, but lacks interpretation, especially when discussing results from the single observation experiments and the real case experiments. With that, I recommend major revisions with many comments and questions listed below.**
Response: We sincerely appreciate the reviewer's constructive feedback and the recognition of the importance of our study. We acknowledge the concerns regarding the clarity of the writing and the need for more interpretation of the results.

(1) English Writing: We have carefully reviewed and thoroughly revised the manuscript to improve the clarity, grammar, and overall readability. We have corrected spelling and grammatical errors, refined word choices, and restructured sentences where necessary to enhance the coherence of the text.

(2) Interpretation of Results: We appreciate the reviewer's suggestion to provide more interpretation of our findings. In the revised manuscript, we have significantly expanded the discussion of the results, particularly in the sections on single observation experiments and real case experiments.

We acknowledge that the original expression in Section 4.4 was not sufficiently clear, which may have caused confusion. We apologize for any misunderstanding. After carefully considering the reviewer's feedback, along with comments from the other two reviewers, we have completely rewritten this section, which is now presented as Section 5.3 in the revised manuscript.

In the updated version, we have clearly introduced the objective of the four experiments, which is

to investigate the impact of different BC assimilation strategies on both BC and atmospheric variables. We have renamed the four experiments as DA_BC, DA_MET, DA_MET_then_BC, and DA_MET_BC_simult. The revised Table 3 now provides a clear description of the four experiments. We have also compared the BC analysis increments obtained from the DA_BC, DA_MET_then_BC, and DA_MET_BC_simult experiments, noting that the BC analysis increments from the DA_MET experiment are very small. Additionally, we compare the atmospheric analysis increments caused by BC assimilation in DA_BC, DA_MET_then_BC (DA_MET_then_BC - DA_MET), and DA_MET_BC_simult (DA_MET_BC_simult - DA_MET).

Our main conclusions from this analysis are as follows: The preliminary results obtained from this set of four experiments indicate that different BC assimilation strategies have little impact on BC analysis increments but significantly affect the analysis increments of atmospheric variables. When only BC observations are assimilated, the influence of BC on atmospheric variables is more pronounced, whereas the simultaneous assimilation of meteorological observations moderates this influence. This suggests that in BC assimilation, meteorological observations can help constrain the uncertainty introduced by BC observations on atmospheric variables, thereby improving the reliability of the assimilation results. Moreover, these results demonstrate the successful implementation of the newly developed CMA-GFS-AERO 4D-Var system and highlight it as an effective approach for investigating the feedback of BC data assimilation on meteorological forecasts.

In the future, we will conduct batch experiments using CMA-GFS-AERO 4D-Var to gain deeper insights into the role of BC assimilation in numerical weather prediction and further refine the system for broader applications.

Additionally, in response to Comment #24, we have adjusted the radius of influence for BC observations to 2 km, 10 km, and 20 km for urban, rural, and remote stations, respectively, according to Elbern et al. (2007). Consequently, all experiments in Section 5.3 have been redone using the updated radii, and the corresponding figures and text have been revised accordingly to reflect the new results.

For more details on the analysis, please refer to Section 5.3 of the revised manuscript. We once again appreciate the reviewer's valuable suggestions.

1. **As stated earlier, the writing is quite difficult to comprehend, preventing the readers from understanding the many seemingly important concepts and the value of the paper as a whole. Here are a few major concerns:**

   Response: Thanks for the valuable feedback. We sincerely apologize for the confusion caused by unclear writing in the manuscript. We have carefully revised the manuscript to improve its clarity and readability. We truly appreciate the reviewer's thoughtful suggestions for enhancing the quality of our manuscript.

   I. **Too many acronyms are used without being introduced at all. I only listed a few here: AURA/MLS, ARPEGE/MOCAGE, MOZART3, TM5, EUARD, IMAGES, STEM-III, CAMx, CMAQ, GEOS-Chem, GRAPES-CUACE, etc. Please make sure to introduce them when they were first mentioned and pay attention when referring to them at a later time.**

Response: Thanks for pointing this out. We have carefully reviewed the manuscript and ensured that all acronyms, including AURA/MLS (Microwave Limb Sounder aboard the Aura satellite), ARPEGE/MOCAGE (Action de Recherche Petite Echelle Grande Echelle/Modèle de Chimie Atmosphérique de Grande Echelle), MOZART3 (Model for Ozone and Related Chemical Tracers, version 3), TM5 (Transport Model, version 5), EUARD (The University of Cologne European Air Pollution Dispersion Chemistry Transport Model), IMAGES (Intermediate Model of Global Evolution of Species), STEM-III (Sulfur Transport Eulerian Model), CAMx (Comprehensive Air Quality Model with Extensions model), CMAQ (Community Multiscale Air Quality model), GRAPES-CUACE (Global/Regional Assimilation and PrEdiction System coupled with CMA Unified Atmospheric Chemistry Environmental Forecasting System), etc., are properly introduced when they are first mentioned.

GEOS-Chem is a global 3-D model of atmospheric chemistry driven by meteorological input from the Goddard Earth Observing System (GEOS) of the NASA Global Modeling and Assimilation Office. CHIMERE is a three-dimensional chemical transport model used for atmospheric chemistry and air quality simulations. It was initially developed by the Pierre-Simon Laplace Institute (IPSL) in Paris, France. In the scientific literature, GEOS-Chem and CHIMERE are commonly referred to by their acronym, and there is no widely accepted or official full form for them. Therefore, we have followed the standard practice of using the acronym, consistent with other studies. Interested readers can refer to the relevant references in the manuscript for more detailed information.

Additionally, we have also ensured consistency and clarity when referring to them later in the text to avoid any confusion for the readers.

II. **The word "field" is spelled incorrectly as "filed" in many places. Also, background field error covariance can be shortened to background error covariance. Furthermore, "feedback" does not always have to go with "effect".**

Response: We sincerely apologize for the misspelling of the word "field", and this has been revised throughout the manuscript. Additionally, we have shortened "background field error covariance" to "background error covariance" as suggested. And "feedback effect" has also been revised to "feedback".

III. **Inadequate use of word: the most frequently mis-used word is "set" in this paper, often times causing confusion and misunderstanding. For example, "the observation is set at 0300 UTC". A more accurate way could be "the observation is placed at 0300 UTC" or "the observation is assumed to take place at 0300 UTC". Here are a few more examples of imprecise use of word: "we set five experiments", "we set the single-point observation ideal experiment for BC", and "we further set the full observation experiment", etc.**

Response: We sincerely apologize for the inadequate and imprecise use of the word "set", and we have revised "the observation is set at …" to "the observation is placed at …" as suggested. Additionally, we have revised "we set five experiments", "we set the single-point observation ideal experiment for BC", and "we further set the full

observation experiment" to "we conducted five experiments", "we performed the single observation experiment for BC", "we further conducted the full observation experiment", respectively.

IV. **There is nothing wrong with calling it observation "increment", but a more appropriate term is observation "innovation". Also, I believe "single observation experiment" is a common term in data assimilation and there is no need to press on its "idealized" part.**

Response: According to the reviewer's good instructions, we have revised "observation increment" to "observation innovation". And we have also changed "single-point observation ideal experiment" to "single observation experiment" as suggested.

V. **There are many grammar errors and sentences that don't quite make sense. See comments below.**

Response: Thanks for the valuable feedback. We sincerely apologize for the grammar errors and unclear sentences in the manuscript. We have carefully reviewed the comments below and revised the manuscript to improve both clarity and grammatical accuracy. Specifically, we have addressed the highlighted issues and ensured that all sentences are well-structured and easy to understand. We truly appreciate the reviewer's attention to these details and are committed to enhancing the quality of the manuscript.

2. **Abstract: I am not against calling it a *chemistry* meteorology coupled data assimilation system, however, it makes more sense and less misleading to call it an *aerosol* meteorology coupled one since only black carbon (BC) aerosol is considered so far. Besides, the name of the system, CMA-GFS-AERO, actually already suggests that it is an aerosol-meteorology coupled system. Otherwise, it would be called CMA-GFS-Chem.**

Response: Thanks for the insightful comment. Following the recommendation, we have changed "chemistry meteorology coupled data assimilation system" to "aerosol-meteorology coupled data assimilation system" in Abstract and the main text.

3. **Introduction: While it makes sense to review the previous efforts on CCMM data assimilation focusing on the variational perspective, given that this study uses a 4DVar approach, it is also important to address the previous efforts on coupled aerosol-atmosphere data assimilation using the ensemble-based approaches. I suggest shortening the description on variational approach in the introduction and include some description of the ensemble approaches and highlighting the pros and cons of a variational choice relative to an ensemble approach. Obviously 4DVar is used since it's part of the CMA-GFS, it makes sense to extend upon the CMA-GFS 4DVar framework for aerosol coupling. Nevertheless, it is important to point out to the readers what to expect from coupling under a variational setup as opposed to an ensemble approach. For example, in a variational setup, the modeling of cross-variable component in background error covariance could be difficult, especially for aerosol vs. atmospheric processes, while in an ensemble setup one relies on ensemble estimation for**

**cross-variable correlations. On the other hand, in a variational setup, the TLM and ADM are essential, and this can serve a natural transition to the next paragraph on the importance of ADM starting at line 52.**

Response: Thanks for the insightful comment. Following the recommendation, we have shortened the description on variational approach in the introduction and included previous efforts on coupled aerosol-atmosphere data assimilation using the Ensemble Kalman filter (EnKF) method (Pagowski and Grell, 2012; Bocquet et al., 2015). We have also discussed the advantages and disadvantages of the 4D-Var method relative to the EnKF approach. Specifically, we have highlighted that the EnKF approach relies on ensemble-based estimates for the background error covariance, while in a variational setup, modeling cross-variable components in the background error covariance can be challenging in data assimilation for CCMM. Additionally, we have emphasized that in a 4D-Var framework, the TLM and ADM are essential, which naturally leads into the subsequent discussion on the significant advancements in atmospheric chemistry adjoint modeling.

We have updated the corresponding section of the Introduction as follows:

"…Flemming et al. (2011) utilized the 4D-Var system of the Integrated Forecast System (IFS) coupled with three different $O_3$ chemistry mechanisms, including a linear chemistry, the MOZART3 (Model for Ozone and Related Chemical Tracers, version 3) chemistry, and the TM5 (Transport Model, version 5) chemistry, to assimilate $O_3$ data from four satellite-borne sensors to improve the simulation of the stratospheric $O_3$ hole in 2008. Previous efforts have also explored the application of ensemble-based methods for data assimilation with a CCMM (Pagowski and Grell, 2012; Bocquet et al., 2015). Pagowski and Grell (2012) assimilated surface measurements of fine aerosols using the Weather Research and Forecasting-Chemistry model (WRF-Chem) and the Ensemble Kalman filter (EnKF) method. Bocquet et al. (2015) also presented an application of the EnKF to assimilate surface fine particulate matter observations and meteorological observations with the WRF-Chem model over the eastern part of North America. Results demonstrated that a large positive impact of aerosol data assimilation on aerosol concentrations, while the effect of meteorological observation assimilation on aerosol concentration is rather minor. All the preceding studies have laid good foundations for data assimilation with CCMM. However, since CCMM are fairly recent, the development and applications of data assimilation in CCMM are still limited. Further research and more attention are required, especially in terms of the potential feedbacks of chemical data assimilation on meteorological forecasts. Additionally, EnKF estimates background error covariance through ensemble forecasts, which rely on a limited number of ensemble members (Zhu et al., 2022). In high-dimensional problems, the limited number of samples may not be able to fully capture all the error characteristics, resulting the inaccurate of the estimation of background error covariance. In contrast, 4D-Var generally offers higher accuracy for high-dimensional problems by incorporating both the full observational data and model dynamics within the assimilation window, resulting in more precise state estimation. While the flow dependence of the background error covariance is implicitly realized within the assimilation window in 4D-Var, modeling the cross-variable component of the covariance presents a significant challenge in data assimilation for CCMM. Furthermore, the tangent linear model (TLM) and the adjoint model (ADM) are essential components of 4D-Var, but their development is often fraught with difficulties.

Significant efforts have been made in the field of atmospheric chemistry adjoint modeling. Elbern and Schmidt (1999) first constructed the ADM of a 3D CTM, EUARD (The University of Cologne European Air Pollution Dispersion Chemistry Transport Model). Inspired by this work…"

4.  **Sections 3.1 & 3.2: It reads like there is a bunch of processes, programs, subroutines, and interfaces, but how they all work together to fulfill a coupled system is unclear. Please consider re-organize/re-write these two sections to increase clarity and make sure to stay consistent with what is being shown in Figs. S2 and S3. The key is to address the main processes in AERO-BC and describe what each process does. With that, it would make the readers easier to follow the subsequent TLM and ADM of AERO-BC section since all the pieces are there in the forward section already. In addition, it is not very clear what the interfaces that connect CMA-GFS with AERO-BC in all three model components (forward, TLM, and ADM) actually do in terms of coupling, other than knowing that they act to "couple" the aerosol with the atmosphere.**

    Response: Thanks for the insightful comment. Following the recommendation, we have rewritten Sections 3.1 and 3.2 to improve clarity. In doing so, we also carefully considered Comment #16 and Comment #17 and concluded that Figures S2 and S3 were not essential. The revised Sections 3.1 and 3.2 now provide a clearer and more self-contained description of the key processes in AERO-BC, making the figures unnecessary. The revised Sections 3.1 and 3.2 explicitly describe the key processes in AERO-BC and their respective roles. Additionally, we have clarified the function of the interface programs that connect CMA-GFS with AERO-BC in all three model components (forward model, TLM, and ADM), ensuring a clearer explanation of their coupling mechanism. These interface programs are responsible for transferring meteorological parameters (e.g., temperature, wind, and humidity) from CMA-GFS to AERO-BC, extending the spatial dimension from 1-D to 3-D, and reading emissions for AERO-BC.

    The revised Sections 3.1 and 3.2 now reads:

    "3.1 CMA-GFS-AERO forward model

    In this work, for the sake of interest in BC and the consideration of computational efficiency, we developed the CMA-GFS-AERO forward model by integrating the aerosol module AERO-BC into CMA-GFS v4.0. The AERO-BC module was created by extracting BC-related codes from the CUACE model, with its functionality aligning with the BC aerosol processes in the CAM module of CUACE. In other words, the physical processes for BC in AERO-BC are identical to those in the CAM module, with no changes made. The main differences lie in the engineering aspect: (1) while the CAM module was originally written in Fortran 77, the AERO-BC code has been rewritten in Fortran 90; (2) since CAM in CUACE deals with six types of aerosols, the code structure is somewhat complex and redundant, whereas AERO-BC focuses solely on BC, resulting in a simpler and more streamlined structure. These updates improve code readability and enhance computational efficiency, without affecting the underlying physical processes.

    …

    The main processes in AERO-BC include: (1) calculating the emission flux of BC through the surface flux calculation module, (2) calculating the vertical diffusion trend of BC by

solving the vertical diffusion equation, and (3) simulating key BC aerosol processes in the atmosphere, including hygroscopic growth, coagulation, nucleation, condensation, dry deposition/sedimentation, and below-cloud scavenging. For more details, please refer to the relevant literature on the CAM module (Gong et al., 2003; Gong and Zhang et al., 2008; Wang et al., 2010; Zhou et al., 2012). In the integration of AERO-BC with CMA-GFS, the interface programs transfer meteorological parameters (e.g., temperature, wind, and humidity) from CMA-GFS to AERO-BC, extend the spatial dimension from 1-D to 3-D, and read emissions for AERO-BC. The transport processes for $\psi_{bc}^{n}$ are the same as those for the variables associated with the different water species in CMA-GFS, using the hybrid PRM and QMSL schemes (Su et al., 2013).

…

3.2 CMA-GFS-AERO TLM and ADM

…

The TL of the AERO-BC can be obtained by linearizing $\boldsymbol{F}$, expressed as

$$\delta Y = \mathbf{F}\delta C = \frac{\partial \boldsymbol{F}}{\partial C}\delta C, \tag{5}$$

where $\mathbf{F}$ is the TL model operator, $\delta C$ and $\delta Y$ represent perturbations of input and output variables of the AERO-BC, respectively.

The adjoint of the AERO-BC is essentially the transpose of the AERO-BC TL, expressed as

$$\delta C^{*} = \mathbf{F}^{T}\delta Y^{*}, \tag{6}$$

where $\mathbf{F}^{T}$ is the adjoint operator of $\mathbf{F}$, $\delta Y^{*}$ and $\delta C^{*}$ represent input and output variables of the adjoint of AERO-BC, respectively.

In constructing the TL and the adjoint of AERO-BC, no simplifications were made to the AERO-BC processes. Specifically, no regularization was applied to the nonlinear equations, nor were any complex processes, which were difficult to linearize, omitted. As a result, the TL and the adjoint of AERO-BC fully include all processes related to emission flux, vertical diffusion, and aerosol physical processes as described in Section 3.1.

The TL and the adjoint of AERO-BC are 1-D modules with fixed latitude and longitude coordinates. To extend them to 3-D, the tangent linear and the adjoint of the interface programs were also constructed. Furthermore, the tangent linear and the adjoint of BC transport processes follow the same framework as those for the variables associated with the different water species in the CMA-GFS TLM and ADM, utilizing the tangent linear and the adjoint of QMSL. In this way, the 3-D parameters could be transferred from CMA-GFS to AERO-BC. Thus, we obtained the CMA-GFS-AERO TLM and ADM."

5. **"Section 4.1 Model setup" should be separated from the Result section since it is not a result but a description of model configuration or model setup. It might be better to consider it as a standalone section or to be included as a sub-section of section 3.**

Response: We sincerely appreciate the reviewer's valuable suggestion. Following the recommendation, we have separated the "Model setup" from the Results section and designated it as a standalone section, now titled "4 Model setup". Consequently, the Results and Conclusions sections have been renumbered as Section 5 and 6, respectively. Additionally, we have updated the corresponding descriptions in the Introduction as follows:

"Section 2 introduces the methods, Section 3 describes the development of CMA-GFS-AERO

4D-Var, Section 4 provides the model setup, Section 5 presents the results, and the conclusions are found in Section 6."

6. **Page 2, Lines 32-34: What exactly are these moisture and temperature perturbations? And what these perturbations to dynamics?**

Response: Thanks for the valuable comment. The "moisture and temperature perturbations" refer to changes in the moisture and temperature fields in the atmosphere that result from aerosol microphysics and radiative forcing. These changes occur due to the interactions between aerosols, radiation, and cloud processes, which alter the local moisture and temperature distributions. The "perturbations to dynamics" refer to how these changes in moisture and temperature affect atmospheric processes such as convection, circulation, and stability, which in turn influence the overall atmospheric dynamics. This feedback mechanism is incorporated in the coupled chemistry meteorology models (CCMM), but is not typically included in chemical transport models (CTM).

To enhance clarity, we have revised lines 32-34 as follows:

"…CCMM account for the feedback mechanism between aerosols and meteorology, specifically the moisture and temperature perturbations resulting from aerosol microphysics and radiative forcing, which, in turn, affect atmospheric dynamics such as convection, circulation, and stability, whereas CTM lack the capability to incorporate these feedback mechanisms ..."

7. **Page 2, Line 35: "enabling to produce the optimal initial values for …" > "enabling the production of an optimal initial condition for …"**

Response: This has been revised as suggested.

8. **Page 2, Line 52: I am not sure whether "international mainstream" is a good way to say it here. How about just "major"? Also, it should be "numerical weather prediction centers", not "numerical weather centers".**

Response: Thanks for pointing this out. We agree that "international mainstream" is not a good way to say it here, and "major" is a more appropriate term. We also agree that it should be "numerical weather prediction centers", not "numerical weather centers". Based on the third suggestion in the previous comment, we have revised this section and removed the sentence entirely.

9. **Page 3, Lines 72-74: only the surface temperature? What happened to the 3D temperature field?**

Response: Thanks for the insightful comment. The impact of black carbon (BC) on temperature is indeed not limited to the surface. BC influences both the surface temperature and the three-dimensional (3D) temperature field through its absorption of solar radiation in the visible to infrared wavelength range. We have revised Lines 72-74 as follows:

"…BC is also the main optically absorbing component of atmospheric aerosols, effectively absorbing solar radiation in the visible to infrared wavelength range, thus affecting not only the surface temperature but also the 3D temperature field…"

10. **Page 3, Line 86 & Page 8, Line 211: "adding the control variable of BC into …" > "adding BC as a control variable into …"**

   Response: This has been revised as suggested.

11. **Page 4, Lines 103-104: I am not sure what is meant by "freely combinable"? These physical parameterization processes are common to many global models. What is more important is which "schemes" are being used in each of these physical processes in CMA-GFS.**

   Response: Thanks for the insightful comment. By "freely combinable", we are referring to the flexibility in choosing among several physical parameterization schemes for a specific physical process, allowing users to select the most suitable one for their needs.

   According to the reviewer's valuable suggestions, we have modified the expression to clarify the specific schemes used for each physical process in CMA-GFS in this study, as follows:

   "…The physical parameterization schemes used in this work mainly include the Simplified Arakawa Schubert (SAS) cumulus convection scheme (Arakawa and Schubert, 1974; Liu et al., 2015), the double-moment cloud microphysics scheme (Liu et al., 2003a, 2003b; Li et al., 2024), the Rapid Radiative Transfer Model for the GCM (RRTMG) longwave and shortwave radiation schemes (Mlawer et al., 1997; Morcrette et al., 2008), the Common Land Model (CoLM) land surface scheme (Dai et al., 2003), and the New Medium Range Forecast (NMRF) boundary layer scheme (Hong and Pan, 1996; Han and Pan, 2011)…"

12. **Page 4, Line 114: what is sectional representation method? And is there a reference for that?**

   Response: Thanks for pointing this out. Sectional representation is one of the common methods to represent particle size distributions in atmospheric chemistry models. In the sectional representation approach, the aerosol size distribution is generally approximated by a set of contiguous, nonoverlapping and discrete size bins. This representation of aerosol size distribution is employed for its flexibility to treat processes including multicomponent interactions such as coagulation, condensation and chemical processes. We list several references related to the sectional representation method here:

   [1] Gelbard, F., Tambour, Y., Seinfeld, J. H.: Sectional representations for simulating aerosol dynamics. J. Colloid Interf. Sci., 76, 541-556, https://doi.org/10.1016/0021-9797(80)90394-X, 1980.

   [2] Meng, Z., Dabdub, D., Seinfeld, J. H.: Size-resolved and chemically resolved model of atmospheric aerosol dynamics, J. Geophys. Res., 103, 3419-3435, https://doi.org/10.1029/97JD02796, 1998.

   [3] Gong, S.L., Barrie, L.A., Blanchet, J.P., Von Salzen, K., Lohmann, U., Lesins, G., Spacek, L., Zhang, L.M., Girard, E., Lin, H.: Canadian Aerosol Module: A size-segregated simulation of atmospheric aerosol processes for climate and air quality models 1. Module development, J. Geophys. Res.-Atmos., 108, AAC 3-1-AAC 3-16, https://doi.org/10.1029/2001JD002002, 2003.

   We have also added an explanation and the references of sectional representation method in the revised manuscript as follows:

   "…and each of them utilizes the sectional representation method (Gelbard et al., 1980; Meng

et al., 1998; Gong et al., 2003), in which the aerosol size distribution is generally approximated by a set of contiguous, nonoverlapping and discrete size bins, to represent particle size distributions…"

13. **Page 5, Lines 125: 137: M0>i and M$^T$0>i are actually linear and adjoint "models", not "operators".**
Response: This has been revised as suggested.

14. **Page 5, Lines 134-135: "after the physical and preconditioning transformation" can be omitted since it has already been stated in line 132.**
Response: This has been revised as suggested.

15. **Page 6, Line 159: To be consistent with the wordings at line 155, please consider using "forward model" instead of CCMM.**
Response: According to the reviewer's good instructions, we have used "CMA-GFS-AERO forward model" instead of "CMA-GFS-AERO CCMM" throughout the manuscript.

16. **Page 6, Lines 160-164: These are not very relevant information.**
Response: Thanks for the comment. The content in Lines 160-164 has been rewritten in the revised manuscript. For details, please refer to our response to Comment #4.

17. **Page 6, Lines 163-165 and Figure S2: These descriptions are not consistent with what is shown in Fig. S2 (a) and (b). If the idea is to show that bc_driver is part of the CMA-GFS-AERO model and acts as the interface of AERO-BC to CMA-GFS, if can be simply stated without showing Fig. S2a. As for Fig. S2b, while sf_bc, trac_vert_diff, and aerosol_bc are listed, the constant/parameter program (as stated in the texts) is missing. If the subroutines under each program is important for the readers to know and will be used/mentioned in the later part of the paper, then they deserve some explanation (e.g., what is cal_aerosol_prop? some sort of calculation of aerosol optical properties?), otherwise, they need not to be mentioned or shown. For example, the q2rh program seems to be irrelevant to AERO-BC, perhaps it can be omitted to help the readers put their focus on only the relevant parts.**
Response: Thanks for the detailed comments. In response to both Comment #4 and Comment #16, we have rewritten Sections 3.1 and 3.2. The revised Sections 3.1 and 3.2 now provide a clearer and more self-contained description of the key processes in AERO-BC, making the figures unnecessary. Therefore, we have removed Figures S2 and S3 from the manuscript. For more details, please refer to our response to Comment #4.

18. **Page 6, Line 172: it makes more sense to mention the index for size bin of BC here, instead of later at section 3.3.1, as the idea of 6 diameter bins is introduced here: Ψbc > Ψbcn where n = 1, 6.**
Response: We sincerely appreciate the reviewer's valuable suggestion. Following the recommendation, we have introduced the index for the size bin of BC in Section 3.1 as follows:
"Thus, six new prognostic variables for the mass mixing ratio of BC, denoted as $\psi_{bc}^n$(unit: kg/kg),

where $n = 1, ..., 6,$ are added in the dynamical framework of CMA-GFS."

Accordingly, we have removed the introduction of $\psi_{bc}^n$ in Section 3.3.1. Additionally, all occurrences of $\psi_{bc}$ throughout the manuscript have been updated to $\psi_{bc}^n$.

19. **Page 6, Line 173: "water-matter variables": are these water vapor and hydrometeor habits mass mixing ratios?**

Response: Thanks for the comment. We sincerely apologize for the lack of clarity regarding the term "water-matter variables". By "water-matter variables", we were referring to the variables associated with the different water species. In the revised manuscript, we have replaced "water-matter variables" with "variables associated with the different water species" for clarity.

20. **Page 7, Lines 183-184: this last sentence about TLM and ADM codes being written line-by-line manually doesn't seem quite necessary. Why is it important to mention that the code is written manually without using any automatic differentiation tool?**

Response: Thanks for pointing this out. Zou et al. (1997) emphasized that due to the complexity of numerical model codes, even when independent, dependent, and active variables are correctly specified, automatic adjoint generators do not necessarily produce correct adjoint codes. Clean and accurate adjoint codes often require manual intervention. Our experience in developing the CMA-GFS adjoint model (Zhang et al., 2019) further confirms that adjoint codes generated by automatic differentiation tools often suffer from issues such as poor readability, poor maintainability, low efficiency, and even errors. To ensure the quality, readability, maintainability, and efficiency of the tangent linear and adjoint codes, we opted to write them line-by-line manually, without using any automatic differentiation tool.

We have also added an explanation in the revised manuscript as follows:

"…Since adjoint codes generated by automatic differentiation tools often suffer from issues such as poor readability and maintainability, low efficiency and even errors due to the complexity of numerical models (Zou et al., 1997), the tangent linear and adjoint codes in this study were written line-by-line manually, without using any automatic differentiation tool."

21. **Page 8, Lines 231-233: does this suggest that distribution weight only depends on the size bin, and does not vary spatially? meaning that all grid points use the same distribution weight for a given size bin? If so, is it guaranteed that BC mass conserved after the re-distribution?**

Response: Thanks for the insightful comment. In our methodology, the distribution weights ($\omega^n$) are calculated based on the entire three-dimensional domain, following the equation $\omega^n = \frac{\sum_1^N \psi_{bc}{}^n}{\sum_{n=1}^6 \left(\sum_1^N \psi_{bc}{}^n\right)}$, where $N$ represents the number of three-dimensional grid points. This means that for a given size bin, $\omega^n$ is uniform across all grid points and does not vary spatially. This approach ensures that the weight distribution reflects the global characteristics, rather than being influenced by local variations. By doing so, a global weighting factor is provided, which allows for a reasonable allocation of analysis increments. While there may be small variations in the distribution weights ($\omega^n$) across different grid points, these differences are

relatively minor. Therefore, using a global weighting factor does not result in a significant violation of BC mass conservation.

22. **Page 8, Line 245: what is AE31?**

Response: Thanks for pointing this out. The AE31 is a model of the Aethalometer manufactured by Magee Scientific (USA), which is widely used for the real-time measurement of BC concentration. The AE31 determines the mass concentration of BC particles collected from air samples, flowing through a quartz filter. The instrument measures the transmission through the filter over a wide spectrum of wavelengths from 370 nm to 950 nm. Light at the selected wavelength is transmitted through control and sample filters, and the attenuation change in the filter is then translated into the BC mass concentration. In our study, we used the BC concentration measured at the recommended wavelength of 880 nm.

We have revised the description of AE31 in Section 3.3.2 as follows:

"…The monitoring of BC in CAWNET was conducted using an Aethalometer, AE31, which is one of the models produced by Magee Scientific (USA, https://www.aerosolmageesci.com). The AE31 determines mass concentration of BC particles collected from air samples, flowing through a quartz filter. The instrument measures the transmission through the filter over a wide spectrum of wavelengths from 370 nm to 950 nm. Light at the selected wavelength is transmitted through control and sample filters, and the attenuation change in the filter is then translated into the BC mass concentration. In this study, we used the BC concentration measured at the recommended wavelength of 880 nm…"

23. **Page 9, Lines 246-247: what are the quality control procedures?**

Response: Thanks for pointing this out. The original sampling frequency of the AE31 is 5 minutes. Due to various factors, including instrument-related issues or human error, the BC observations may contain abnormal values such as missing values, negative values, or extreme outliers. Therefore, quality control is necessary before using the BC concentrations in our analysis. In this study, the quality control procedures for BC observations mainly focus on eliminating abnormal values and filling in missing data. The quality control steps are as follows:

(1) Eliminating abnormal values. During the calculation of hourly average values from the 5-minute sampled data, any BC concentration values that are significantly different from the hourly average (i.e., those where the absolute difference exceeds three times the standard deviation) are considered abnormal and discarded. Additionally, any bad data flagged by the instrument's monitoring system are also removed.

(2) Filling in missing values. If more than one-third of the data for a given hour is missing, or if there are more than three consecutive missing values, the entire hour's data is discarded. For other cases, linear interpolation is applied to fill in the missing values.

After applying these quality control procedures, we obtained the hourly average BC concentrations used in this study.

We have added detailed quality control procedures in Section 3.3.2 as follows:

"…The AE31 measures BC concentrations every 5 minutes. We performed quality control on the original data and obtained the hourly average values, which were used in the BC assimilation experiments. The quality control procedures are as follows:

(1) Eliminating abnormal values. During the calculation of hourly averages from the 5-minute

sampled data, any BC concentration values that differ significantly from the hourly average (i.e., those where the absolute difference exceeds three times the standard deviation) are considered abnormal and discarded. Additionally, any bad data flagged by the instrument's monitoring system are also removed.

(2) Filling in missing values. If more than one-third of the data for a given hour is missing, or if there are more than three consecutive missing values, the entire hour's data is discarded. For other cases, linear interpolation is applied to fill in the missing values."

24. **Page 9, Line 257: According to Table 3 of Elbern et al. (2007), the radius of influence varies with station types, and a radius of 10 km corresponds to a rural station. Since 10 km is selected here, does that mean all 32 CAWNET stations are all rural stations? If not, please provide justifications for using a radius of 10 km.**

Response: Thanks for the insightful comment. The 32 CAWNET stations include 11 urban, 17 rural and 4 remote stations. As noted in Table 3 of Elbern et al. (2007), the radius of influence does vary with station types. Our initial selection of a uniform 10 km radius for all 32 CAWNET stations was indeed inappropriate. In the revised manuscript, we adopted radii of 2 km, 10 km, and 20 km for urban, rural, and remote stations, respectively, according to Table 3 of Elbern et al. (2007). The following revisions have been made in the text:

"The BC observation data were collected from 32 stations (Guo et al., 2020), including 11 urban, 17 rural and 4 remote stations…"

"…and $L$ is the radius of influence of a BC observation. According to Elbern et al. (2007), $L$ was set to 2 km, 10 km, and 20 km for urban, rural, and remote stations, respectively…"

Additionally, all experiments in Section 5.3 have been redone using the updated radii, and the corresponding figures and text have been updated to reflect the new results. Please refer to the revised Section 5.3 for details.

25. **Page 9, Lines 264-268: I have trouble understanding this sentence… what is point jump and what does layer jump mean?**

Response: We sincerely apologize for the unclear expression of "point jump" and "layer jump". What we intended to express is as follows:

In a data assimilation system, the observation operator serves two primary functions: (1) transforming model state variables into observed physical quantities and (2) interpolating the background (or analysis) field to the observation locations. The transformation of physical quantities depends on the observation type, while the spatial interpolation consists of both horizontal and vertical components. Since the CMA-GFS-AERO 4D-Var system adopts the Charney-Phillips staggered grid in the vertical direction and the Arakawa-C grid in the horizontal direction, different physical variables (e.g., wind, temperature, etc.) are positioned at different locations within the model grid. To minimize errors introduced by variable transformations and spatial interpolation in the observation operator, it is necessary to correctly account for the staggered placement of variables. This involves handling the horizontal staggering of grid points and the vertical staggering of layers when interpolating observations.

Additionally, we have revised the corresponding sentence in the manuscript as follows:

"…Since the CMA-GFS-AERO 4D-Var system adopts the Charney-Phillips staggered grid in

the vertical direction and the Arakawa-C grid in the horizontal direction, the observation operator must account for the staggered locations of different physical variables. To minimize errors introduced by variable transformations and spatial interpolation, appropriate handling of horizontal staggering and vertical layer transitions is required…"

26. **Page 10, Line 269: "accumulated" > "summed" ?**

Response: This has been revised as suggested.

27. **Page 10, Lines 281-282: what is the physical meaning of such a simplification that assumes correlation coefficient is a product of vertical one times the horizontal one? What does this simplification imply?**

Response: Thanks for the valuable comment. In the CMA-GFS 4D-Var assimilation system, the background error covariance matrix is highly complex due to the presence of both inter-variable correlations and correlations at different spatial locations of the same variable. Given the large dimensionality of the problem, direct computation is not feasible, and a dimensionality reduction approach is necessary.

In the CMA-GFS 4D-Var system, the state variables (such as the non-dimensional pressure, potential temperature, horizontal wind components, vertical wind component, and specific humidity) are first transformed through physical transformations into the stream function, non-equilibrium velocity potential, non-equilibrium dimensionless pressure    , and specific humidity. These variables are independent of each other, so only the error covariance between the same variables at different spatial locations exists. As a result, the background error covariance matrix becomes a block diagonal matrix.

Further, through preconditioning transformations, the background error covariance matrix is simplified to an identity matrix. During this transformation process, we assume that the correlations at different spatial positions for the same variable can be separated into horizontal and vertical components. Specifically, the vertical correlation model is assumed to be identical in the horizontal direction, and the horizontal correlation model is assumed to be the same in the vertical direction. This allows us to represent the background error covariance matrix using the Kronecker product of the horizontal and vertical correlation matrices, significantly simplifying the computation of the matrix.

For further details, please refer to Zhang et al. (2019): Zhang, L., Liu, Y., Liu, Y., Gong, J., Lu, H., Jin, Z., Tian, W., Liu, G., Zhou, B., Zhao, B.: The operational global four - dimensional variational data assimilation system at the China Meteorological Administration, Q. J. Roy. Meteor. Soc., 145, 1882-1896, https://doi.org/10.1002/qj.3533, 2019.

We have also included this reference in the revised manuscript as follows:

"…Therefore, in the CMA-GFS 4D-Var assimilation system, a simplification is made by assuming that the correlation coefficient can be expressed as the product of the vertical correlation coefficient and the horizontal correlation coefficient (Zhang et al., 2019)…"

28. **Page 10, Line 290: what does $K_p$ represent and why set it to 10 here?**

Response: Thanks for the comment. $k_p$ is the constant coefficient in the formula. In this study, we set the value of $k_p$ to 10 for the control variable BC, following the value used for the control variable of humidity in the CMA-GFS 4D-Var system.

Additionally, we have included an explanation for $k_p$ in the revised manuscript as follows:
"…and $k_p$ is the constant coefficient (Bergman, 1979). Following the value of $k_p$ used for the control variable of humidity in the CMA-GFS 4D-Var system, we set $k_p$ to 10 for the control variable BC…"

29. **Page 11, Lines 301-302: "referenced to the relationship between length scale of humidity and the height": I have trouble understanding this one as well. Why is a relationship between humidity length scale and "height" being used for the "horizontal" length scale of BC?**

Response: Thanks for pointing this out. We sincerely apologize for the confusion caused by the unclear expression in Lines 301-302. What we intended to convey is that the length scale for the control variable BC varies with height in the model, following the way the length scale of the humidity variable varies with height in the CMA-GFS 4D-Var system.

We have revised the expression in Lines 301-302 as follows:

"…The length scale for the control variable BC varies with height in the model, following the way the length scale of the humidity variable varies with height in the CMA-GFS 4D-Var system, which is shown in Table 1…"

30. **Page 12, Lines 313-315: does this mean that BC is not cycled since the model is restarted every 6 h from CMA-GS analysis that does not have BC? But the next sentence seems to indicate that 6-h forecast of BC is used as background for the next cycle… these are conflicting ideas.**

Response: We sincerely apologize for the unclear expression in Lines 313-315, which may have caused confusion. To clarify, we have revised the text in the revised manuscript as follows:

"…The forecast of the CMA-GFS-AERO model started at 0300 UTC on October 1, 2016, and was restarted every 6 h. The meteorological initial fields for each 6-h cycle were obtained from the operational CMA-GFS analysis. The BC field was initialized with null concentrations at 0300 UTC on October 1, 2016. From the second forecast cycle onward, the initial conditions of BC were derived from the BC field at the end of the previous 6-h forecast, allowing the BC field to be cycled…"

31. **Page 12, Line 321: what does a global scale actually mean here? Resolution, data coverage, etc?**

Response: Thanks for pointing this out. By "a global scale", we are referring to the data coverage of the HTAP, which provides emissions data for a wide range of regions worldwide.

We have modified the expression in the revised manuscript as follows:

"…and the global datasets of the Task Force Hemispheric Transport of Air Pollution (HTAP) (Janssens-Maenhout et al., 2015) …"

32. **Page 12, Line 329: "an important part of introducing an adjoint model" > "an important part of introducing a new modeling component, such as the AERO-BC module"?**

Response: This has been revised as suggested.

33. **Page 13, Lines 345-346: "in an approximately linear way" > "in an approximately linear manner"?**

Response: This has been revised as suggested.

34. **Page 14, Lines 377-378: 6-h integration seems a rather long time. Is it possible that the AERO-BC processes are not very nonlinear?**

Response: Thanks for the insightful comment. Since black carbon (BC) does not participate in complex chemical reactions, the AERO-BC processes in the CMA-GFS-AERO forward model primarily include emission flux, vertical diffusion, coagulation, nucleation, condensation, and dry deposition/sedimentation. As the reviewer correctly points out, these processes are treated as approximately linear in the model and do not exhibit strong nonlinear behavior. Consequently, a 6-hour integration time is reasonable and does not introduce significant errors due to process nonlinearity. Additionally, the assimilation time window of the CMA-GFS-AERO 4D-Var system is 6 hours. Using a 6-hour integration time aligns with this window and enables the evaluation of the tangent linear model (TLM) performance. Our results confirm that the CMA-GFS-AERO TLM demonstrates good performance in the tangent linear approximation of BC.

35. **Page 15, Lines 391-392: I have trouble understanding this one. Which coupled variable? And which physical process variable? Is it also possible that AERO-BC processes are not very nonlinear such that TL approximation is not too much different from the NL one?**

Response: We sincerely apologize for the lack of clarity in this sentence. By "coupled variable", we are referring to BC, and "physical process variable" refers to variables such as specific humidity. What we want to explain is that compared with variables such as potential temperature and specific humidity in the CMA-GFS-AERO model, the tangent linear approximation for BC is quite effective, making it well-suited for constructing a 4D-Var system. We have revised the sentence in the manuscript for better clarity as follows:

"…This phenomenon indicates that, in comparison to variables such as potential temperature and specific humidity in the CMA-GFS-AERO model, the tangent linear approximation for BC is quite effective, making it well-suited for constructing a 4D-Var system."

As the reviewer correctly points out, the AERO-BC processes are treated as approximately linear in the model and do not exhibit strong nonlinear behavior. As a result, TL approximation of the AERO-BC is not too much different from the NL one.

36. **Page 18, Lines 453-457: It will be quite helpful to add more texts to address the links between Fig. 5a and Fig. 6a as these two figures are results from the same single observation experiment with observation placed at the beginning of the window (i.e., 0300 UTC) where Fig. 5a shows the initial analysis increment while Fig. 6a shows the propagated analysis increment valid at the end of the window. Same idea for Fig. 5b and Fig. 6b, while the only difference is the timing of the observation.**

Response: Thanks for the valuable suggestion. We have added more explanations to clarify the links between Fig. 5 and Fig. 6 in the revised manuscript as follows:

"…For the case where the BC observation is placed at 0300 UTC, the initial analysis increment at 0300 UTC (Fig. 5a) exhibits an isotropic structure due to the static B. In contrast, the propagated analysis increment at the end of the assimilation time window (0900 UTC, Fig. 6a) exhibits an anisotropic structure under the influence of the flow-dependent $M_{0\to i}BM_{0\to i}^T$. Similarly, when the BC observation is placed at 0600 UTC, both the initial analysis increment at 0600 UTC (Fig. 5b) and the propagated analysis increment at 0900 UTC (Fig. 6b) exhibit an anisotropic structure. In addition, the horizontal distribution structure of the BC analysis increments in Fig. 6a and Fig. 6b closely resembles that of the analysis increments at the observation time of 0900 UTC (Fig. 5c). This indicates the significant impact of flow-dependent dynamics on the evolution of the analysis increments. No matter what time the observation is placed at, the spatial propagation of the observation information is effectively achieved through the model integration."

37. **Page 19, Lines 467-469: while I think I understand what the authors are trying to say, it is not entirely correct and perhaps not necessary to end the sentence like this. The way the system is setup (i.e., the CMA-GFS-AERO 4DVar system) by minimizing both BC and atmospheric variables together suggests it is a coupled assimilation. I think what the authors are trying to suggest is that the merits of a coupled data assimilation system cannot be fully manifested or exploited by only assimilating a BC observation at the beginning of the window.**

Response: We sincerely thank the reviewer for the valuable comment. We agree with the reviewer's suggestion and have revised the sentence for clarity and accuracy. The original wording was indeed not entirely correct, and we appreciate the clarification regarding the coupled nature of the system. To better reflect the intended meaning, we have rephrased the sentence to emphasize that the full potential of a coupled data assimilation system is not realized by only assimilating a BC observation at the beginning of the assimilation window. The revised sentence now reads:

"…In this case, the merits of a coupled data assimilation system cannot be fully manifested by only assimilating a BC observation at the beginning of the window…"

38. **Page 19, Lines 467-476: I think it is nice to have a paragraph detailing the processes in the 4DVar component of CMA-GFS-AERO that induces non-zero cross-covariance between the atmosphere and BC variables via evolving the initial background covariance with the TL modeling, even though the initial crosscovariance is zero. The current paragraph is trying to do so but remains rather descriptive and lacks interpretation. For that, I suggest checking out Section 2.1 "Coupled data assimilation" of Smith et al. (2015). Smith, P. J., Fowler, A. M., & Lawless, A. S. (2015). Exploring strategies for coupled 4D-Var data assimilation using an idealised atmosphere–ocean model. Tellus A: Dynamic Meteorology and Oceanography, 67(1). https://doi.org/10.3402/tellusa.v67.27025**
**In addition, I do not think "co-correlation" is a proper word.**

Response: Thanks for the insightful comment. We have referred to Section 2.1 "Coupled data assimilation" of Smith et al. (2015) as suggested and have added Section 3.3.5 in the revised manuscript to provide a more detailed interpretation of the mechanisms behind the evolution of the initial background covariance. The updated section reads as follows:

"3.3.5 Flow-dependent background error covariance in CMA-GFS-AERO 4D-Var

In the strongly coupled aerosol-meteorology assimilation system, interactions between the atmospheric variables and BC allow BC observations to influence the analysis increment of atmospheric variables and vice versa. The incremental 4D-Var algorithm implicitly evolves the background error covariances ($\mathbf{B}$) throughout the assimilation window according to the TL model dynamics. This process modifies prior background error variance estimates and induces non-zero correlations between model variables (Smith et al., 2015). By utilizing the fully coupled TLM and ADM in the inner loops of the strongly coupled assimilation system, cross-covariance information between BC and atmospheric variables is generated. This enables observations of one variable to produce analysis increments in the other, leading to more consistent analyses.

Specifically, if the BC observation is assumed to take place at the initial of the assimilation window, the 4D-Var assimilation is equivalent to the 3D-Var assimilation. Since the BC variable is assumed to be uncorrelated with the atmospheric variables in the static $\mathbf{B}$, and there is no direct relationship between the BC observation operator and the atmospheric variables, the BC observation does not lead to the generation of the analysis increments of atmospheric variables. In this case, the merits of a coupled data assimilation system cannot be fully manifested by only assimilating a BC observation at the beginning of the window. If the BC observation is assumed to take place at the middle and the end of the assimilation window, $\mathbf{B}$ evolves within the assimilation time window through the TLM $\mathbf{M}_{0 \to i}$, obtaining the implicit background error covariance matrix $\mathbf{M}_{0 \to i} \mathbf{B} \mathbf{M}_{0 \to i}^{T}$ that evolves with time. $\mathbf{M}_{0 \to i} \mathbf{B} \mathbf{M}_{0 \to i}^{T}$ includes the cross-covariances information of BC and atmospheric variables, and can realize the feedback of the BC observation to the atmospheric variables through the CMA-GFS-AERO ADM $\mathbf{M}_{0 \to i}^{T}$, further producing analysis increments of atmospheric variables."

Additionally, we have revised the description of Figure 7 in the manuscript to make it more concise:

"Figure 7 depicts the analysis increments of temperature at the first model level at the initial time of the assimilation time window (0300 UTC), with the BC observation placed at 0600 and 0900 UTC, respectively. It can be seen that when the BC observation is placed at 0600 and 0900 UTC, positive analysis increments of temperature are generated, with the value of about 0.02 K near the observation location. The mechanism behind the generation of these temperature increments is detailed in Section 3.3.5. This indicates that the temperature of the analysis field will increase due to the assimilation of the BC observation."

Regarding the use of the term "co-correlation," we agree that it is not the most appropriate choice. We have replaced it with "cross-covariance".

39. **Page 19, Line 487: "in fact" should be "in reality" and one can also go on to say "in reality, unlike the single observation experiment, the BC observation is …" to further distinguish the real case from the single observation case.**

Response: According to the reviewer's good instructions, we have revised it as follows:
"In reality, unlike the single observation experiment, the BC observation is …"

40. **Page 20, Lines 507-508: "assimilated all observations within the assimilation time**

window": How frequent is BC observation available for assimilation? I realized that this is actually mentioned in section 3.3.2 that the BC observations are hourly averaged. However, it still didn't say how frequent BC observations are assimilated in the real-case experiments.

Response: Thanks for the insightful comment. In the revised manuscript, we have clarified that in the real-case experiments, the BC observations are assimilated hourly. The revised text now reads as follows:

"…Different from the single observation experiment in Section 5.2, in which the observations are placed at a fixed time, we assimilated all available BC observations with an hourly frequency within the assimilation time window in the full observation experiment…"

41. **Section 4.4: are BC and atmospheric variables minimized together in EXP1 and EXP2 as well? If so, please consider adding a new column in Table 3 to address whether these variables are minimized together or separately. In addition, it might be a good idea to use names that reflects the design of the experiments instead of calling them in numerical order. For example, EXP1 to EXP4 may be renamed to SCDA_BC, SCDA_MET, WCDA_BC+MET, SCDA_BC+MET where SCDA stands for strongly coupled data assimilation while WCDA refers to weakly coupled data assimilation.**

Response: Thanks for the valuable comment. As we mentioned in our previous response to the general comment, our expression in Section 4.4 in the original manuscript was not sufficiently clear, which may have caused confusion. We apologize for any misunderstanding. After carefully considering the reviewer's comments, we have completely rewritten the original Section 4.4, which is now presented as Section 5.3 in the revised manuscript.

In the updated version, Table 3 provides a clear description of the four experiments. Additionally, based on the reviewer's suggestion and our revised text, we have renamed the four experiments as DA_BC, DA_MET, DA_MET_then_BC, and DA_MET_BC_simult. We once again thank the reviewer's valuable comments.

42. **Page 21, Lines 517-519: I am not sure if one can really say so without showing results from EXP2.**

Response: Thanks for pointing this out. Our experimental results show that in the DA_MET experiment, which assimilates operational meteorological observations while excluding BC surface observations, the BC analysis increments are very small. In the revised manuscript, we have added the following clarification:

"…In the following analysis, we primarily compare the BC analysis increments obtained from DA_BC, DA_MET_then_BC, and DA_MET_BC_simult experiments, noting that the BC analysis increments from the DA_MET experiment are very small (figure omitted) …"

Additionally, the wording in Lines 517-519 in the original manuscript has been modified as follows:

"…This indicates that the three BC assimilation strategies have similar assimilation effects on BC, further demonstrating that the assimilation of meteorological observations has a relatively small impact on BC analysis increments…"

43. **Pages 21-22, Lines 539-541: ok, but why? please consider including some interpretation.**

**Are BC and atmospheric variables minimized together in EXP1 but separately in EXP3? It doesn't seem quite straightforward and easy to understand, at least to me, why would assimilating only BC observations in a strongly coupled setup leads to similar impact from assimilating both BC and atmospheric observations in a weakly coupled setup? What could be the mechanism that leads to such a consequence?**

Response: We sincerely appreciate the reviewer's insightful comments. As we mentioned in our previous response, our expression in Section 4.4 in the original manuscript was not sufficiently clear, which may have caused confusion. We apologize for any misunderstanding. After carefully considering the reviewer's comments, we have completely rewritten the original Section 4.4, which is now presented as Section 5.3 in the revised manuscript.

In the updated version, we provide a clear description of the DA_BC, DA_MET, DA_MET_then_BC, and DA_MET_BC_simult experiments. We specifically clarify that the atmospheric variable analysis increments shown in Figure 10 are solely due to BC assimilation, without contributions from the assimilation of atmospheric observations.

We also explain in detail the reasons behind the similarity and differences in the atmospheric variable analysis increments produced by BC assimilation in the DA_MET_then_BC experiment and the DA_BC experiment. The key points are as follows:

"…This is because, although DA_MET_then_BC first assimilates operational meteorological observations and then BC surface observations, the BC assimilation step only incorporates BC observations, just like in DA_BC. Therefore, the analysis increments of atmospheric variables caused by BC observations in both DA_MET_then_BC and DA_BC are similar. Additionally, the values in each sub-image of the middle panel in Fig. 10 differ slightly from those on the left. These differences are attributed to the distinct basic-state values of the atmospheric variables used in the two experiments. In DA_BC, the basic-state values of the atmospheric variables used in the tangent linear and adjoint processes are derived from the atmospheric background field information without assimilating operational meteorological observations, while in DA_MET_then_BC, the basic-state values are based on the atmospheric analysis field information after assimilating the operational meteorological observations…"

We once again appreciate the reviewer's valuable feedback, which has significantly contributed to improving the clarity and completeness of our study.

44. **Page 22, Lines 553-556: I am not sure if one can make this statement by comparing the *differences* of analysis increments between EXP4 and EXP2 with *actual* analysis increments from EXP1 or EXP3. In addition, I am puzzled while trying to understand how the feedback of BC assimilation on atmospheric variables is reduced by having also assimilated atmospheric observations in a coupled setup without actually seeing the analysis increments in EXP2 and EXP4. Some thought processes and reasonings from the authors are definitely required to be stated.**

Response: We sincerely appreciate the reviewer's insightful comments. We once again apologize for the confusion caused by the unclear description in Section 4.4 of the original manuscript. In the revised manuscript, we have clarified that the atmospheric variable analysis increments shown in Figure 10 are solely due to BC assimilation, without contributions from the assimilation of atmospheric observations. Below, we present the reasons why the analysis increments of atmospheric variables are smaller when both

atmospheric and BC observations are assimilated simultaneously:

"…The differences in analysis increments of the four atmospheric variables caused by BC assimilation between DA_MET_BC_simult and DA_BC/DA_MET_then_BC may be due to the fact that information fusion reduces the impact of individual observation. As mentioned above, DA_MET_then_BC is similar to DA_BC in that, in the process of BC assimilation, only BC surface observations are incorporated into the assimilation system. At this stage, the system relies solely on BC observations to correct the initial field. In the absence of atmospheric observations, BC observations play a dominant role, leading to larger analysis increments of atmospheric variables. In contrast, in DA_MET_BC_simult, both operational meteorological observations and BC surface observations are assimilated simultaneously. In this scenario, atmospheric observations provide more comprehensive or reliable information, which may reduce the dominant influence of the BC observations on the analysis increments of atmospheric variables. As a result, a more balanced adjustment of atmospheric variables is achieved in DA_MET_BC_simult…"

We once again appreciate the reviewer's valuable feedback.

45. **Page 22, Lines 556-558: This statement is maybe a little too strong. It sounds like having amplified feedback is not a good thing. Without verifying the analysis with the truth (e.g., re-analysis, or observations that are not assimilated), we do not know if the strongly coupled analysis is actually more accurate than the other ones. Hence, we do not know if amplified feedback is good or not good. Although we'd like to think (or theoretically correct to think) that analysis from a strongly coupled setup is better, we still need some evidence to prove it.**

Response: We sincerely appreciate the reviewer's thoughtful comments. Indeed, the original wording in Lines 556-558 of Section 4.4 was perhaps too strong, and we apologize for that. In the revised manuscript, we have revised this statement to avoid overly strong assertions. The revised statement is as follows:

"…As mentioned above, DA_MET_then_BC is similar to DA_BC in that, in the process of BC assimilation, only BC surface observations are incorporated into the assimilation system. At this stage, the system relies solely on BC observations to correct the initial field. In the absence of atmospheric observations, BC observations play a dominant role, leading to larger analysis increments of atmospheric variables. In contrast, in DA_MET_BC_simult, both operational meteorological observations and BC surface observations are assimilated simultaneously. In this scenario, atmospheric observations provide more comprehensive or reliable information, which may reduce the dominant influence of the BC observations on the analysis increments of atmospheric variables. As a result, a more balanced adjustment of atmospheric variables is achieved in DA_MET_BC_simult…"

We once again appreciate the reviewer's valuable feedback.

46. **Page 23, Line 565: "only 10%": does this mean 10% is not much of an increase? And what is 10% increased computation time relative to? Say, if the microphysics process also takes about 10% computation time, then the readers can have a reference to judge whether 10% is large or small. Without any context, it is just a number.**

Response: Thanks for the valuable suggestion. We have clarified this point in the revised

manuscript as follows:

"the CMA-GFS-AERO simulations increase only about 10% of the computational time of the CMA-GFS simulations (As a reference, the microphysics process accounts for approximately 5% of the total computation time in CMA-GFS simulations) …"

**47. Page 24, Lines 591-592: "three component models" > "three model components"**
Response: This has been revised as suggested.

**48. Figure 2: I believe the x-axis is missing a base 10 and a minus sign in the power of 10.**
Response: Thanks for pointing this out. We have revised Figure 2 accordingly and ensured that the x-axis now includes the correct base 10 and minus sign in the power of 10.

**49. Figures S2-S3 and almost all figures: figure captions are rather vague and not very helpful. Both Figs. S2 and S3 present rather complicated ideas and deserve a clearer and informative description.**
Response: Thanks for the comment. Based on the feedback from Comments #4, #16, and #17, we have removed Figures S2 and S3 in the revised manuscript. Additionally, we have reviewed all figure captions and revised those that were vague to ensure greater clarity and informativeness.

**50. Figure 9: When are these analysis increments valid at? beginning, middle, or the end of the window?**
Response: Thanks for the comment. We would like to clarify that these analysis increments are valid at the beginning of the window.

**References**

Bergman, K. H.: Multivariate Analysis of Temperatures and Winds Using Optimum Interpolation, Mon. Wea. Rev., 107, 1423-1444, https://doi.org/10.1175/1520-0493(1979)107<1423:MAOTAW>2.0.CO;2, 1979.

Bocquet, M., Elbern, H., Eskes, H., Hirtl, M., Žabkar, R., Carmichael, G. R., Flemming, J., Inness, A., Pagowski, M., Pérez Camaño, J. L., Saide, P. E., San Jose, R., Sofiev, M., Vira, J., Baklanov, A., Carnevale, C., Grell, G., and Seigneur, C.: Data assimilation in atmospheric chemistry models: current status and future prospects for coupled chemistry meteorology models, Atmos. Chem. Phys., 15, 5325-5358, https://doi.org/10.5194/acp-15-5325-2015, 2015.

Pagowski, M., Grell, G. A.: Experiments with the assimilation of fine aerosols using an Ensemble Kalman Filter, J. Geophys. Res., 117, D21302, doi:10.1029/2012JD018333, 2012.

Zhang, L., Liu, Y., Liu, Y., Gong, J., Lu, H., Jin, Z., Tian, W., Liu, G., Zhou, B., Zhao, B.: The operational global four‐dimensional variational data assimilation system at the China Meteorological Administration, Q. J. Roy. Meteor. Soc., 145, 1882-1896, https://doi.org/10.1002/qj.3533, 2019.

Zhu, S., Wang, B., Zhang, L., Liu, J., Liu, Y., Gong, J., Xu, S., Wang, Y., Huang, W., Liu, L., He, Y., Wu, X., Zhao, B., Chen, F.: A 4DEnVar‐based ensemble four‐dimensional variational (En4DVar) hybrid data assimilation system for global NWP: system description and primary

tests., J. Adv. Model. Earth Sy., 14(8), https://doi.org/10.1029/2022MS003023, e2022MS003023, 2022.

Zou, X., Vandenberghe, F., Pondeca, M. and Kuo, Y.-H. Introduction to adjoint techniques and the MM5 adjoint modeling system (No. NCAR/TN-435 þ STR). University Corporation for Atmospheric Research. 1997.

---

## Author Comment (AC5)

**Response to Reviewer #2 for Geoscientific Model Development:**
**Manuscript gmd-2024-148**
**By Liu et al.**

We sincerely thank Reviewer #2 for thoughtful and constructive feedback. We have carefully considered each comment and made every effort to implement all the suggested changes. The notes below address each comment in detail. Please note that Reviewer's comments are shown in bold type and our responses in plain type.

Reviewer #2

**Summary & General Comments**
**The article presents an overview of the development of a strongly coupled 4D-Var assimilation system where an aerosol atmospheric component, the total mass concentration of black carbon (BC), is added to the 4D-Var control vector. The article contains a detailed explanation of how the necessary linear models have been developed, by extracting and recoding the BC-related aerosols physical modelling codes and by formulating a specific B-matrix model (and control-vector conversion) for the BC mass concentration. Rather technical validation results are displayed showing the correctness of the TL and AD models, along with preliminary experimental results. The article finishes with an outlook mostly referring to a Part II where the authors intend to discuss comprehensive experimental results on the impact of assimilating BC in a strongly coupled formulation, on the other atmospheric fields (wind, temperature, pressure, humidity).**
Response: We sincerely appreciate the reviewer's thorough summary and insightful comments on our study. The reviewer has correctly captured the key aspects of our work, including the development of the strongly coupled 4D-Var assimilation system with BC added to the control vector, the extraction and recoding of BC-related aerosol processes, and the construction of the background error covariance model for BC mass concentration. Furthermore, we appreciate the reviewer's recognition of the validation of the TL and AD models, as well as the preliminary experimental results presented in this work. As noted, we plan to present a more comprehensive analysis of the impact of BC assimilation on meteorological fields in Part II. Thanks again for the valuable feedback.

**This article Part I is overall clearly structured, with each section well introduced. As stated by the authors, the aim of the paper is to present the methodology without entering into a complete, comprehensive evaluation of experimental results in full, long-period 4D-Var assimilation experiments. Taking into account that strongly coupled atmosphere-aerosol chemistry assimilation systems have been very little presented so far in open literature (to the reviewer's knowledge), the authors' choice to propose such an introductory Part I can be supported. Nevertheless, the paper focuses too strongly on technical sanity checks (such as the results of tests of TL and AD models which are standard and well-known tests when developing variational codes) which for themselves bring no innovative information.**
**Conversely, the paper lacks explanations on specific scientific challenges that would strengthen the scientific interest of the paper:**

Response: We sincerely appreciate the reviewer's constructive feedback and recognition of the clear structure and methodological focus of this Part I paper. Indeed, as the reviewer correctly pointed out, our intention is to present the methodology and necessary technical developments rather than providing a full, long-period evaluation of 4D-Var assimilation experiments, which will be the focus of Part II.

Regarding the reviewer's concern about the strong emphasis on technical sanity checks, we would like to clarify that while TL and AD model verification is indeed a standard step in variational data assimilation, it is particularly crucial in our study due to the complexity of incorporating aerosol-related processes into a strongly coupled system. Ensuring the correctness of the TL and AD models is essential for establishing a solid foundation for the strongly coupled 4D-Var assimilation system.

We acknowledge the reviewer's suggestion to provide a more detailed explanation of the specific scientific challenges. We have carefully considered the reviewer's concerns regarding Section 4.4 in the original manuscript, as well as the comments from the other two reviewers. Based on these valuable suggestions, we have thoroughly revised this section, which is now presented as Section 5.3 in the revised manuscript.

In the updated version, we have clearly introduced the objective of the four experiments, which is to investigate the impact of different BC assimilation strategies on both BC and atmospheric variables. We have renamed the four experiments as DA_BC, DA_MET, DA_MET_then_BC, and DA_MET_BC_simult. The revised Table 3 now provides a clear description of the four experiments. We have also compared the BC analysis increments obtained from the DA_BC, DA_MET_then_BC, and DA_MET_BC_simult experiments, noting that the BC analysis increments from the DA_MET experiment are very small. Additionally, we compare the atmospheric analysis increments caused by BC assimilation in DA_BC, DA_MET_then_BC (DA_MET_then_BC - DA_MET), and DA_MET_BC_simult (DA_MET_BC_simult - DA_MET).

Our main conclusions from this analysis are as follows: The preliminary results obtained from this set of four experiments indicate that different BC assimilation strategies have little impact on BC analysis increments but significantly affect the analysis increments of atmospheric variables. When only BC observations are assimilated, the influence of BC on atmospheric variables is more pronounced, whereas the simultaneous assimilation of meteorological observations moderates this influence. This suggests that in BC assimilation, meteorological observations can help constrain the uncertainty introduced by BC observations on atmospheric variables, thereby improving the reliability of the assimilation results. Moreover, these results demonstrate the successful implementation of the newly developed CMA-GFS-AERO 4D-Var system and highlight it as an effective approach for investigating the feedback of BC data assimilation on meteorological forecasts.

In the future, we will conduct batch experiments using CMA-GFS-AERO 4D-Var to gain deeper insights into the role of BC assimilation in numerical weather prediction and further refine the system for broader applications.

Additionally, in response to another reviewer's suggestion, we have adjusted the radius of influence for BC observations to 2 km, 10 km, and 20 km for urban, rural, and remote stations, respectively, according to Elbern et al. (2007). Consequently, all experiments in Section 5.3 have been redone using the updated radii, and the corresponding figures and text have been revised

accordingly to reflect the new results.

For more details on the analysis, please refer to Section 5.3 of the revised manuscript. We once again appreciate the reviewer's valuable comments.

**1. Compared to the BC physics available in the original CUACE codes, how much has the BC physics for the CMA-GFS-AERO codes been adapted in terms of the representation of the physical processes, such as transport, chemical transformation and the interaction with radiative processes ? Taking this comment one step further, has there been any kind of simplification made when developing the BC physics modules for the linear models, any step of regularization of a non-linear formulation, or any omission of specific complex processes whose linearization was felt too difficult (at least for this v1.0 of the system) ? More explicit explanations should be provided, likely in Sections 3.1 and 3.2.**

Response: Thanks for the insightful comment. The AERO-BC module was created by extracting BC-related codes from the CUACE model, with its functionality aligning with the BC aerosol processes in the CAM module of CUACE. In other words, the physical processes for BC in AERO-BC are identical to those in the CAM module, with no changes made. The main differences lie in the engineering aspect: (1) while the CAM module was originally written in Fortran 77, the AERO-BC code has been rewritten in Fortran 90; (2) since CAM in CUACE deals with six types of aerosols, the code structure is somewhat complex and redundant, whereas AERO-BC focuses solely on BC, resulting in a simpler and more streamlined structure. These updates improve code readability and enhance computational efficiency, without affecting the underlying physical processes.

In constructing the tangent linear (TL) and the adjoint of AERO-BC, no simplifications were made to the AERO-BC processes. Specifically, no regularization was applied to the nonlinear equations, nor were any complex processes, which were difficult to linearize, omitted. As a result, the TL and the adjoint of AERO-BC fully include all processes related to emission flux, vertical diffusion, and aerosol physical processes (e.g. hygroscopic growth, coagulation, nucleation, condensation, dry deposition/sedimentation, and below-cloud scavenging).

Additionally, we have added these clarifications in Sections 3.1 and 3.2 of the revised manuscript.

**2. More explanation of why the strongly coupled case provides significantly different results on the analysis fields of the "traditional" atmospheric fields, compared with no BC assimilation or with the weakly coupled case, is missing in Section 4.4. Two striking results are displayed but eventually with very little physical interpretation while both seem to be systematic results:**

Response: We sincerely appreciate the reviewer's insightful comment. In the original manuscript, our expression in Section 4.4 was not sufficiently clear, which may have caused confusion. We apologize for any misunderstanding. After carefully considering the reviewer's comments, we have completely rewritten this section, which is now presented as Section 5.3 in the revised manuscript. In the updated version, we have added the necessary explanations to improve clarity and address the reviewer's concerns.

**a. Adding BC in the modelling and assimilation system rather than omitting this component induces a positive analysis increment on temperature. So question here: should one**

understand that adding BC in the forecast trajectory *anyway* will slightly increase temperature via the radiative effect of absorption? Is this effect then very systematic ? Is it local or even global ?

Response: Thanks for the insightful comment. Currently, the radiative effect of BC on atmospheric temperature is not yet considered in the CMA-GFS-AERO forward model, and this will be a key area of focus for our future work. In the revised manuscript, Section 3.3.5 introduces how CMA-GFS-AERO 4D-Var incorporates cross-covariances between BC and atmospheric variables through the background error covariance matrix. The adjoint model then propagates the impact of BC observations onto atmospheric variables, leading to corresponding analysis increments. Additionally, Section 5.2 presents single-observation experiments that systematically analyze the generation of temperature analysis increments at observation times when BC observations are assimilated at the initial, middle, and end of the assimilation time window. These results align with the theoretical framework described in Section 3.3.5. Therefore, in the present version of the CMA-GFS-AERO 4D-Var system, the positive analysis increment on temperature is primarily due to the assimilation of BC observations in the 4D-Var system, rather than the radiative heating effect of BC.

**b. Why precisely is strongly coupled assimilation of BC causing an overall decrease of the amplitude of the analysis increments by an order of magnitude ? What are the damping retro-actions ?**

Response: We sincerely appreciate the reviewer's insightful comment. As we mentioned in our previous response, after carefully considering the reviewer's comments, we have completely rewritten this section. In this revision, we have refined the explanation of the experiments and their results to improve clarity. Please refer to Section 5.3 in the revised manuscript for the updated expression. Below, we present the reasons why the analysis increments of atmospheric variables are smaller when both atmospheric and BC observations are assimilated simultaneously:

"…The differences in analysis increments of the four atmospheric variables caused by BC assimilation between DA_MET_BC_simult and DA_BC/DA_MET_then_BC may be due to the fact that information fusion reduces the impact of individual observation. As mentioned above, DA_MET_then_BC is similar to DA_BC in that, in the process of BC assimilation, only BC surface observations are incorporated into the assimilation system. At this stage, the system relies solely on BC observations to correct the initial field. In the absence of atmospheric observations, BC observations play a dominant role, leading to larger analysis increments of atmospheric variables. In contrast, in DA_MET_BC_simult, both operational meteorological observations and BC surface observations are assimilated simultaneously. In this scenario, atmospheric observations provide more comprehensive or reliable information, which may reduce the dominant influence of the BC observations on the analysis increments of atmospheric variables. As a result, a more balanced adjustment of atmospheric variables is achieved in DA_MET_BC_simult…"

We once again appreciate the reviewer's valuable comments.

**The article is fairly clearly written though some specific checking of English phrase construction could be worthwhile. In the specific comments below, a few particularly unclear phrasings are stressed, which deserve further attention and rewrite by the authors. In the**

bibliographical section, 5 references relate to documents in Chinese. It is unclear to the reviewer what GMD's policy about references in languages other than English is. It might be appropriate that the authors confirm that they can commit to make available translated texts, should they be asked by future readers.

Response: Thanks for the valuable feedback. We appreciate your comments on the clarity of the article, and we have reviewed the phrasing as suggested to improve the overall readability. Regarding the specific comments below, a few unclear phrasings have been revised as suggested. Concerning the references in Chinese, we understand the importance of ensuring accessibility for international readers. We confirm that, should future readers request, we will make efforts to provide translated texts for the Chinese references cited in the manuscript.

In conclusion, my recommendation is to accept the paper, as a Part I component to be complemented by a Part II, after revision. The goal of the revision, following the comments above, should be to strengthen the scientific explanations of the implementation of BC in the 4D-Var framework as well as to strengthen the physical interpretation of the experimental results displayed in Section 4. A further recommendation could be to extend the paper's title from "System description" to "System description and preliminary experimental results".

Response: We sincerely appreciate the reviewer's positive recommendation and constructive suggestions. We are grateful for the recognition of our work as a Part I study and fully acknowledge the importance of strengthening both the scientific explanations of BC implementation in the 4D-Var framework and the physical interpretation of the experimental results in Section 4 (now presented as Section 5 in the revised manuscript).

Following the reviewer's comments, we have carefully revised the manuscript to enhance these aspects. Regarding the suggestion to extend the paper's title, we acknowledge its potential benefits in better reflecting the content of the study. After careful consideration, we have updated the title from "System description" to "System description and preliminary experimental results" in the revised manuscript.

We sincerely appreciate the reviewer's valuable feedback, which has significantly contributed to improving the clarity and completeness of our study.

**Specific Comments & Typos**
**Section 1.**
**line 69: what does "PM" stand for ?**
Response: Thanks for pointing this out. "PM" stands for "particulate matter". $PM_{2.5}$, also known as fine particulate matter, refers to the particulate matter with an aerodynamic diameter of 2.5 micrometers or less.

We have added an explanation of $PM_{2.5}$ in the revised manuscript as follows:

"Black carbon (BC) aerosol, a major component of the fine particulate matter ($PM_{2.5}$) defined by an aerodynamic diameter of 2.5 micrometers or less, primarily originates from the incomplete combustion of biomass and fossil fuels (Kuhlbusch, 1998)…"

**Section 2. None.**
Response: We sincerely appreciate the reviewer's feedback and patience.

**Section 3.**

**line 169: "The transport processes for $\psi_{bc}$ are the same as _those_ for the variables _associated with the different water species_ …"**

**Re-phrase "water-matter" everywhere in the paper (not sure this is a good English wording, though it is understandable)**

Response: According to the reviewer's good instructions, we have changed "that" to "those" in this sentence, and "water-matter" has also been revised to "variables associated with the different water species" throughout the manuscript.

**lines 175-179 (end of section 3.1):**

**1. the whole text should be re-written, splitting it into two separate sentences.**

Response: According to the reviewer's good instructions, we have re-written the text in the revised manuscript as follows:

"Besides, according to the vertical distribution of BC in the MERRA-2 (Modern-Era Retrospective analysis for Research and Applications, Version 2) reanalysis data (https://daac.gsfc.nasa.gov), we observed that the BC mass mixing ratio decreases rapidly after entering the stratosphere, reaching values of about $10^{-12}$ kg/kg. This is 2-3 orders of magnitude smaller compared to the surface. To improve computational efficiency and balance memory usage with the effectiveness of BC forecasting, we set the height of $\psi_{bc}^n$ in the CMA-GFS-AERO model to 65 levels (approximately 30 hPa), which corresponds to the middle layer of the stratosphere…."

**2. An additional explanation of how the absence of BC above model level 65 is dealt with in the models should be added. What happens regarding vertical transport for instance ? What's the impact in the adjoint code ?**

Response: Thanks for the insightful comment. We have added an explanation regarding the treatment of BC above model level 65. We assume that BC concentrations above this level are negligible, given their small magnitude in the stratosphere. For vertical transport, this approximation does not have a significant impact on the model. In the adjoint code, BC concentrations above model level 65 are also treated as negligible, and this does not significantly affect the adjoint calculations.

We have added the explanation in the revised manuscript as follows:

"…Regarding the absence of BC above model level 65, we handled vertical transport by assuming that any BC concentrations above this level are negligible. This approximation does not significantly affect the model's performance, as the BC mass mixing ratio is very small in the upper layers. Correspondingly, in the adjoint code, BC concentrations above model level 65 are also treated as negligible, and this does not significantly affect the adjoint calculations."

**lines 231-234: "Firstly …" and later "Secondly …" => reformulated these two sentences such that there is a verb. Perhaps, try with "Firstly, the distribution weights … are calculated.". The same construction would apply to the next sentence.**

Response: Thanks for the insightful comment. Following the recommendation, we have revised the two sentences in the revised manuscript as follows:

"…Firstly, the distribution weights ($\omega^n$) of each size bin of $\psi_{bc}^n$ in the background field are

calculated …Secondly, the analysis increment of $\psi_{bc}^n$ ($\delta\psi_{bc}^n$) is calculated based on the analysis increment of $C_{bc}$ ($\delta C_{bc}$), following the equation…"

**lines 274-277: reformulate that sentence (much too long). Make two separate ones.**

Response: Thanks for pointing this out. We have revised the sentence into two separate ones in the revised manuscript as follows:

"…If the observation height is less than the height of the first model layer and the difference between the two heights is less than 300 meters, the BC concentration at the first model layer is regarded as the equivalent BC observation. However, if the difference between the two heights is greater than or equal to 300 meters, the data from that site is discarded."

**line 282: I don't think that a sentence should start abruptly by "And …". Simply remove this word with no loss of clarity of the text ?**

Response: Thanks for pointing this out. We agree that a sentence should not start abruptly by "And …". Following the recommendation, we have removed this word in the revised manuscript without affecting the clarity of the text.

**lines 286-288:**
**1. The vertical correlation model of the background error _is_ expressed as …**

Response: This has been revised as suggested.

**2. Some additional explanation of how this formula has been obtained is required (by analogy to the water species case ? by specific experimental trials ? from external works and then add a reference ?)**

Response: Thanks for the comment. We appreciate the suggestion for additional clarification regarding the derivation of the vertical correlation model for background error. This formula was derived through a combination of theoretical considerations (Bergman, 1979) and experimental tuning, with particular reference to the methodology used for humidity in the CMA-GFS 4D-Var system.

In the revised manuscript, we have added the following clarification:

"…The vertical correlation model of the background error is derived through a combination of theoretical considerations (Bergman, 1979) and experimental tuning, with particular reference to the methodology used for humidity in the CMA-GFS 4D-Var system. It is expressed as…"

**Section 4.**
**lines 315-316: link the two sentences together and remove the start with "And …" (just use "... and …")**

Response: Thanks for pointing this out. Following the recommendation, we have revised the sentence to remove the "And ..." at the beginning. The revised sentence now reads as follows:

"…The first 9 days were used as the spin-up time to establish a realistic BC distribution…"

**line 322: again, what does "PM" stand for ?**

Response: Thanks for pointing this out. "PM" stands for "particulate matter". For the definition of $PM_{2.5}$, please refer to our previous response. Additionally, $PM_{10}$, also known as inhalable particulate

matter, refers to the particulate matter with an aerodynamic diameter of 10 micrometers or less. We have added an explanation of $PM_{10}$ in the revised manuscript as follows:

"…and particulates (OC, BC, $PM_{2.5}$ and $PM_{10}$), where $PM_{10}$ refers to the inhalable particulate matter with an aerodynamic diameter of 10 micrometers or less. These data were processed into grid-point emission data applicable to the CUACE model through the EMIPS emission source processing system…"

**lines 364-365: replace "filed" by "_field_" . Note that the same typo appears several times later in the text, so the simplest is to make a systematic search and replace.**

Response: We apologize for the misspelling of the word "field", and this has been revised throughout the manuscript as suggested.

**lines 390-391: The sentence "This phenomenon indicates that …" definitely _requires a complete reformulation_. It is currently simply not understandable ! What do you want to explain ?**

Response: We sincerely apologize for the lack of clarity in this sentence. What we want to explain is that compared with variables such as potential temperature and specific humidity in the CMA-GFS-AERO model, the tangent linear approximation for BC is quite effective, making it well-suited for constructing a 4D-Var system. We have revised it in the manuscript as follows:

"…This phenomenon indicates that, in comparison to variables such as potential temperature and specific humidity in the CMA-GFS-AERO model, the tangent linear approximation for BC is quite effective, making it well-suited for constructing a 4D-Var system."

**line 395: the caption of figure 4 mentions "simple physics" => what does "simple physics" refer to ? Do the authors refer to specific simplified physics involved in the 4D-Var models (TLM, ADM) ? If this is the case, then more explanations should be provided earlier in the text, for instance in section 3.2 (and also check my general comment above)**

Response: We sincerely apologize for the confusion caused by the phrase "simplified physics" in the caption of Figure 4. This was an incorrect expression, and we have removed it in the revised manuscript. Additionally, we would like to clarify that in constructing the tangent linear and adjoint of AERO-BC, the physical processes in AERO-BC were not simplified. For further details, please refer to our response to the general comment above.

**line 406: Remind explicitly that the time step is 300s as it's of interest here for the reader to promptly be able to convert time steps into a forecast time length**

Response: We sincerely appreciate the reviewer's valuable suggestion. In the revised manuscript, we have explicitly clarified that the time step is 900 s (300 s in the outer loop and 900 s in the inner loop). The revised text is as follows:

"…Following Eq. (16), we conducted five experiments with the integration time equal to 1, 6, 12, 24, and 36 steps with the time step of 900 s…"

**line 460 and Figure 6. Is the propagation of the BC increment by the wind the generally dominant effect ? Is this what the authors quite generally have been observing in their results ? Or conversely does this statement only apply to the very simplified context of the**

single-point observation experiments ? (this is what I would derive from the results later in section 4.4 when the effect of full observations strongly-coupled assimilation is shown). Nevertheless, an additional sentence here could be clarifying, in order to avoid misinterpretation with other results shown later on.

Response: Thanks for the insightful comment. Regarding line 460 and Figure 6 in the original manuscript, in the context of the single-point observation experiments, the propagation of BC increments is indeed primarily dominated by advection due to the limited observational constraint. When more comprehensive observations are assimilated, advection remains a key factor in BC increment propagation. However, its dominance is less pronounced compared to the single-point experiment, as other processes, such as vertical mixing and deposition, also contribute to BC distribution adjustments.

To clarify this, we have added the following sentence at the end of the discussion of Figure 6:

"…In this idealized single-point observation experiment, the propagation of BC increments is primarily dominated by advection due to the limited observational constraint. When more comprehensive observations are assimilated, advection remains a key factor, but its dominance is less pronounced as other processes also influence the adjustment of BC distributions (see Section 5.3)."

**line 470. "Figure 7 depicts … _at the initial time of_ the assimilation window …"**

Response: Thanks for pointing this out. We have revised it as suggested.

**lines 516-519: should be totally re-written as they are not clear at present. A proposal : "These results suggest that the assimilation of meteorological observations has a small impact on the BC analysis increments. Furthermore, weakly and strongly coupled assimilation seem to lead to similar BC analysis increments."**

Response: We sincerely appreciate the reviewer's suggestion. In the revised manuscript, we have completely rewritten this section (now presented as Section 5.3 in the revised manuscript), and the content in lines 516-519 has been updated accordingly. The new version provides a clearer and more precise explanation.

**line 533: "there are certain degrees of analysis increments …" => this wording is very obscure, _please reformulate_.**

Response: Thanks for the suggestion. We have revised the sentence to improve clarity. The updated wording is as follows:

"When only BC surface observations are assimilated (DA_BC), analysis increments of temperature (Fig. 10a), pressure (Fig. 10d), east-west component of horizontal wind (Fig. 10g), and relative humidity (Fig. 10j) are present in North China and Eastern China…"

**lines 552-558: only to mention that this is the part of section 4.4 that explicitly describes what seem to be interesting physics-related results of assimilating BC, already in these preliminary experiments. This is the part where more physical interpretation of these results is expected, on the feedback mechanisms in strongly-coupled assimilation and about the warm bias on the temperature analysis increment. (Refer to my general comments)**

Response: Thanks for the insightful comment. As we mentioned in our response to the general

comments, we have carefully considered the reviewer's concerns regarding Section 4.4 of the original manuscript and have thoroughly rewritten it in Section 5.3 of the revised manuscript.

Regarding the feedback mechanisms in strongly coupled assimilation, we have also added discussions in Section 3.3.5 of the revised manuscript. Specifically, CMA-GFS-AERO 4D-Var incorporates cross-covariances between BC and atmospheric variables through the background error covariance matrix. The adjoint model then propagates the impact of BC observations onto atmospheric variables, leading to corresponding analysis increments.

Additionally, regarding the warm bias on the temperature analysis increment, Section 5.2 presents single-observation experiments that systematically analyze the generation of temperature analysis increments at different observation times when BC observations are assimilated at the initial, middle, and end of the assimilation time window. These results align with the theoretical framework described in Section 3.3.5.

We appreciate the reviewer's valuable suggestions, which have helped improve the clarity and depth of our analysis.

**Section 4.5.**

**Only to mention that I am supportive of this section explaining the computational figures of the enhanced 4D-Var system.**

Response: We sincerely thank the reviewer for the supportive comment on this section. This section aims to provide a clear description of the computational performance of the CMA-GFS-AERO 4D-Var system, highlighting its high efficiency and good scalability. We are glad that the reviewer finds this explanation satisfactory.

**Section 5.**

**line 614: "surface BC observation_s_"**

Response: This has been revised as suggested.

**line 615: "through batch test_s_"**

Response: This has been revised as suggested.

**Acknowledgments.**

**line 633: "The development of _the_ CMA-GFS-AERO 4D-Var system is a systematic project"**
**=> what do you mean by "systematic project" ?**

Response: Thanks for pointing this out. By "systematic project", we mean that the development of the CMA-GFS-AERO 4D-Var system involved the collaboration of many colleagues from various disciplines and teams. To make this clearer, we have revised the sentence as follows:

"…The development of the CMA-GFS-AERO 4D-Var system is a collaborative effort involving contributions from many colleagues. We sincerely thank the entire team for their cooperation…"

**References**

Bergman, K. H.: Multivariate Analysis of Temperatures and Winds Using Optimum Interpolation, Mon. Wea. Rev., 107, 1423-1444, https://doi.org/10.1175/1520-0493(1979)107<1423:MAOTAW>2.0.CO;2, 1979.

---

## Author Comment (AC7)

**Response to Reviewer #3 for Geoscientific Model Development:**
**Manuscript gmd-2024-148**
**By Liu et al.**

We sincerely thank Reviewer #3 for thoughtful and constructive feedback. We have carefully considered each comment and made every effort to implement all the suggested changes. The notes below address each comment in detail. Please note that Reviewer's comments are shown in bold type and our responses in plain type.

**Reviewer #3**

**This work develops a coupled chemistry-meteorology data assimilation (DA) system for the China Meteorological Administration (CMA). It proposes a coupled chemistry meteorology model and adds black carbon (BC) mass concentration to the operational CMA-GFS 4D-Var system as a control variable. This work (part 1) focuses on system description, validation of tangent linear and adjoint models, and investigation of the analysis increment.**

Response: We sincerely appreciate the reviewer's thorough summary and insightful comments on our study. The reviewer has correctly captured the key aspects of our work. As stated, this work (Part 1) primarily focuses on the system description, validation of the tangent linear and adjoint models, and investigation of the analysis increment. In subsequent work, we plan to present a more comprehensive analysis of the impact of BC assimilation on BC and meteorological fields in Part II. Thanks again for the valuable feedback.

**General comments:**

- **The authors generally clearly describe the DA system. However, some important information is discussed in the result section, and some reordering may be needed (see specific comments below).**

  Response: We thank the reviewer for the constructive feedback. We appreciate the suggestion regarding the reordering of the content. In response, we have moved the relevant information from the results section to the section describing the DA system, as recommended. For details, please refer to the response to the issue regarding Lines 464-476. This restructuring allows for a clearer presentation of the methodology and ensures that the information is logically organized. We believe this improves the flow of the manuscript and enhances the clarity of the DA system description.

- **The authors compare the differences in analysis increments between uncoupled, weakly coupled and strongly coupled DA experiments in Section 4.4. The reviewer has some questions about the results, particularly on Figure 10 (see below).**

  Response: Thanks for the comments and interest in our results. We acknowledge that the original expression in Section 4.4 may not have been sufficiently clear, and we appreciate the opportunity to improve it. After carefully considering the reviewer's feedback, along with comments from the other two reviewers, we have completely rewritten this section, which is now presented as Section 5.3 in the revised manuscript.

  In the updated version, we have clearly introduced the objective of the four experiments,

which is to investigate the impact of different BC assimilation strategies on both BC and atmospheric variables. We have renamed the four experiments as DA_BC, DA_MET, DA_MET_then_BC, and DA_MET_BC_simult. The revised Table 3 now provides a clear description of the four experiments. We have also compared the BC analysis increments obtained from the DA_BC, DA_MET_then_BC, and DA_MET_BC_simult experiments, noting that the BC analysis increments from the DA_MET experiment are very small. Additionally, we compare the atmospheric analysis increments caused by BC assimilation in DA_BC, DA_MET_then_BC (DA_MET_then_BC - DA_MET), and DA_MET_BC_simult (DA_MET_BC_simult - DA_MET).

Our main conclusions from this analysis are as follows: The preliminary results obtained from this set of four experiments indicate that different BC assimilation strategies have little impact on BC analysis increments but significantly affect the analysis increments of atmospheric variables. When only BC observations are assimilated, the influence of BC on atmospheric variables is more pronounced, whereas the simultaneous assimilation of meteorological observations moderates this influence. This suggests that in BC assimilation, meteorological observations can help constrain the uncertainty introduced by BC observations on atmospheric variables, thereby improving the reliability of the assimilation results. Moreover, these results demonstrate the successful implementation of the newly developed CMA-GFS-AERO 4D-Var system and highlight it as an effective approach for investigating the feedback of BC data assimilation on meteorological forecasts.

In the future, we will conduct batch experiments using CMA-GFS-AERO 4D-Var to gain deeper insights into the role of BC assimilation in numerical weather prediction and further refine the system for broader applications.

Additionally, in response to another reviewer's suggestion, we have adjusted the radius of influence for BC observations to 2 km, 10 km, and 20 km for urban, rural, and remote stations, respectively, according to Elbern et al. (2007). Consequently, all experiments in Section 5.3 have been redone using the updated radii, and the corresponding figures and text have been revised accordingly to reflect the new results.

For more details on the analysis, especially the discussion related to Figure 10, please refer to Section 5.3 of the revised manuscript. We once again appreciate the reviewer's valuable suggestions.

**Specific comments:**

- **Line 137: The reviewer is not sure whether "inverse integration" is a proper expression for the adjoint model.**

  Response: Thanks for pointing this out. We have revised the sentence to use "backward integration", which better reflects the role of the adjoint model in integrating backward in time.

- **Line 230-235: The authors provide a pragmatic way to convert the BC mass concentration ($C_{bc}$) to the mass mixing ratio of BC ($\psi_{bc}$). The reviewer has two questions regarding this:**
  - **Are the weights ($\omega^n$) calculated locally?**

Response: Thanks for the insightful comment. The distribution weights ($\omega^n$) are not calculated locally; instead, they are determined based on the entire three-dimensional background field. This approach ensures that the weight distribution reflects the global characteristics, rather than being influenced by local variations. By doing so, a global weighting factor is provided, which allows for a reasonable allocation of analysis increments.

Additionally, we have added the following explanation in the revised manuscript: "…Firstly, the distribution weights ($\omega^n$) of each size bin of $\psi_{bc}^n$ in the background field are calculated based on the entire three-dimensional domain, following the equation $\omega^n = \frac{\Sigma_1^N \psi_{bc}{}^n}{\Sigma_{n=1}^6 \left( \Sigma_1^N \psi_{bc}{}^n \right)}$, where $N$ represents the number of three-dimensional grid points…"

- **In reality, could the change in Cbc be mainly caused by ψbc for one size bin (one of the six variables)? For example, the observation of the BC surface concentration is larger than the background, and it is mainly due to ψbc of one size bin. However, the DA system will assign the analysis increments to ψbc of all size bins.**

  Response: We appreciate the reviewer's concern. In reality, the change in $C_{bc}$ may indeed be primarily driven by the variation of $\psi_{bc}^n$ in a specific size bin. While the use of distribution weights ($\omega^n$) ensures that the analysis increment is proportionally assigned across all size bins. For instance, if the change in $C_{bc}$ is mainly due to the second size bin of $\psi_{bc}^n$, then the weight $\omega^n$ for this bin will be relatively larger, leading to a greater proportion of the analysis increment being assigned to it. This approach maintains consistency while still reflecting the dominant contributors to $C_{bc}$ variations.

- **Line 253: The forward model was used to refer to the forecast model. It should be clear here whether this is an observation forward model (observation operator) or a forecast forward model.**

  Response: Thanks for the insightful comment. To clarify, the "forward model" in this context refers specifically to the forecast forward model. Following the recommendation, we have revised it as follows:

  "…Representativeness errors reflect the inaccuracies in the forecast forward model…"

- **Line 298: It would be a good idea to add some references to show the use of the SOAR function in operational data assimilation, e.g., Ballard et al., (2016). https://doi.org/10.1002/qj.2665**

  Response: Thanks for the valuable suggestion. We have added a reference to Ballard et al. (2016) to highlight the use of the SOAR function in operational data assimilation systems. We appreciate the recommendation, as it strengthens the context and support for this statement. The revised sentence now reads as follows:

  "…The horizontal correlation of the background error for the control variable BC is calculated by the second-order auto-regressive (SOAR) correlation function, which is

commonly used in operational data assimilation systems (Ballard et al., 2016), expressed as…"

- **Line 312-313: Information is described repeatedly.**

  Response: Thanks for pointing this out. We have simplified the description in Line 312-313 to avoid duplicating the detailed information already provided in Section 2.2. The revised sentence now reads as follows:

  "…Referring to the running scheme of the CMA-GFS 4D-Var system described in Section 2.2, the CMA-GFS-AERO 4D-Var system adopts the same 6-h cycling schedule and assimilation windows…"

- **Line 316: How about the iteration number in the outer loop?**

  Response: Thanks for pointing this out. In our configuration, the outer loop is performed only once. This is consistent with the operational setup of the CMA-GFS 4D-Var system and has been found sufficient to achieve convergence within the experimental framework of this study. We have updated the text to include this information for clarity in the revised manuscript as follows:

  "…The maximum minimization iteration number in the inner loop was set to 50, while the outer loop was performed only once. This setting is consistent with the operational configuration of the CMA-GFS 4D-Var system and has been found sufficient for achieving convergence in our experiments…"

- **Line 464-476: The authors discuss the coupling of the DA system in the result section (Section 4.3). The reviewer suggests providing this information in the section that describes the DA system.**

  Response: We sincerely appreciate the reviewer's valuable suggestion. Following the recommendation, we have provided this information in Section 3 that describes the DA system, as follows:

  "3.3.5 Flow-dependent background error covariance in CMA-GFS-AERO 4D-Var

  In the strongly coupled aerosol-meteorology assimilation system, interactions between the atmospheric variables and BC allow BC observations to influence the analysis increment of atmospheric variables and vice versa. The incremental 4D-Var algorithm implicitly evolves the background error covariances ( $\mathbf{B}$ ) throughout the assimilation window according to the TL model dynamics. This process modifies prior background error variance estimates and induces non-zero correlations between model variables (Smith et al., 2015). By utilizing the fully coupled TLM and ADM in the inner loops of the strongly coupled assimilation system, cross-covariance information between BC and atmospheric variables is generated. This enables observations of one variable to produce analysis increments in the other, leading to more consistent analyses.

  Specifically, if the BC observation is assumed to take place at the initial of the assimilation window, the 4D-Var assimilation is equivalent to the 3D-Var assimilation. Since the BC variable is assumed to be uncorrelated with the atmospheric variables in the static $\mathbf{B}$, and there is no direct relationship between the BC observation operator and the atmospheric variables, the BC observation does not lead to the generation of the analysis

increments of atmospheric variables. In this case, the merits of a coupled data assimilation system cannot be fully manifested by only assimilating a BC observation at the beginning of the window. If the BC observation is assumed to take place at the middle and the end of the assimilation window, $\mathbf{B}$ evolves within the assimilation time window through the TLM $\mathbf{M}_{0\to i}$, obtaining the implicit background error covariance matrix $\mathbf{M}_{0\to i}\mathbf{B}\mathbf{M}_{0\to i}^{T}$ that evolves with time. $\mathbf{M}_{0\to i}\mathbf{B}\mathbf{M}_{0\to i}^{T}$ includes the cross-covariances information of BC and atmospheric variables, and can realize the feedback of the BC observation to the atmospheric variables through the CMA-GFS-AERO ADM $\mathbf{M}_{0\to i}^{T}$, further producing analysis increments of atmospheric variables."

Additionally, we have revised the description of Figure 7 in the manuscript to make it more concise:

"Figure 7 depicts the analysis increments of temperature at the first model level at the initial time of the assimilation time window (0300 UTC), with the BC observation placed at 0600 and 0900 UTC, respectively. It can be seen that when the BC observation is placed at 0600 and 0900 UTC, positive analysis increments of temperature are generated, with the value of about 0.02 K near the observation location. The mechanism behind the generation of these temperature increments is detailed in Section 3.3.5. This indicates that the temperature of the analysis field will increase due to the assimilation of the BC observation."

- **Line 540: In EXP3, if the atmospheric and the BC variable were minimized separately (as stated in line 503), would the assimilation of the BC variable affect the atmospheric variable? If so, is it due to a change in the background when the model is coupled? Does this also mean that the results of EXP2 and EXP1 should be similar (the authors did not show the results of EXP2)?**

  Response: We sincerely appreciate the reviewer's insightful comments. We once again apologize for the confusion caused by the unclear description in Section 4.4 of the original manuscript. After carefully considering the reviewer's comments, we have completely rewritten this section, which is now presented as Section 5.3 in the revised manuscript. In this revision, we have clarified the distinction between different assimilation approaches and refined the explanation of how BC assimilation interacts with atmospheric variables, ensuring a more accurate and intuitive presentation of the results. These updates also address the reviewer's concerns about the impact of BC assimilation on atmospheric variables and the relationship between different experiments.

- **Figure 10: How many BC observations are used in this plot?**

  Response: Thanks for the insightful comment. The assimilation window of CMA-GFS-AERO 4D-Var is 6 hours. We assimilated all available BC observations at an hourly frequency within this window, excluding any missing data, resulting in 135 observations in total.

- **Line 549: 10f is not similar to 10d and 10e.**

  Response: Thanks for pointing this out. As we mentioned in our previous response, after carefully considering the reviewer's comments, we have completely rewritten the original Section 4.4, which is now presented as Section 5.3 in the revised manuscript. In the

updated version, the statement that "10f is similar to 10d and 10e" has been removed. We once again appreciate the reviewer's valuable feedback.

- **Line 553-556: This sentence seems to compare the analysis increments in EXP4 with those in EXP1 and EXP3. However, the right panels of Figure 10 show the differences in analysis increments between EXP4 and EXP2, not the analysis increments in EXP4.**
  Response: We sincerely appreciate the reviewer's insightful comments. As we mentioned in our previous response, after carefully considering the reviewer's comments, we have completely rewritten the original Section 4.4, which is now presented as Section 5.3 in the revised manuscript. In the updated version, we have explicitly clarified that the atmospheric analysis increments shown in Figure 10 are solely generated by BC assimilation and do not include contributions from atmospheric observations assimilation. Throughout the analysis, we have ensured that all related descriptions are precise and clearly articulated. We once again thank the reviewer's valuable comments.

- **Line 557-558 and 606-608: To reveal the difference in analysis increments between weak and strong coupling, a comparison between EXP4-EXP2 and EXP3-EXP2 is needed.**
  Response: Thanks for the valuable comment. As we mentioned in our previous response, our expression in Section 4.4 in the original manuscript was not sufficiently clear, which may have caused confusion. We apologize for any misunderstanding. In Section 5.3 of the revised manuscript, we have compared the atmospheric analysis increments caused by BC assimilation in DA_MET_then_BC (DA_MET_then_BC - DA_MET) and DA_MET_BC_simult (DA_MET_BC_simult - DA_MET). Additionally, the statement in Lines 606-608 of the Conclusion section has been removed accordingly. We once again appreciate the reviewer's valuable suggestions.

- **Tables 4, 5 and 6: Are the computation times in these tables the average of multiple realisations?**
  Response: Thanks for the insightful comment. Yes, the computation times in Tables 4, 5, and 6 are the averages of multiple realizations. We conducted several repeated experiments to ensure the robustness and representativeness of the results. The values shown in the tables are the mean computation times from all the repeated runs.

**Technical corrections:**

- **Line 111-112: "… (Canadian Aerosol Module; (Gong et al., 2003)) …)"**
  Response: Thanks for pointing this out. We have revised it as follows:
  "…CAM (Canadian Aerosol Module; Gong et al., 2003) ..."

- **Equation 12: R is used to define two things in the same equation.**
  Response: Thanks for pointing this out. We revised the equation to avoid ambiguity by replacing $R$ with $R_d$ in $k_z = \frac{g^2}{(R_d T_0)^2} k_p$. The revised text now reads as follows:

"$\ldots k_z = \frac{g^2}{(R_d T_0)^2} k_p$, $g$ denotes the gravitational acceleration, $R_d$ represents the gas constant for dry atmospheric air …"

- **Line 552: Extra space between "values" and "in".**
  Response: Thanks for pointing out this formatting issue. We have corrected the extra space between "values" and "in".

---

## Referee Report (RR1)

**Review of the *revised version* of the paper "Development of the CMA-GFS-AERO 4D-Var assimilation system v1.0 - Part I: System description and preliminary experimental results."**

by Yongzhu Liu & Colleagues, DOI: https://doi.org/10.5194/gmd-2024-148

MS type: Development and technical paper
Iteration: Revised submission
2025-03-10

**General Comments**

I in general welcome the changes in the reviewed version of the manuscript as they provide more clarification and enter a little bit more into explaining the results. I however still have concerns _before_ being able to recommend the paper for publication.

At first, I am now confused by the modified pictures in Figures 9 and 10. Following the suggestion of Reviewer 1, the authors have renamed the labels of the experiences in a more explicit manner (Table 3), which is good. However, my understanding is that these panels refer to exactly the same experiments. I would thus expect the pictures to show exactly the same fields based on the same data. Why then are the panels displaying changed fields of increments and differences of increments differing from the original version of the paper? I have no explanation for that myself, and I see none in the answers to the reviewers nor in the paper.

Regarding Figure 7, can the authors really be explicit and state in the paper whether or not the analysis increments for temperature systematically are positive when only one single BC observation is assimilated ? I actually see no reason for it being so, but I would like the authors to provide full clarity on this, based on their experimental results. If their answer is yes (this is systematic), an explanation why would be definitely welcome.

The statements in lines 748-750 are either too strong or totally obscure, and likely a bit of both. I simply cannot understand why the experimental results in Section 5 enable one to state that "meteorological observations can help constrain the uncertainty introduced by BC observations on atmospheric variables, thereby improving the reliability of the assimilation results". My own strong reaction could be due to the fairly unclear formulation in the text:

why or how do BC observations "introduce uncertainty" on atmospheric variables ? What is meant by this formulation ? How is "reliability" defined in this context ?

Likewise, I actually dispute that "these results _demonstrate_ the successful implementation of the newly developed CMA-GFS-AERO 4D-Var system".

What Is meant by "successful" here ? Should the authors wish to provide proof (a demonstration) of the successful implementation in terms of exactitude of _expected_ results, then the only way would be to use their coupled assimilation system in a _fully controlled_ modus operandi. One possibility would be OSSE with fully simulated observations of all types, and using prescribed error statistics in the system error covariances and for perturbing the simulated observations from a ground truth. In such experimental mode, the output results should follow some statistical results that separately could be derived from the expected posterior probability density functions (i.e. independently from the experiments).

As this is not what the authors have done, and I am not claiming they should do this, my understanding is that the authors have proven that their 4D-VAR system produces "credible and realistic-looking" results: "credible" in the sense that the results could lead to further explanations with the knowledge the authors have, and "realistic-looking" in the sense that the increments display realistic structures and amplitudes case-by-case.

I therefore strongly invite the authors to consider reformulating their statements at this location in Section 5 (around lines 748-750), and elsewhere where relevant such as in the Conclusion (lines 819-820).

I would recommend some English proofreading of the paper.

**Specific Comments & Typos**

line 408: "Specifically, if the BC observation is assumed to take place at the initial of the assimilation window," => "Specifically, if the BC observation is assumed to take place at the  **beginning** of the assimilation window,"
(same suggestion holds for other locations in the text such as for instance in the caption of Figure 7)

Figure 10 and everywhere in the paper where "pressure" is referred to: what is exactly this field ? is it "surface pressure" (ie following elevation), is it "mean sea level pressure" ? is it "pressure at the first model level" ? or any other definition of "pressure" ?

---

## Referee Report (RR2)

A Review of

A revised version of "Development of the CMA-GFS-AERO 4D-Var assimilation system v1.0- Part 1: System description and preliminary experimental results"

submitted to Geoscientific Model Development by Liu et al. (2024)

Review Decision: Minor Revisions

Manuscript type: _Development and technical papers_

General Comments:

The revised manuscript is a huge improvement from the original version in various aspects, including readability, clarity, and coherence. The authors' effort on revising the manuscript is greatly applauded. I've also found the revised manuscript to be very educational in guiding the readers on developing a coupled aerosol-meteorology 4D-Var assimilation system step by step with important details. After a thorough review, I only have a few minor comments that I'd recommend the authors to address before the manuscript could be considered for publication.

Minor comments:

1.  What is a batch test? My guess is that it is an integrated set of tests that involves many cases, but I am not 100% sure since it sounds more like a test used in software development. Is batch test a common terminology used in operational NWP centers? I wondered whether it is okay to assume that the readers understand what a batch test is. I would rather use more words in plain language to describe what the author intends to do next instead of a terminology that might not be universally recognized.

2.  Page 1, Line 13: chemical should be replaced by aerosol.

3.  Pages 2-3, Lines 36-66: this paragraph is too long. Please consider breaking it into two paragraphs. I recommend that a new paragraph could be started from line 58, focusing on the role of background error covariance in high-dimensional CCMM system.

4.  Page 4, Line 103: chemical should be replaced by aerosol.

5.  Page 4, Lines 104-105: it should be five sections, not four, as the remaining sections are sections 2, 3, 4, 5, and 6 (total of 5).

6. Pages 4-5, Lines 123-125: aren't those water matter species produced by the double-moment cloud microphysics scheme also considered state variables of the CMA-GFS NLM?

7. Page 8, Line 236: Upon reading this, I couldn't help but wondering that isn't AERO-BC also 1-D modules? And then, I found the authors actually hinted this at page 7 line 198. I suggest making this point clearer upfront. Also, I am puzzled by the description about "1-D modules with fixed latitude and longitude coordinates"… if it has lat/lon coordinates, then isn't it 2-D?

8. Page 13, Line 362: "at the initial of the assimilation window," it would be better to put "only" right after assimilation window to highlight this very special scenario.

9. Page 22, Line 575 and Table 3: I think there is a need to explicitly explain how the two experiments DA_MET_then_BC and DA_MET_BC_simult are conducted differently in terms of their workflows. For DA_MET_then_BC, does that mean the CMA-GFS-AERO 4DVar system is run two times with the same setup, except for the observations being assimilated? In the first run, only the operational meteorological observations are assimilated and then the resulting analysis is used as background for the second run, where only the BC surface observations are assimilated. Is that right?

10. Page 23, Line 602: I am not sure what is the purpose of this sentence.

11. Page 24, Lines 626-627: ok, but why not just show the analysis increment of DA_MET_then_BC and have it compared with those from DA_BC? I still have trouble understanding using the "difference of analysis increment of two experiment" to compare with the "analysis increment of one experiment". If the concern is because the analysis increments of DA_MET_then_BC and DA_BC includes not only the impact of meteorological observations but also the impact from using different background fields (the nature of DA_MET_then_BC), and the authors would like to isolate the impact of BC from all other factors, then it has to be stated clearly to justify such comparison (which also echoes my minor comment 9). However, I am not sure whether the same reasoning applies to the case for comparing the difference of analysis increment of DA_MET_BC_simult and DA_MET with the analysis increment of DA_BC.

---

## Referee Report (RR3)

[referee-annotated manuscript omitted]

---

## Author Response (AR2)

**Response to Reviewer #1 for Geoscientific Model Development:**
**A revised version of manuscript gmd-2024-148**
**By Liu et al.**

We sincerely thank Reviewer #1 for insightful and constructive feedback on the revised version of our manuscript. We have carefully considered each comment and made further revisions accordingly. The notes below address each comment in detail. Please note that the reviewer's comments are shown in bold type and our responses are in plain type.

**Reviewer #1**

**Review Decision: Minor Revisions**
**Manuscript type: _Development and technical papers_**

**General Comments:**
**The revised manuscript is a huge improvement from the original version in various aspects, including readability, clarity, and coherence. The authors' effort on revising the manuscript is greatly applauded. I've also found the revised manuscript to be very educational in guiding the readers on developing a coupled aerosol-meteorology 4DVar assimilation system step by step with important details. After a thorough review, I only have a few minor comments that I'd recommend the authors to address before the manuscript could be considered for publication.**

Response: We sincerely thank the reviewer for the encouraging and constructive feedback. We are very grateful for the positive assessment of our revised manuscript, especially the comments on the improvements in readability and clarity, as well as the educational value of the work. We truly appreciate the recognition of our efforts in developing and describing the coupled aerosol-meteorology 4DVar assimilation system.

We have carefully addressed all the remaining minor comments in the revised manuscript, as detailed below. We also sincerely appreciate the reviewer's time and effort in thoroughly reviewing the revised version and offering valuable suggestions. We hope that the modifications and corrections we have made will meet the standards for publication.

**Minor comments:**
1. **What is a batch test? My guess is that it is an integrated set of tests that involves many cases, but I am not 100% sure since it sounds more like a test used in software development. Is batch test a common terminology used in operational NWP centers? I wondered whether it is okay to assume that the readers understand what a batch test is. I would rather use more words in plain language to describe what the author intends to do next instead of a terminology that might not be universally recognized.**

   Response: We sincerely appreciate the insightful comments and valuable suggestions. In our study, the term "batch test" refers to a series of cycling assimilation experiments conducted over a certain period (e.g., three months), in which the CMA-GFS-AERO 4D-Var system is run at 6-hour assimilation cycles to evaluate its performance in a realistic forecast-assimilation framework. We acknowledge that "batch test" may not be a widely

recognized term in the NWP community. To enhance clarity, we have revised the manuscript to replace "batch test" with "cycling assimilation experiments". The specific modifications are as follows:

Abstract:

"…This study focuses on the theoretical architecture and practical implementation of the system, the detailed analysis of a series of cycling assimilation experiments will be described in part 2 of this paper."

4 Model setup

"…The detailed analysis of the cycling assimilation experiments will be further elaborated in part 2 of this paper."

5.3 Case study on BC and atmosphere assimilation

"…In the future, we will conduct cycling assimilation experiments using CMA-GFS-AERO 4D-Var to…"

6 Conclusions

"…We intend to explore the impact of assimilating surface BC observations on the forecast fields of BC and atmospheric variables through cycling assimilation experiments…"

2. **Page 1, Line 13: chemical should be replaced by aerosol.**

Response: Thanks for pointing this out. We have revised "chemical" to "aerosol" as suggested.

3. **Pages 2-3, Lines 36-66: this paragraph is too long. Please consider breaking it into two paragraphs. I recommend that a new paragraph could be started from line 58, focusing on the role of background error covariance in high-dimensional CCMM system.**

Response: Thanks for the valuable comment. We have revised this section by splitting it into two paragraphs, starting a new one from line 58 as suggested. The new paragraph emphasizes the role of background error covariance in high-dimensional CCMM systems, improving the clarity of the structure.

4. **Page 4, Line 103: chemical should be replaced by aerosol.**

Response: Thanks again for the valuable suggestion. We have made the change accordingly.

5. **Page 4, Lines 104-105: it should be five sections, not four, as the remaining sections are sections 2, 3, 4, 5, and 6 (total of 5).**

Response: Thanks for the careful review. We have corrected this mistake and changed "four" to "five" as suggested.

6. **Pages 4-5, Lines 123-125: aren't those water matter species produced by the double-moment cloud microphysics scheme also considered state variables of the CMA-GFS NLM?**

Response: Thanks for the insightful comment. In the previous text, we did not explicitly list those water matter species produced by the double-moment cloud microphysics scheme as state variables, primarily because, in operational practice, they are not preserved in the initial fields used for official forecasts.

Specifically, although the analysis fields (produced at 03, 09, 15, and 21 UTC) include these water matter species, the initial conditions used in official forecasts (00, 06, 12, and 18 UTC) are derived from 3-hour forecasts based on those analysis fields. After the 3-hour forecast integration, only a subset of model variables (wind components, potential temperature, specific humidity, and non-dimensional pressure) is saved due to the limitation of I/O time and disk space, while the fields of all hydrometeor contents and cloud cover are discarded (Ma et al., 2021). These variables are then regenerated during model spin-up in the forecast phase. Therefore, these hydrometeor species are generally not considered part of the state variables in the context of the CMA-GFS NLM.

Ma, Z., Zhao, C., Gong, J., Zhang, J., Li, Z., Sun, J., Liu, Y., Chen, J., and Jiang, Q.: Spin-up characteristics with three types of initial fields and the restart effects on forecast accuracy in the GRAPES global forecast system, Geosci. Model Dev., 14, 205-221, https://doi.org/10.5194/gmd-14-205-2021, 2021.

7. **Page 8, Line 236: Upon reading this, I couldn't help but wondering that isn't AERO-BC also 1-D modules? And then, I found the authors actually hinted this at page 7 line 198. I suggest making this point clearer upfront. Also, I am puzzled by the description about "1-D modules with fixed latitude and longitude coordinates"… if it has lat/lon coordinates, then isn't it 2-D?**

Response: We sincerely appreciate the insightful comments and valuable suggestions. We apologize for any confusion caused by our previous description. The AERO-BC module and its tangent linear (TL) and adjoint versions are designed as one-dimensional (1-D) column modules. Each module operates independently on a single vertical column corresponding to a fixed horizontal location (i.e., fixed latitude and longitude). In this context, "1-D modules with fixed latitude and longitude coordinates" refers to the fact that each 1-D module processes only the vertical profile at a single horizontal grid point, without involving horizontal interactions within the module itself. To form a global 3-D system, these 1-D modules are applied independently at every horizontal grid point, and the interface programs are responsible for passing 3-D meteorological and emission fields from CMA-GFS to each of these 1-D modules. This 3-D extension is also implemented for their TL and adjoint versions.

Following the reviewer's suggestion, we have revised Section 3.1 to make this point clearer. The updated description now reads:

"…The AERO-BC module is designed as one-dimensional (1-D) column module, which operates at individual vertical columns corresponding to fixed horizontal locations (i.e., fixed latitude and longitude). In the integration of AERO-BC with CMA-GFS, the interface programs transfer meteorological parameters (e.g., temperature, wind, and humidity) from CMA-GFS to AERO-BC, extend the spatial dimension from 1-D to 3-D, and read emissions for AERO-BC…"

We have also revised the corresponding sentence in Section 3.2 for clarity:

"The TL and the adjoint of AERO-BC are 1-D column modules, meaning they operate independently at each fixed horizontal grid point (i.e., fixed latitude and longitude), with vertical variation only. To extend them to 3-D, the tangent linear and the adjoint of the interface programs were also constructed…"

8. **Page 13, Line 362: "at the initial of the assimilation window," it would be better to put "only" right after assimilation window to highlight this very special scenario.**

   Response: Thanks for the valuable suggestion. We have revised this sentence by placing "only" right after "assimilation window" to highlight this special scenario.

9. **Page 22, Line 575 and Table 3: I think there is a need to explicitly explain how the two experiments DA_MET_then_BC and DA_MET_BC_simult are conducted differently in terms of their workflows. For DA_MET_then_BC, does that mean the CMA-GFS-AERO 4DVar system is run two times with the same setup, except for the observations being assimilated? In the first run, only the operational meteorological observations are assimilated and then the resulting analysis is used as background for the second run, where only the BC surface observations are assimilated. Is that right?**

   Response: Thanks for pointing this out. Yes, the reviewer's understanding is correct. In DA_MET_then_BC, the CMA-GFS-AERO 4D-Var system is indeed run twice. First, we assimilate only operational meteorological observations, and then the resulting analysis is used as the background field for a second assimilation run that includes only BC surface observations. In contrast, DA_MET_BC_simult assimilates both operational meteorological and BC surface observations simultaneously within a single assimilation run. We have revised the text to explicitly describe their workflows, which now reads as follows:

   "…These experiments are listed in Table 3…For the DA_MET_then_BC experiment, the CMA-GFS-AERO 4D-Var system was executed twice sequentially within the same assimilation window. In the first step, only operational meteorological observations were assimilated, and the resulting analysis was used as the background field for the second step, in which only BC surface observations were assimilated. Except for the observational datasets, the model configurations and assimilation settings in both steps remained identical. This two-step procedure allows us to separate the effect of BC observations from the influence of meteorological observations and their associated background adjustment, thereby facilitating a clearer attribution of the BC assimilation impact. In contrast, DA_MET_BC_simult assimilated both operational meteorological observations and BC surface observations simultaneously within a single 4DVar run. This one-step assimilation strategy allows all observations to jointly influence the analysis field, reflecting the integrated effect of both meteorological and BC observations…"

10. **Page 23, Line 602: I am not sure what is the purpose of this sentence.**

    Response: Thanks for the comment. We agree that the original sentence was unclear. The purpose was to emphasize that, in Fig. 10, we aim to isolate the impact of BC assimilation. Therefore, we show the difference between DA_MET_then_BC and DA_MET, and between DA_MET_BC_simult and DA_MET, in order to remove the contributions from the assimilation of meteorological observations. Considering this comment together with Comment #11, we have revised the text as follows:

    "…Figure 10 shows the analysis increments of temperature, pressure, east-west component of horizontal wind, and relative humidity at the first model layer, resulting from BC assimilation in DA_BC, DA_MET_then_BC, and DA_MET_BC_simult. It is worth noting that in DA_BC,

only BC observations are assimilated, so the analysis increments of atmospheric variables purely reflect the response to BC. In contrast, both DA_MET_then_BC and DA_MET_BC_simult assimilate BC and meteorological observations, and thus their analysis increments include the combined effects of both types of observations. To isolate the influence of BC assimilation alone on atmospheric variables, and under the assumption that the contribution of meteorological observations is comparable between DA_MET_then_BC/DA_MET_BC_simult and DA_MET, we calculated the differences between the analysis increments of these experiments and those from DA_MET, which assimilates only meteorological observations. This subtraction effectively removes the contributions from meteorological observations, allowing the resulting increments to be attributed solely to the assimilation of BC observations. In this way, a more direct and fair comparison can be made with DA_BC…"

11. **Page 24, Lines 626-627: ok, but why not just show the analysis increment of DA_MET_then_BC and have it compared with those from DA_BC? I still have trouble understanding using the "difference of analysis increment of two experiment" to compare with the "analysis increment of one experiment". If the concern is because the analysis increments of DA_MET_then_BC and DA_BC includes not only the impact of meteorological observations but also the impact from using different background fields (the nature of DA_MET_then_BC), and the authors would like to isolate the impact of BC from all other factors, then it has to be stated clearly to justify such comparison (which also echoes my minor comment 9). However, I am not sure whether the same reasoning applies to the case for comparing the difference of analysis increment of DA_MET_BC_simult and DA_MET with the analysis increment of DA_BC.**

Response: We sincerely appreciate the reviewer's insightful comments. The purpose of comparing the difference in analysis increments between DA_MET_then_BC and DA_MET ( DA_MET_then_BC – DA_MET) with the increments from DA_BC is to isolate the net impact of BC observations on atmospheric variables, excluding the influence of meteorological observations. In the DA_BC experiment, only BC observations are assimilated, so the analysis increments of atmospheric variables purely reflect the response to BC.

In contrast, both DA_MET_then_BC and DA_MET_BC_simult assimilate BC and meteorological observations, and thus their analysis increments include the combined effects of both types of observations. By subtracting DA_MET (which only assimilates meteorological observations), we remove the meteorological component, thereby isolating the contribution of BC assimilation. This approach enables a more direct and fair comparison with DA_BC, where only BC observations are assimilated.

Regarding the reviewer's question on whether the same reasoning applies to DA_MET_BC_simult, we confirm that it does. Although DA_MET_BC_simult assimilates BC and meteorological observations simultaneously, subtracting the analysis increments of DA_MET from those of DA_MET_BC_simult still allows us to isolate the impact of BC assimilation alone-under the assumption that the contribution of meteorological observations is comparable between DA_MET_BC_simult and DA_MET.

We have clarified this in the revised manuscript as follows:

"…Figure 10 shows the analysis increments of temperature, pressure, east-west component of

horizontal wind, and relative humidity at the first model layer, resulting from BC assimilation in DA_BC, DA_MET_then_BC, and DA_MET_BC_simult. It is worth noting that in DA_BC, only BC observations are assimilated, so the analysis increments of atmospheric variables purely reflect the response to BC. In contrast, both DA_MET_then_BC and DA_MET_BC_simult assimilate BC and meteorological observations, and thus their analysis increments include the combined effects of both types of observations. To isolate the influence of BC assimilation alone on atmospheric variables, and under the assumption that the contribution of meteorological observations is comparable between DA_MET_then_BC/DA_MET_BC_simult and DA_MET, we calculated the differences between the analysis increments of these experiments and those from DA_MET, which assimilates only meteorological observations. This subtraction effectively removes the contributions from meteorological observations, allowing the resulting increments to be attributed solely to the assimilation of BC observations. In this way, a more direct and fair comparison can be made with DA_BC…"

We sincerely thank Reviewer #2 for insightful and constructive feedback on the revised version of our manuscript. We have carefully considered each comment and made further revisions accordingly. The notes below address each comment in detail. Please note that the reviewer's comments are shown in bold type and our responses are in plain type.

**Reviewer #2**

**MS type: Development and technical paper**
**Iteration: Revised submission**
**2025-03-10**

**General Comments**
**I in general welcome the changes in the reviewed version of the manuscript as they provide more clarification and enter a little bit more into explaining the results. I however still have concerns _before_ being able to recommend the paper for publication.**
Response: We sincerely thank the reviewer for the careful review of our revised manuscript and for acknowledging the improvements we have made. We greatly appreciate the reviewer's continued engagement and constructive feedback, which are invaluable in further improving the clarity and scientific rigor of our work.
In the revised version, we have carefully addressed the remaining concerns and added further clarifications where necessary. Detailed point-by-point responses are provided below. We also sincerely appreciate the reviewer's time and effort in thoroughly reviewing the revised version and offering valuable suggestions. We hope that the modifications and corrections we have made will meet the standards for publication.

**At first, I am now confused by the modified pictures in Figures 9 and 10. Following the suggestion of Reviewer 1, the authors have renamed the labels of the experiences in a more explicit manner (Table 3), which is good. However, my understanding is that these panels refer to exactly the same experiments. I would thus expect the pictures to show exactly the same fields based on the same data. Why then are the panels displaying changed fields of increments and differences of increments differing from the original version of the paper? I have no explanation for that myself, and I see none in the answers to the reviewers nor in the paper.**
Response: We sincerely apologize for omitting an explanation regarding the updates to Figures 9 and 10 during the revision process and for any confusion this may have caused. We greatly appreciate the insightful comments and careful examination of the figures.
The differences between the revised and original Figures 9 and 10 are due to a modification in the radius of influence for BC observations. Specifically, Reviewer #1 pointed out the importance of considering the station type when determining the radius of influence for BC observations in the calculation of BC observation error. According to Table 3 of Elbern et al. (2007), the radius of

influence differs for urban, rural, and remote stations. Initially, we used a uniform 10 km radius for all 32 CAWNET stations, but after reviewing this suggestion, we realized that this approach was inappropriate. In the revised manuscript, we adopted station-type-specific radii: 2 km for urban stations, 10 km for rural stations, and 20 km for remote stations, following Table 3 of Elbern et al. (2007).

This change is explicitly described in Section 3.3.2 of the revised manuscript as follows:

"The BC observation data were collected from 32 stations (Guo et al., 2020), including 11 urban, 17 rural, and 4 remote stations…"

"…and $L$ is the radius of influence of a BC observation. According to Elbern et al. (2007), $L$ was set to 2 km, 10 km, and 20 km for urban, rural, and remote stations, respectively…"

Since we updated the radius of influence of BC observations, all experiments in Section 5.3 have been redone using the revised settings. Consequently, the corresponding figures and text have been updated to reflect the new results.

We once again appreciate the reviewer's careful examination of Figures 9 and 10 in our revised manuscript.

**Regarding Figure 7, can the authors really be explicit and state in the paper whether or not the analysis increments for temperature systematically are positive when only one single BC observation is assimilated? I actually see no reason for it being so, but I would like the authors to provide full clarity on this, based on their experimental results. If their answer is yes (this is systematic), an explanation why would be definitely welcome.**

Response: We sincerely thank the reviewer for pointing out the need for clarification. We apologize for any confusion caused by our earlier description. The temperature analysis increment is not systematically positive when a single BC observation is assimilated. Instead, it depends on several factors, including the BC observation innovation, the location of the observation, and the meteorological conditions during the assimilation time window.

In the specific case shown in Figure 7, the BC observation innovations are negative (i.e., -9.5 μg/m³ at 0600 UTC and -9.0 μg/m³ at 0900 UTC), indicating that the background BC concentration is lower than the observed values. Assimilation of these observations increases the BC concentrations in the analysis, which, under the prevailing meteorological conditions, leads to positive temperature increments near the observation site. As explained in Section 3.3.5, the coupling between BC and atmospheric variables within the system allows this type of feedback to occur.

Therefore, the positive temperature increments observed in Figure 7 are a result of this specific experiment setup and should not be interpreted as a systematic behavior. We have revised the manuscript to make this point clearer. The revised text now reads as follows:

"Figure 7 depicts the analysis increments of temperature at the first model level at the beginning time of the assimilation time window (0300 UTC), with the BC observation placed at 0600 and 0900 UTC, respectively. In this specific case, the analysis increments of temperature are positive, with the value of about 0.02 K near the observation location, when the BC observation is placed at 0600 and 0900 UTC. The temperature analysis increment depends on several factors, including the BC observation innovation, the location of the observation, and the meteorological conditions during the assimilation time window. Here, the positive analysis increments of temperature may be due to the fact that the BC observation innovations at 0600 and 0900 UTC are negative (-9.5

μg/m³ at 0600 UTC and -9.0 μg/m³ at 0900 UTC), indicating that the background BC concentration is lower than the observed values. Assimilation of these observations increases the BC concentrations in the analysis, which, under the prevailing meteorological conditions, leads to positive temperature increments near the observation site. As explained in Section 3.3.5, the coupling between BC and atmospheric variables within the system allows this type of feedback to occur."

We once again appreciate the reviewer's insightful comment and valuable suggestions.

**The statements in lines 748-750 are either too strong or totally obscure, and likely a bit of both. I simply cannot understand why the experimental results in Section 5 enable one to state that "meteorological observations can help constrain the uncertainty introduced by BC observations on atmospheric variables, thereby improving the reliability of the assimilation results". My own strong reaction could be due to the fairly unclear formulation in the text: why or how do BC observations "introduce uncertainty" on atmospheric variables ? What is meant by this formulation ? How is "reliability" defined in this context ?**

**Likewise, I actually dispute that "these results _demonstrate_ the successful implementation of the newly developed CMA-GFS-AERO 4D-Var system".**

**What Is meant by "successful" here ? Should the authors wish to provide proof (a demonstration) of the successful implementation in terms of exactitude of _expected_ results, then the only way would be to use their coupled assimilation system in a _fully controlled_ modus operandi. One possibility would be OSSE with fully simulated observations of all types, and using prescribed error statistics in the system error covariances and for perturbing the simulated observations from a ground truth. In such experimental mode, the output results should follow some statistical results that separately could be derived from the expected posterior probability density functions (i.e. independently from the experiments).**

**As this is not what the authors have done, and I am not claiming they should do this, my understanding is that the authors have proven that their 4D-VAR system produces "credible and realistic-looking" results: "credible" in the sense that the results could lead to further explanations with the knowledge the authors have, and "realistic-looking" in the sense that the increments display realistic structures and amplitudes case-by-case.**

**I therefore strongly invite the authors to consider reformulating their statements at this location in Section 5 (around lines 748-750), and elsewhere where relevant such as in the Conclusion (lines 819-820).**

Response: We sincerely appreciate the insightful comments and valuable suggestions. We fully agree that some of the statements in the original manuscript were too strong or not clearly formulated. In the revised version, we have removed ambiguous or overstated terms such as "introduce uncertainty," "improve reliability," and "successful implementation." Instead, we now emphasize what the experimental results actually suggest: the presence of meteorological observations during assimilation may provide additional constraints on the adjustment of atmospheric fields, potentially reducing the degree to which the assimilation of BC observations alone can alter the atmospheric state. In this way, the integration of meteorological observations helps stabilize the adjustment process, supporting more consistent and interpretable assimilation results.

We also agree that the term "successful implementation" may imply a rigorous demonstration of

the system's exactitude against a defined ground truth, such as in an Observing System Simulation Experiment (OSSE) framework with prescribed error statistics and known truth states. Since our study is not based on such a controlled setup, we acknowledge that using the term "successful implementation" could be misleading. In the revised manuscript, we have replaced this term with a more accurate description of what the current experiments actually demonstrate: the CMA-GFS-AERO 4D-Var system has been technically implemented and is able to produce credible assimilation increments when assimilating BC and meteorological observations. The increments exhibit realistic structures and amplitudes, which suggests that the system performs as intended under the current configuration and available observations.

We have updated the corresponding statements in Section 5.3 and the Conclusion as follows:

Section 5.3:

"…This suggests that the presence of meteorological observations during assimilation may provide additional constraints on the adjustment of atmospheric fields, potentially reducing the degree to which the assimilation of BC observations alone can alter the atmospheric state. In this way, the integration of meteorological observations helps stabilize the adjustment process, supporting more consistent and interpretable assimilation results. Moreover, the four experiments demonstrate that the CMA-GFS-AERO 4D-Var system has been technically implemented and is able to produce credible analysis increments in both BC and atmospheric fields. These increments display realistic spatial structures and amplitudes, indicating that the system performs as intended under the current configuration and available observations. These results offer practical evidence of the system's functionality and its potential utility for exploring the feedback of BC data assimilation on meteorological forecasts. In the future..."

Conclusion:

"…This demonstrates that the newly developed CMA-GFS-AERO 4D-Var system has been technically implemented and is capable of producing credible assimilation outcomes, highlighting its potential as a useful tool for exploring the feedback of BC data assimilation on meteorological forecasts…"

We once again appreciate the reviewer's guidance, which helped improve the precision and clarity of our manuscript.

**I would recommend some English proofreading of the paper.**

Response: Thanks for the helpful comment. We have thoroughly proofread the manuscript to enhance the clarity and correctness of the English. We believe that the revised version is now clearer and more readable.

**Specific Comments & Typos**

**line 408: "Specifically, if the BC observation is assumed to take place at the initial of the assimilation window," => "Specifically, if the BC observation is assumed to take place at the  beginning of the assimilation window,"**

**(same suggestion holds for other locations in the text such as for instance in the caption of Figure 7)**

Response: Thanks for the valuable suggestion. We have revised "at the initial of the assimilation window" to "at the beginning of the assimilation window" in this line. We have also carefully checked the entire manuscript and made similar corrections in other relevant locations, such as the

caption of Figure 7.

**Figure 10 and everywhere in the paper where "pressure" is referred to: what is exactly this field ? is it "surface pressure" (ie following elevation), is it "mean sea level pressure" ? is it "pressure at the first model level" ? or any other definition of "pressure" ?**

Response: We sincerely appreciate the reviewer's insightful comments. The term "pressure" in Figures 8 and 10 and the corresponding text refers to "pressure at the first model level". This has been explicitly stated in the manuscript, as shown in the following examples:

"Figure 8 shows the analysis increments of pressure, east-west component of horizontal wind, and relative humidity at the first model level…"

"…Figure 10 shows the analysis increments of temperature, pressure, east-west component of horizontal wind, and relative humidity at the first model layer…"

To further clarify this, we have revised the manuscript by explicitly specifying "pressure at the first model level" when it is first introduced:

"Figure 8 shows the analysis increments of pressure at the first model level, as well as east-west component of horizontal wind and relative humidity at the same level…"

Regarding the other occurrences of "pressure" in the manuscript, such as the non-dimensional pressure ($\pi$) and the unbalanced Exner pressure ($\pi_u$), their definitions are already provided in the text, and no further clarification is necessary. We once again thank the reviewer's valuable comments.

We sincerely thank Reviewer #3 for insightful and constructive feedback on the revised version of our manuscript. We have carefully considered each comment and made further revisions accordingly. The notes below address each comment in detail. Please note that the reviewer's comments are shown in bold type and our responses are in plain type.

**Reviewer #3**

**The authors have addressed the reviewers' comments and the manuscript has been improved. However, I still have some concerns about the results in Section 5.3. Please find the detailed comments in the attached pdf.**

Response: We sincerely appreciate the reviewer's continued efforts in reviewing our manuscript and for pointing out further concerns regarding the results in Section 5.3. We have carefully reviewed the detailed comments provided in the attached PDF and have revised the manuscript accordingly to address each of them. Please find our point-by-point responses below. We hope that our revisions and explanations have adequately addressed the remaining concerns.

1. **Page 3, Line 68: Ensemble Kalman smoothers also use an assimilation window.**

   Response: We sincerely appreciate the insightful comment. We acknowledge that ensemble Kalman smoothers (EnKS) also utilize an assimilation window. To ensure clarity and avoid any misunderstanding that only 4D-Var employs an assimilation window, we have revised the text as follows:

   "…In high-dimensional problems, the limited number of samples may not be able to fully capture all the error characteristics, resulting in inaccuracies in the estimation of background error covariance. Although ensemble Kalman smoothers (EnKS) extend the EnKF framework by incorporating an assimilation window to leverage temporal observational information, they remain constrained by similar limitations in ensemble size. In contrast, 4D-Var explicitly integrates both the complete observational dataset and the full model dynamics within the assimilation window to constrain state evolution, rather than relying solely on ensemble statistics. This generally allows 4D-Var to achieve higher accuracy in high-dimensional problems by making better use of both observational data and model constraints, leading to more precise state estimation…"

2. **Page 3, Line 70: Does the CMA-GFS use an ensemble of variational data assimilations? If so, the background error covariances can be estimated in a similar way as in the EnKFs.**

   Response: We sincerely appreciate the insightful comment. The current version we used in this work, CMA-GFS v4.0, does not employ an ensemble of variational data assimilations; it strictly follows the 4D-Var framework. In later developments, the CMA-GFS En4DVar system has been introduced (e.g., in CMA-GFS v4.2), which incorporates ensemble-based methods to estimate background error covariances.

In coupled chemistry-meteorology models, estimating the cross-variable component of the covariance remains a significant challenge, regardless of whether a 4D-Var or ensemble-based approach is used. This challenge arises from the complex interactions between meteorological and chemical variables, making it difficult to accurately represent their covariance. In future work, we plan to explore hybrid data assimilation approaches within the CMA-GFS En4DVar framework to improve the estimation of background error covariances in coupled systems.

We once again appreciate the reviewer's insightful comment and valuable suggestion.

3. **Page 4, Line101: This sentence sounds like that the surface temperature is not part of the 3D temperature field.**

   Response: We appreciate the reviewer's careful reading. To avoid any potential misunderstanding, we have revised the sentence to clarify that the surface temperature is part of the overall atmospheric temperature field. The revised text now reads as follows:

   "…BC is also the main optically absorbing component of atmospheric aerosols, effectively absorbing solar radiation in the visible to infrared wavelength range, thus affecting the temperature field throughout the atmosphere, including the surface temperature…"

4. **Page 5, Lines 133-134: This is not a must, but it would be better if the authors could add some simple justification for the physical parameterization schemes chosen.**

   Response: Thanks for the valuable suggestion. We have added a brief justification for the selection of physical parameterization schemes. Specifically, we have stated that these schemes are consistent with those used in the operational application of CMA-GFS v4.0 and have been proven to perform well in global numerical weather prediction. The revised text now reads as follows:

   "…The physical parameterization schemes used in this work are consistent with those adopted in the operational application of CMA-GFS v4.0, which have been proven to perform well in global numerical weather prediction. The selected schemes mainly include..."

5. **Page 6, Section 2.2: The authors do not use a consistent rule for denoting matrices. For example, If bolded symbols are for matrices, then the matrix U should also be bold. I would suggest also using bolded but lowercase symbols for vectors. In addition, the symbols for matrices and vectors should not be italicized.**

   Response: We sincerely appreciate the reviewer's careful reading and valuable suggestions. We have revised the manuscript accordingly to ensure consistency in notation: matrices (e.g., $\mathbf{U}$, $\mathbf{B}$, $\mathbf{R}$) are now represented using bold uppercase letters, and vectors (e.g., $\mathbf{w}$, $\mathbf{d}$, $\mathbf{x}$) are represented using bold lowercase letters, as suggested. Additionally, we clarify that in the manuscript, italic and non-italic bold uppercase letters may appear simultaneously: the italic bold symbols denote nonlinear model operators, while the non-italic bold symbols represent their corresponding tangent linear versions. This notation is consistent with conventions commonly used in variational data assimilation literature.

6. **Page 11, Line 305: This is a follow-up question related to Reviewer #3's comment on the**

weights. **Estimating the weights based on observations (if possible) rather than on model forecasts seems to be a better idea to me.**

Response: We sincerely appreciate the reviewer's insightful follow-up question. We fully agree that estimating the size bin weights ($\omega^n$) based on observations, if available, could potentially improve the accuracy. However, currently available BC observations do not provide the detailed size-resolved distribution of BC. All observational datasets available to us only include the total mass concentration (unit: μg/m³) without distinguishing between size bins, as also mentioned in Section 3.3.1. As a result, a single observation of BC mass concentration cannot be used to determine the weights of the six size-bin variables. Therefore, it is technically infeasible to derive $\omega^n$ directly from observations.

Instead, we estimate $\omega^n$ based on the distribution of $\psi_{bc}^n$ in the background field, which is generated by the CMA-GFS-AERO model. Our analysis shows that the spatial variability of $\omega^n$ over China is relatively small, indicating that using domain-averaged model-based weights does not significantly affect the mass conservation or the quality of the reconstructed $\psi_{bc}^n$ field.

In future work, we agree that if high-resolution size-resolved BC observational data become available, they could be incorporated to refine the estimation of $\omega^n$ and potentially improve the overall performance of the system.

We once again appreciate the reviewer's valuable suggestion.

7. **Page 12, Line 338: As the authors stated, representativeness errors are part of the observation error. I understand that this kind of error can be caused by the interpolation, but I do not understand why it can be caused by the forecast forward model.**

   Response: Thanks for the valuable comment. Upon further consideration, we realize that our previous revision may have introduced some confusion. The reviewer is absolutely right that representativeness errors are part of the observation error, and they are typically associated with the observation operator, rather than the forecast model itself.

   To clarify this point, we have revised the sentence in the manuscript as follows:

   "…Representativeness errors reflect the inaccuracies in the observation operator and in the interpolation from the model grid to the observation location…"

   We sincerely appreciate the reviewer's insightful feedback, which helped us improve the clarity and accuracy of the manuscript.

8. **Page 15, Lines 408-409: I understand the point of the authors, but I think more care should be taken when writing. Observations of atmospheric variables in the window may affect the model control variable, BC mass concentration. Therefore, in a cycling environment, there is a difference to the 3D-Var system.**

   Response: Thanks for the valuable comment. We agree that the original sentence was too absolute and could be misleading. In the revised manuscript, we have clarified that our discussion refers to a single assimilation window without cycling. Specifically, under the assumption that the BC observation is assimilated at the beginning of the assimilation window only and cycling is not considered, the 4D-Var system behaves similarly to the 3D-Var system. We fully agree that in a cycling data assimilation environment, where

atmospheric observations are also assimilated, feedback from the atmosphere to BC can occur through model dynamics, which may affect BC concentration. Therefore, the 4D-Var system is indeed different from a standard 3D-Var system in such cases.

The revised text now reads as follows:

"…Specifically, if the BC observation is assumed to take place at the beginning of the assimilation window only, and under the assumption of a single, non-cycling assimilation window, the 4D-Var assimilation behaves similarly to the 3D-Var assimilation. In this case…"

9.  **Page 15, Lines 415-416: Again, caution is needed. It is better to refer the readers to the functions in Section 2.2. The matrix $MBM^T$ is the background error covariance matrix at an observation time. But here what we need should be cross-time error covariances, something like $MB$.**

    Response: We thank the reviewer for the helpful suggestion. To avoid any potential misunderstanding, we have revised the text as follows:

    "…If the BC observation is assumed to take place at the middle and the end of the assimilation window, $\mathbf{B}$ evolves within the assimilation time window through the TLM $\mathbf{M}_{0\to i}$, obtaining the implicit background error covariance matrix $\mathbf{M}_{0\to i}\mathbf{B}\mathbf{M}_{0\to i}^{T}$ at the observation time. $\mathbf{M}_{0\to i}\mathbf{B}\mathbf{M}_{0\to i}^{T}$ includes the cross-covariances information of BC and atmospheric variables, and can realize the feedback of the BC observation to the atmospheric variables through the CMA-GFS-AERO ADM $\mathbf{M}_{0\to i}^{T}$, further producing analysis increments of atmospheric variables. In other words, the distribution of the analysis increment at the observation time is determined by the cross-time error matrix $\mathbf{M}_{0\to i}\mathbf{B}$."

10. **Page 24, Section 5.3: It will be easier for the readers to follow if each point in Section 5.3 is described and explained in a separate paragraph.**

    Response: We thank the reviewer for the helpful suggestion. In the revised manuscript, we have restructured Section 5.3 by dividing the discussion into separate paragraphs, each focusing on a specific point or aspect of the analysis. This improves the clarity and readability of the section and makes it easier for readers to follow the logic and findings of the case study.

    Specifically, Section 5.3 now consists of the following seven paragraphs:

    (1) Overview of the experimental setup and design.

    (2) Presentation of BC analysis increments.

    (3) General introduction to the impact of BC assimilation on analysis increments of atmospheric variables (introduction of Figure 10).

    (4) Detailed description of the impact of BC assimilation on analysis increments of atmospheric variables in the DA_BC experiment.

    (5) Detailed description of the impact of BC assimilation on analysis increments of atmospheric variables in the DA_MET_then_BC experiment and its comparison with DA_BC.

    (6) Detailed description of the impact of BC assimilation on analysis increments of atmospheric variables in the DA_MET_BC_simult experiment and its comparison with DA_BC and DA_MET_then_BC.

    (7) Concluding remarks on the results of all four experiments.

We believe that this new structure better aligns with the reviewer's suggestion and significantly improves the flow and understanding of the analysis.

11. **Page 25, Line 642: Table 3?**
Response: Thanks for the careful review. We have corrected "Table 1" to "Table 3".

12. **Page 26, Line 675: Did the authors forget to implement their response to Comment 50 from the first reviewer?**
Response: We thank the reviewer for pointing out this oversight. In our response to Comment 50 from the first-round review, we clarified that the analysis increments presented in Figure 9 are valid at the beginning of the assimilation window, which is standard practice in 4D-Var. However, we acknowledge that this clarification was only made in the response letter and not explicitly included in the manuscript. To avoid any confusion for readers unfamiliar with this convention, we have now added a clarification in the revised manuscript as follows:
"Figure 9 presents the analysis increments of BC at the first model layer from the DA_BC, DA_MET_then_BC, and DA_MET_BC_simult experiments. These analysis increments are valid at the beginning of the assimilation window, as is standard in 4D-Var systems…"
We appreciate the reviewer's careful attention to detail.

13. **Page 26, Line 675: Why the new results are different to the previous ones, given that the experimental design are not changed?**
Response: We appreciate the reviewer's question regarding the differences in the new results in Figure 9. We sincerely apologize for omitting an explanation of the updates to Figure 9 and related content during the revision process.
Indeed, the overall experimental design remains unchanged. The differences between the new results and the previous ones are due to a modification in the radius of influence for BC observations. Specifically, Reviewer #1 pointed out the importance of considering the station type when determining the radius of influence for BC observations in the calculation of BC observation error. According to Table 3 of Elbern et al. (2007), the radius of influence differs for urban, rural, and remote stations. Initially, we used a uniform 10 km radius for all 32 CAWNET stations, but after reviewing this suggestion, we realized that this approach was inappropriate. In the revised manuscript, we adopted station-type-specific radii: 2 km for urban stations, 10 km for rural stations, and 20 km for remote stations, following Table 3 of Elbern et al. (2007).
This change is explicitly described in Section 3.3.2 of the revised manuscript as follows:
"The BC observation data were collected from 32 stations (Guo et al., 2020), including 11 urban, 17 rural, and 4 remote stations…"
"…and $L$ is the radius of influence of a BC observation. According to Elbern et al. (2007), $L$ was set to 2 km, 10 km, and 20 km for urban, rural, and remote stations, respectively…"
Since we updated the radius of influence of BC observations, all experiments in Section 5.3 have been redone using the revised settings. Consequently, the corresponding figures and text have been updated to reflect the new results.
We once again appreciate the reviewer's careful examination of our revised manuscript.

14. **Page 29, Lines 718-722: For the given result, I will think that the assimilation of meteorological observations does not update too much the model variables used to calculate the model equivalence to the BC observations. Otherwise, the observation-minus-background innovations in DA_MET_then_BC and DA_BC would be different, leading to differences in the analysis increments.**

Response: Thanks for the valuable comment. In the DA_MET_then_BC experiment, although the assimilation of meteorological observations does update atmospheric variables (e.g., wind, temperature, humidity), it does not change the background field of BC itself. Thus, the BC assimilation step in both DA_BC and DA_MET_then_BC uses the same BC background. As a result, the observation-minus-background (OMB) values for BC observations are almost identical in both DA_BC and DA_MET_then_BC, except for minor differences caused by the influence of updated meteorological fields on the observation operator. Therefore, the analysis increments of atmospheric variables caused by BC assimilation remain similar between the two experiments.

We have revised the corresponding text to clarify this point as follows:

"…This is because, although the DA_MET_then_BC experiment assimilates meteorological observations before BC surface observations, the background field of BC remains unchanged. While the assimilation of meteorological observations updates atmospheric variables, it does not directly alter the BC background field. Therefore, the OMB values for BC observations in DA_MET_then_BC are very close to those in DA_BC, with only minor differences caused by the slight influence of updated meteorological fields on the observation operator. As a result, the analysis increments of atmospheric variables due to BC assimilation are similar between the two experiments…"

15. **Page 29, Line 736: It seems that the BC observations and atmospheric observations are pulling the model trajectory in opposite direction. If this is the case, then the quality of the BC observations should be worried, or the specification of the observation error covariance matrix is poor.**

Response: We sincerely appreciate the insightful comments. We agree that the simultaneous assimilation of BC and meteorological observations may reflect some competing influences on the model state. In the revised text, we clarified that the reduced BC-induced atmospheric increments in DA_MET_BC_simult are likely due to the stronger constraints imposed by the atmospheric observations. These can moderate the impact of BC observations during the assimilation process. This behavior also highlights the importance of properly specifying the observation error covariance matrix. In future work, we plan to further examine the specification of the BC observation errors and their impact on assimilation performance.

The revised text now reads:

"…The differences in analysis increments of the four atmospheric variables caused by BC assimilation between DA_MET_BC_simult and DA_BC/DA_MET_then_BC may be attributed to the stronger constraints imposed by the atmospheric observations. In both DA_MET_then_BC and DA_BC, only BC surface observations are incorporated during the BC assimilation step. At this stage, the system relies solely on BC observations to correct the initial field. In the absence of atmospheric observations, BC observations play a dominant role, leading to larger analysis increments of atmospheric variables. In contrast, in

DA_MET_BC_simult, both operational meteorological observations and BC surface observations are assimilated simultaneously. In this scenario, atmospheric observations may provide additional constraints on the adjustment of atmospheric fields, thereby moderating the impact of BC observations during the assimilation process. As a result, a more balanced adjustment of atmospheric variables is achieved in DA_MET_BC_simult. This behavior also highlights the importance of properly specifying the observation error covariance matrix. In future work, we plan to further examine the specification of the BC observation errors and their impact on assimilation performance."

Once again, we thank the reviewer for the thoughtful comments and constructive suggestions.